# Methionine restriction constrains lipoylation and activates mitochondria for nitrogenic synthesis of amino acids

Wen Fang[1,5], Liu Jiang [1,5], Yibing Zhu[1,5], Sen Yang[1], Hong Qiu[1], Jiou Cheng [1], Qingxi Liang[1,2], Zong-cai Tu[2,3] & Cunqi Ye [1,4] ✉

Methionine restriction (MR) provides metabolic benefits in many organisms. However, mechanisms underlying the MR-induced effect remain incompletely understood. Here, we show in the budding yeast *S. cerevisiae* that MR relays a signal of S-adenosylmethionine (SAM) deprivation to adapt bioenergetic mitochondria to nitrogenic anabolism. In particular, decreases in cellular SAM constrain lipoate metabolism and protein lipoylation required for the operation of the tricarboxylic acid (TCA) cycle in the mitochondria, leading to incomplete glucose oxidation with an exit of acetyl-CoA and α-ketoglutarate from the TCA cycle to the syntheses of amino acids, such as arginine and leucine. This mitochondrial response achieves a trade-off between energy metabolism and nitrogenic anabolism, which serves as an effector mechanism promoting cell survival under MR.

Methionine is a sulfur-containing amino acid essential in mammals, and an adequate amount must be obtained from food to sustain life[1–3]. Dietary restriction of this amino acid has been shown to extend lifespan across various species ranging from the single-celled eukaryote budding yeast to rodent models[4–9]. Methionine restriction (MR) also represents a promising nutritional approach that can enhance metabolic health and treatment efficacy of diseases such as cancer[10–13]. However, cellular actions and mechanisms underlying the physiological benefits of MR remain incompletely understood.

After entering cells, methionine is readily activated by ATP and converted to S-adenosylmethionine (SAM). All the moieties at the sulfur of SAM can become leaving groups to nucleophilic substrates for cellular biosynthesis and regulation[14,15]. SAM is a prominent effector of MR as many SAM-dependent methylation processes are vital for biosynthesis and regulation[16]. For example, SAM depletion can signal nutrient starvation and enable adaptive transition by the PP2A methylation-dependent regulation of autophagy[17], modulation of tRNA thiolation[18], and SAM-sensitive histone methylation[19–23]. These SAM-responsive mechanisms can function to maintain cell fitness for stem cells[22,24],

immune cells[25,26], and cancer cells[13,27]. Following SAM-dependent methylation, the sequential transmethylation-transsulfuration reaction generates cysteine, a hub metabolite that supports the synthesis of sulfur-containing molecules, including glutathione, coenzyme A, and iron-sulfur clusters[28–30]. SAM is also required for polyamine synthesis[31] and exploited by enzymes from the radical SAM superfamily, such as biotin and lipoate synthases[32,33]. While a broad swath of metabolic processes is linked to methionine metabolism, it is unsurprising that this complex network is highly controlled to ensure its functionality[34,35].

Cellular regulatory feedback circuits are activated under MR to preserve methionine and SAM. For example, maintaining methionine and SAM homeostasis in yeast cells is under transcriptional regulation, by which ubiquitylation modulates the activity of the transcriptional factor Met4 for expressing methionine and SAM biosynthetic enzymes[36,37]. In mammals, SAM depletion activates a posttranscriptional mechanism whereby N6-methyladenosine methylation regulates intron detention and thus gene expression of a major SAM synthetase[38,39]. However, the autonomous replenishment of methionine and SAM may have veiled discovery of metabolic coordination

[1]Zhejiang Provincial Key Laboratory for Cancer Molecular Cell Biology, Life Sciences Institute, Zhejiang University, Hangzhou, China. [2]National R&D Center for Freshwater Fish Processing, Jiangxi Normal University, Nanchang 330022, China. [3]State Key Laboratory of Food Science and Technology, Nanchang University, Nanchang 330047, China. [4]Kidney Disease Center, The First Affiliated Hospital, Zhejiang University School of Medicine, Hangzhou, China. [5]These authors contributed equally: Wen Fang, Liu Jiang, Yibing Zhu. ✉e-mail: yecunqi@zju.edu.cn

activated by MR. Here we explore metabolic changes and regulation in yeast cells by genetically blocking endogenous synthesis for replenishment. Under these resultant stringent conditions restricting methionine and/or SAM, we report an unforeseen metabolic role of mitochondria in adapting to cellular SAM depletion. We show that glucose oxidation is attuned from energy production under MR to synthesis of particular amino acids, such as arginine and leucine. Mechanistically, we demonstrate that lipoate biosynthesis, a mitochondrial radical SAM-dependent process, is constrained by MR, which conveys the SAM depletion signal to the dysregulation of protein lipoylation that reprograms mitochondrial acetyl-CoA for nitrogenic anabolism.

## Results

### Deprivation of cellular SAM leads to the accumulation of nitrogenous metabolites

To block the autonomous recovery of cellular methionine or SAM, we deleted the methionine synthase Met6 (*met6Δ*) or both SAM synthetases Sam1 and Sam2 (*sam1Δsam2Δ*) in the prototrophic yeast strain CEN.PK (Fig. 1a). In the presence of methionine or SAM, the growth of *met6Δ* and *sam1Δsam2Δ* cells was similar to wild type (WT), but was arrested after the respective nutrient was removed from the culture media (Fig. 1b and Supplementary Fig. 1a). Under methionine starvation, methionine was quickly depleted with a gradual reduction of SAM to less than 20% in *met6Δ* cells (Fig. 1c). Similarly, under SAM starvation, SAM was reduced to about 20% in *sam1Δsam2Δ* cells, however, with a profound increase in methionine content (Fig. 1c). These auxotrophic mutants exhibited varying fluctuations in S-adenosylhomocysteine (SAH) and homocysteine under methionine or SAM starvation (Supplementary Fig. 1b). In particular, SAH levels were rapidly reduced while homocysteine increased, both partially restored (Supplementary Fig. 1b), possibly due to the regulation of SAM-dependent methylation reactions and excretion of excessive homocysteine into growth media[19,23]. The deletion mutants were thus exploited to reveal metabolic responses to this sustained MR and distinguish between methionine depletion versus SAM depletion as an MR effector.

We performed targeted metabolomics to profile metabolic alterations over time after removing exogenous methionine or SAM from these mutants. The KEGG (Kyoto Encyclopedia of Genes and Genomes) pathway analysis enriched many pathways involving nitrogenous metabolites (Fig. 1d). Among them, the most significant were arginine biosynthesis and alanine/aspartate/glutamate metabolism (Fig. 1d and Supplementary Table 1). Nearly all amino acids accumulated in the auxotrophic mutants when methionine[40] or SAM was removed (Fig. 1e). We also observed an increase in nucleoside and nucleobase levels in the *met6Δ* mutant and an increase in nucleotide levels in *sam1Δsam2Δ* (Supplementary Fig. 1c). This difference in nucleotide metabolism might be caused by disparate methionine abundance in the mutants.

We next focused on the common response of amino acid accumulation to SAM depletion. The increase in amino acids was SAM-dependent (Fig. 1g and Supplementary Fig. 1d). Withdrawal of methionine did not cause amino acid accumulation in WT cells, in which cellular SAM was partially restored and less depleted (Supplementary Fig. 1e, f). Likely, MR-activated transcriptional programs replenished SAM for maintaining metabolic homeostasis[41,42]. To investigate whether amino acid accumulation was caused by decreased consumption associated with arrested cell growth or by active synthesis, we performed tracing experiments by switching logarithmically growing *met6Δ* cells to methionine-free medium containing $^{15}$N-ammonium sulfate as the sole nitrogen source. We found that the heavy isotope of $^{15}$N-nitrogen was actively incorporated into amino acids at different rates within 30 min (Fig. 1i). More than 80% of glutamate (E), glutamine (Q), glycine (G), aspartate (D), and

asparagine (N) were replaced by the newly synthesized species (Fig. 1i). Because producing these amino acids needs metabolic inputs from mitochondria (Fig. 1h), such as cataplerotic reactions that dispose of TCA cycle intermediates, the mass isotopologue distribution analysis indicates that the mitochondria can respond to SAM depletion and activate amino acid anabolism, a process acquiring nitrogen in its reduced form of ammonia.

### Arginine biosynthesis actively assimilates ammonia under MR

To quantitatively analyze how much ammonia is assimilated into different amino acids under MR, we first estimated cellular concentrations of amino acids using a reverse stable isotope labeling (RIL) method. In the WT prototrophic yeast cells grown in synthetic-defined (SD) minimal medium, the most abundant amino acids were arginine and glutamine, with their intracellular concentrations around 1 mM (Supplementary Fig. 1g). Comparative analysis showed that *met6Δ* cells contained higher amounts of arginine, with a down-shift in glutamine when methionine was supplemented for growth (Supplementary Fig. 1g). We next calculated the amounts of $^{15}$N-ammonia incorporated into each amino acid in *met6Δ* cells over the period of MR, taking into account the amino acid concentration, the mass isotopologue distribution, and the number of nitrogen atoms in each amino acid. Interestingly, we found that $^{15}$N-ammonia was most readily incorporated into arginine (Fig. 1j). Strikingly, after 30 min of MR, a single yeast cell absorbed ~5 mM ammonia of its own cell volume into arginine. This amount was about 170% of the sum ammonia incorporated into all other amino acids (Fig. 1j). The ammonia consumed by arginine biosynthesis under MR would be even more because many nitrogen-containing intermediates, such as ornithine and citrulline, are generated in this biosynthetic pathway (Supplementary Fig. 1h). Consistent with this, the steady levels of arginine metabolites were mostly increased over time in both *met6Δ* and *sam1Δsam2Δ* mutants after removing methionine or SAM (Fig. 1f and Supplementary Fig. 1j). Exogenous SAM abolished or lessened such increases in *sam1Δsam2Δ* and *met6Δ* cells, respectively (Fig. 1g and Supplementary Fig. 1i), indicating that the upregulation of arginine metabolism was also SAM-dependent. Furthermore, the heavy isotope of $^{15}$N-nitrogen was rapidly incorporated into arginine and its upstream metabolites glutamate, acetyl-glutamate, acetyl-ornithine, ornithine, and argininosuccinate during the 2 h of MR (Fig. 1k), confirming active synthesis of arginine in the *met6Δ* cells. These findings collectively indicate that SAM depletion activates arginine biosynthesis to assimilate ammonia.

### Arginine synthesis promotes cell survival under MR

To explore metabolic benefits associated with MR-induced arginine synthesis, we measured the survival of WT and *met6Δ* cells in minimal medium. The survival rate was calculated by comparing viable cells 24 h after growth saturation to an 8-day prolonged culture. We found that MR enhanced the survival of *met6Δ* cells, and this was attenuated by the disruption of arginine synthesis with *arg7Δ* (Fig. 1l and Supplementary Fig. 1k). Consistent with increased survival rates, *met6Δ* cells exhibited increased autophagy under MR, and this activation of autophagy was mitigated by methionine supplementation and *arg7Δ* (Supplementary Fig. 1l). Thus, under this sustained MR, arginine synthesis can improve cell survival possibly by activation of autophagy.

### Relocation of SAM synthesis into the nucleus and the mitochondria leads to the opposite regulation of arginine synthesis

To understand how cytosolic deprivation of SAM perturbs mitochondrial functions and arginine synthesis, we constructed four mutant strains to genetically manipulate the source of mitochondrial SAM (Fig. 2a) that include (1) deletion of one of the two SAM synthetase (*sam2Δ*), (2) sequestration of cytosolic SAM synthetase into the nucleus using a nuclear localization signal (NLS) (Sam1-NLS-GFP/

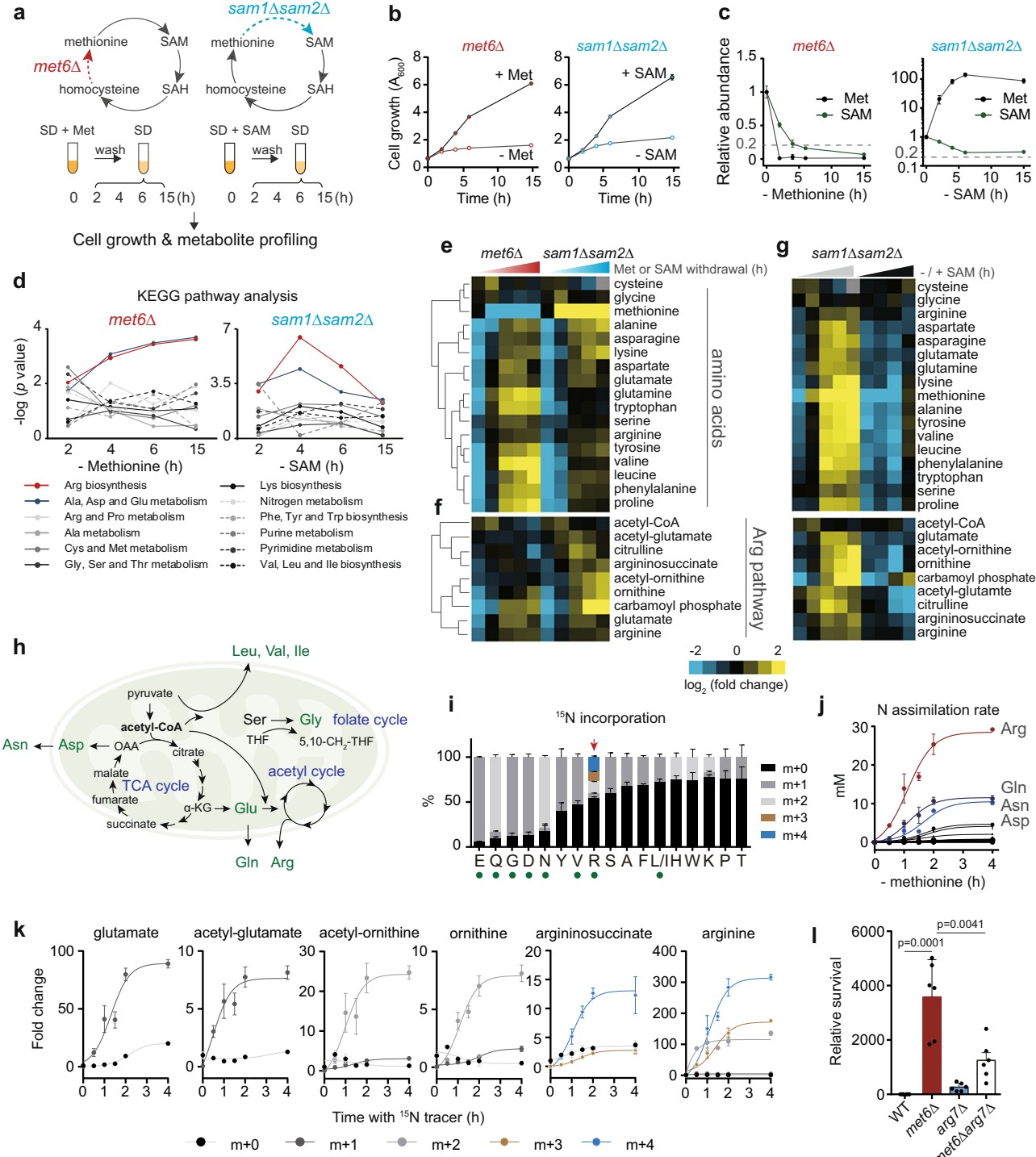

**Fig. 1 | Cellular SAM decrease under MR activates mitochondrial metabolism to assimilate ammonia via arginine synthesis. a** Schematic of the experimental design. The methionine cycle was blocked by *met6Δ* or *sam1Δsam2Δ*. **b** Growth curves of *met6Δ* and *sam1Δsam2Δ* cells in culture media with or without methionine and SAM. Data are represented as mean ± SD (*n* = 3, *n* = biologically independent samples). **c** Relative abundances of cellular methionine and SAM in *met6Δ* and *sam1Δsam2Δ* cells under methionine or SAM starvation. Data are represented as mean ± SD (*n* = 3, *n* = biologically independent samples). **d** KEGG pathway analysis of metabolites altered in *met6Δ* and *sam1Δsam2Δ* cells under methionine or SAM starvation (*n* = 3, *n* = biologically independent samples). Statistical analysis was performed using one-tailed hypergeometric test. **e–g** Heatmap depicting abundances of (**e**) amino acids and (**f**) arginine metabolites in *met6Δ* and *sam1Δsam2Δ* cells under methionine or SAM starvation, (**g**) and their abundances in *sam1Δsam2Δ* cells with or without SAM starvation. Data are represented as average of three

biological replicates. **h** Mitochondrial metabolic pathways for amino acid biosynthesis. **i** The percent abundance of ¹⁵N-incorporated amino acids in *met6Δ* cells after 30 min MR. Data are represented as mean ± SD (*n* = 3, *n* = biologically independent samples). Green dots highlight the amino acids with their biosynthesis occurring partly in the mitochondria. **j** The rate of ammonia assimilation into amino acids in *met6Δ* cells under MR. Data are represented as mean ± SD (*n* = 3, *n* = biologically independent samples) and fit with the nonlinear curves. **k** Relative abundances of isotopic metabolites in the arginine biosynthesis pathway for *met6Δ* cells under MR with ¹⁵N tracer. Data are represented as mean ± SD (*n* = 3) and fit with the nonlinear curves. **l** Relative survival rates of WT, *met6Δ*, *arg7Δ*, and *met6Δarg7Δ* under MR. Data are represented as mean ± SD (*n* = 6, *n* = biologically independent samples). Statistical analysis was performed using two-tailed Student's *t* test. Source data are provided as a Source Data file.

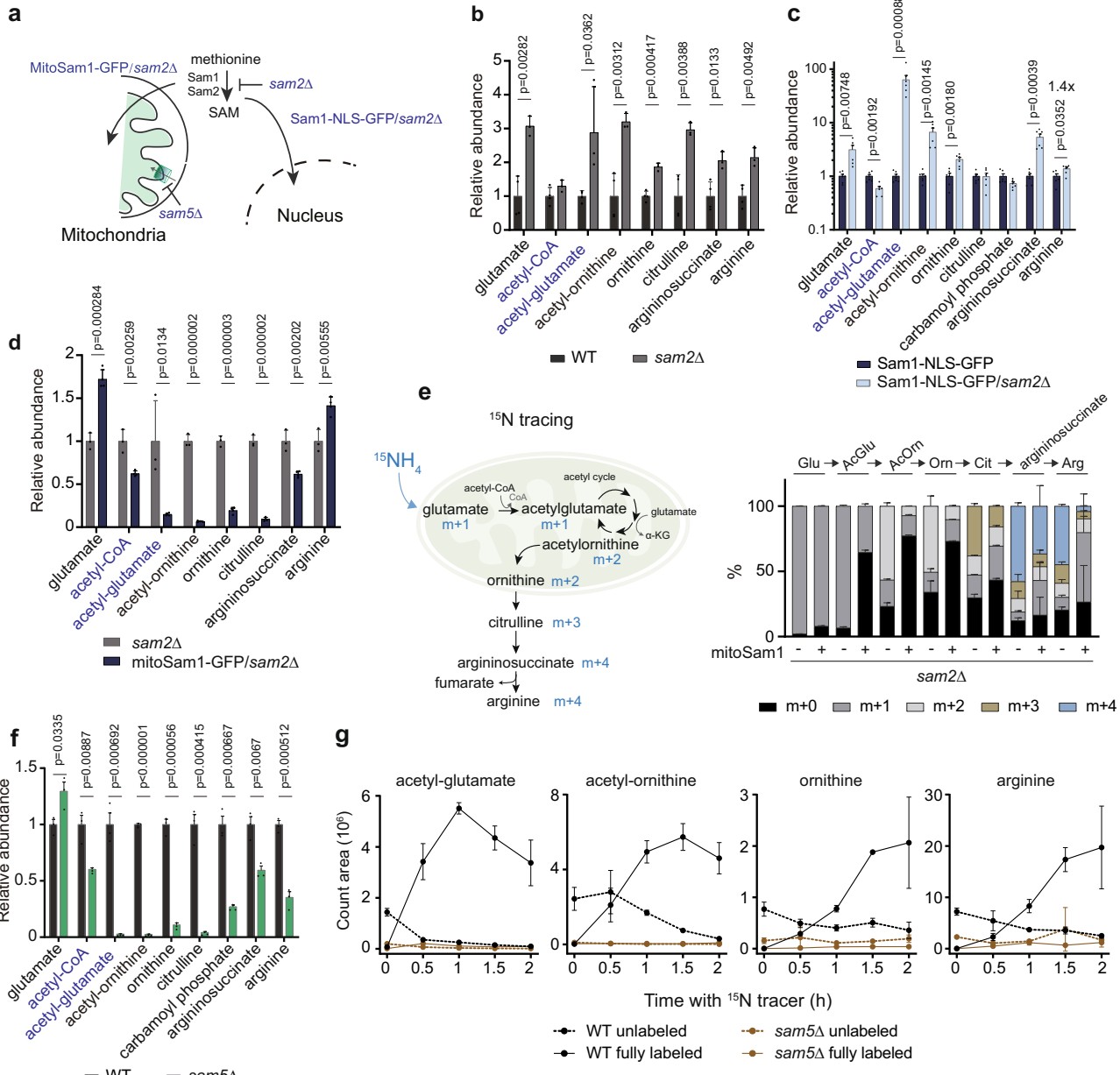

**Fig. 2 | The spatiotemporal supply SAM for the mitochondria imposes regulation of arginine synthesis. a** Schematic of four genetic mutants to manipulate the source of mitochondrial SAM: *sam2Δ*, Sam1-NLS-GFP/*sam2Δ* (*nSam1*), mito-Sam1-GFP/*sam2Δ* (*mSam1*), and *sam5Δ*. **b** Relative abundances of arginine metabolites in WT and *sam2Δ* cells. Data are represented as mean ± SD (*n* = 4, *n* = biologically independent samples). Statistical analysis was performed using two-tailed Student's *t* test. **c** Relative abundances of arginine metabolites in Sam1-NLS-GFP and Sam1-NLS-GFP/*sam2Δ* cells. Data are represented as mean ± SD (*n* = 6, *n* = biologically independent samples). Statistical analysis was performed using two-tailed Student's *t* test. **d** Relative abundances of arginine metabolites in *sam2Δ* and Sam1-NLS-GFP/*sam2Δ* cells. Data are represented as mean ± SD (*sam2Δ*, *n* = 3; Sam1-NLS-GFP/sam2Δ, *n* = 4, *n* = biologically independent samples). Statistical

analysis was performed using two-tailed Student's *t* test. **e** Left: Schematic of $^{15}$N tracing of arginine biosynthesis. Right: The percent abundance of $^{15}$N-labeled arginine metabolites in *sam2Δ* and mitoSam1-GFP/*sam2Δ* cells after 1 h. Data are represented as mean ± SD (*n* = 4, *n* = biologically independent samples). **f** Relative abundances of arginine metabolites in WT and *sam5Δ* cells. Data are represented as mean ± SD (*n* = 3, *n* = biologically independent samples). Statistical analysis was performed using two-tailed Student's *t* test. **g** Spectra intensities of labeled and unlabeled acetyl-glutamate, acetyl-ornithine, ornithine, and arginine in WT and *sam5Δ* cells. Data are represented as mean ± SD (*n* = 4, *n* = biologically independent samples). Statistical analysis was performed using two-tailed Student's *t* test. Source data are provided as a Source Data file.

*sam2Δ*, *nSam1*) (Supplementary Fig. 2a, b), (3) relocation of SAM synthetase into the mitochondrial matrix (mitoSam1-GFP/*sam2Δ*, *mSam1*), and (4) deletion of the mitochondrial SAM transporter gene *SAM5* (*sam5Δ*)[43]. Just as in *met6Δ* and *sam1Δsam2Δ* mutants, *SAM2* deletion decreased total SAM levels and led to an increase in arginine and arginine metabolites (Fig. 2b and Supplementary Fig. 2c). When SAM was supplied from the nucleus in the *nSam1* mutant, acetyl-glutamate

levels increased by more than 60-fold, accompanied by a 40% increase in arginine content and elevated levels of the upstream metabolites such as acetyl-ornithine, ornithine, and argininosuccinate (Fig. 2c), indicating an increase in arginine biosynthesis in this mutant. In contrast, the biosynthesis of arginine decreased when SAM synthesis was relocated to the mitochondrial matrix in the *mSam1* mutant. In this mutant, we observed significant reductions in the steady levels of

intermediary metabolites in arginine biosynthesis (Fig. 2d) and the decreased incorporation of [15]N into arginine and the intermediates, revealed by [15]N tracing experiments (Fig. 2e). Therefore, the contrasting effect on arginine biosynthesis resulting from relocalizing of SAM synthesis to the nucleus and the mitochondria suggests that the spatiotemporal supply of SAM for mitochondria is informative to the regulation of mitochondrial metabolism.

## SAM-dependent activation of arginine synthesis requires glutamate acetylation in the mitochondria

To our surprise, blocking mitochondrial SAM uptake by *sam5Δ* led to very low abundances in the mitochondria-exclusive metabolites acetyl-glutamate and acetyl-ornithine, with overall reductions in cellular contents of ornithine, citrulline, carbamoyl phosphate, argininosuccinate, and arginine (Fig. 2f), reflecting a strong defect in arginine biosynthesis. Consistent with this, we found that in [15]N -labeled and unlabeled arginine metabolites remained very low in *sam5Δ* within the 2 h tracing, while in WT cells the unlabeled levels of arginine and intermediary metabolites, acetyl-glutamate, acetyl-ornithine, and ornithine were replaced gradually by their [15]N-labeled counterparts (Fig. 2g).

While arginine synthesis was decreased in both *sam5Δ* and *mSam1* mutants, what might be the step in arginine synthesis responsive to SAM dysregulation? We noticed that the turnover of acetyl-glutamate was very rapid in WT cells (Fig. 2g), underpinning a mitochondrial step actionable for an immediate response to SAM depletion. Indeed, the changes in acetyl-glutamate were consistent with the activities of arginine synthesis in all four mutants. Acetylation commits the fate of mitochondrial glutamate toward arginine synthesis in the yeast *S. cerevisiae*[44], as the product acetyl-glutamate irreversibly enters the acetyl cycle and exits to form ornithine, a precursor of arginine (Fig. 2e). Therefore, it is possible that *sam5Δ* and *mSam1* mutants have deficiencies in mitochondrial acetyl-CoA, which restricts glutamate activation for arginine synthesis.

In addition to perturbations in arginine biosynthesis, *sam5Δ*, *nSam1*, and *mSam1* cells growing in the nutrient-limiting SD condition exhibited abnormal intracellular pools of amino acids and growth defects of different severity (Supplementary Fig. 2d–h), highlighting the importance of spatial SAM regulation in orchestrating amino acid homeostasis and cell proliferation. Thus, it was unsurprising that arginine alone was insufficient to fully rescue the growth defects (Supplementary Fig. 2d). Interestingly, *sam5Δ*, *nSam1*, and *mSam1* did not affect the total amounts of cellular SAM (Supplementary Fig. 2c). In contrast, the *nSam1* mutant showed profound increases in SAH and methionine (Supplementary Fig. 2c and 2g), underlying the importance of compartment-specific metabolism of methionine. However, technical limitations preclude accurate measurement of mitochondrial SAM levels. We propose that MR-induced cellular SAM deprivation restricts mitochondrial SAM availability and relays a signal to trigger the acetylation of glutamate in the mitochondria, a process contingent on mitochondrial acetyl-CoA production. We term this mitochondrial SAM-induced response **mitoSIR**.

## Acetyl-CoA and α-ketoglutarate (αKG) exit the TCA cycle to support arginine biosynthesis that acquires ammonia upon MR

We next investigated how glutamate was activated via acetylation for arginine synthesis under MR and, therefore, ammonia assimilation, a process reflecting the escape of glucose carbon from mitochondrial oxidation in the form of acetyl-CoA and αKG (Fig. 3a). To pinpoint the steps of mitochondrial metabolism sensitive to SAM depletion, we first determined which enzyme was responsible for cataplerotically removing αKG from the TCA cycle (Fig. 3a). *gdh1Δ*, but not *gdh3Δ* or *glt1Δ*, decreased glutamate, acetyl-glutamate, citrulline, carbamoyl phosphate, argininosuccinate, and arginine levels in *met6Δ* cells under MR (Fig. 3b and Supplementary Fig. 3a). Only *gdh1Δ* lowered the MR-induced amino acid accumulation in *met6Δ*, thus diminishing

ammonia assimilation into amino acids (Supplementary Fig. 3b, c). Therefore, the NADP+-dependent glutamate dehydrogenase Gdh1 contributed to providing glutamate for arginine synthesis. While *gdh1Δ* prevented the increase in acetyl-glutamate, arginine levels could still increase in this mutant under MR (Supplementary Fig. 3d), suggesting that other pathways could also contribute to the synthesis of arginine. However, Gdh1 is not a mitochondrial enzyme[45] (Supplementary Fig. 3e, f) and thus unlikely to serve as a direct sensor to activate glutamate for arginine synthesis.

We next interrupted the TCA cycle by *idh1Δ* at the step of isocitrate oxidation to αKG (Fig. 3a and Supplementary Fig. 3g). The resulting decreases in glutamate, acetyl-glutamate, and other arginine metabolites by *idh1Δ* were associated with acetyl-CoA surplus (Fig. 3c), indicating that the operative TCA cycle supplying αKG was required for this glutamate-to-arginine metabolic shunt, whereas an increase in acetyl-CoA level was not sufficient. To further identify TCA steps responsive to MR, we performed [U-13C6] glucose tracing experiments to analyze the mass isotopologue distribution of TCA cycle metabolites in WT and *met6Δ* cells (Fig. 3d). Critically, [13]C incorporation into the TCA cycle was not saturated at 10 min as the labeled fractions increased at 60 min (Fig. 3e and Supplementary Fig. 3h), which warranted this [13]C-based diagnosis of the TCA cycle. Indeed, the turnover rates of TCA cycle metabolites under MR varied in WT and *met6Δ* cells, particularly at the axis of acetyl-CoA-citrate-aconitate-αKG-succinate (Fig. 3e). The *met6Δ* cells displayed a decrease in the ratio of m + 6 citrate to m + 2 acetyl-CoA and an increase in the ratio of m + 4 succinate to m + 5 αKG, suggesting that both the entry of acetyl-CoA into the TCA cycle and the oxidative decarboxylation of αKG became disrupted under MR (Fig. 3d).

We then examined if acetyl-CoA uncoupled from the TCA cycle might enter other processes in the mitochondria, such as the acetyl cycle (Fig. 3g). The unlabeled and newly synthesized acetyl-CoA molecules were undistinguishably utilized in WT cells under MR, as evidenced by similar increases in m + 5 and m + 7 acetyl-glutamates (Fig. 3h). In *met6Δ*, the abundance of m + 5 acetyl-glutamate was dominant and nearly doubled compared to WT (Fig. 3h). This finding suggests that unlabeled acetyl-CoAs liberated from the TCA cycle upon MR may be quickly funneled to acetylating glutamate, thus irreversibly pulling αKG out of the TCA cycle by the channeling of glutamate into arginine synthesis. After 1 h of MR, *met6Δ* cells indeed enriched more arginines containing [13]C-derived glucose carbon (Fig. 3i), underlying a co-opting carbon flow into this nitrogenous metabolite. Following the disposal of oxidized glucose carbons, extra glucose was later stored in trehalose, a disaccharide synthesized under many stress conditions (Fig. 3j and Supplementary Fig. 3i)[46].

## Lipoylation of pyruvate dehydrogenase (PDH) is required for arginine synthesis

Corresponding to the two defective steps identified above, pyruvate dehydrogenase (PDH) and αKG dehydrogenase (KDH) require lipoic acid as a covalently-bound coenzyme for catalysis (Fig. 4a). We reasoned that lipoic acid might become a limiting factor under MR and act as a signal to reduce the carbon oxidation capacity of the TCA cycle and repurpose a carbon flow for ammonia assimilation. This is because lipoic acid is a sulfur-containing fatty acid synthesized de novo in the mitochondria of eukaryotes via a multistep reaction requiring SAM and iron-sulfur clusters[47]. The 2-keto acid dehydrogenases Lat1 and Kgd2 and the glycine decarboxylase Gcv3 are the three lipoyl enzymes in the mitochondria of *S. cerevisiae* (Fig. 4a). Lipoylation is matured via Lip5-mediated sulfur insertion into an octanoyl group attached to Gcv3, and the lipoyl group formed on Gcv3 can be transferred to modify Lat1 and Kgd2 by Lip3 (Fig. 4a)[48].

We examined if arginine synthesis was affected in deletion mutants disrupting lipoyl enzymes or lipoate metabolism. We found that arginine metabolites were greatly decreased by *lat1Δ*, *gcv3Δ*, *lip3Δ*,

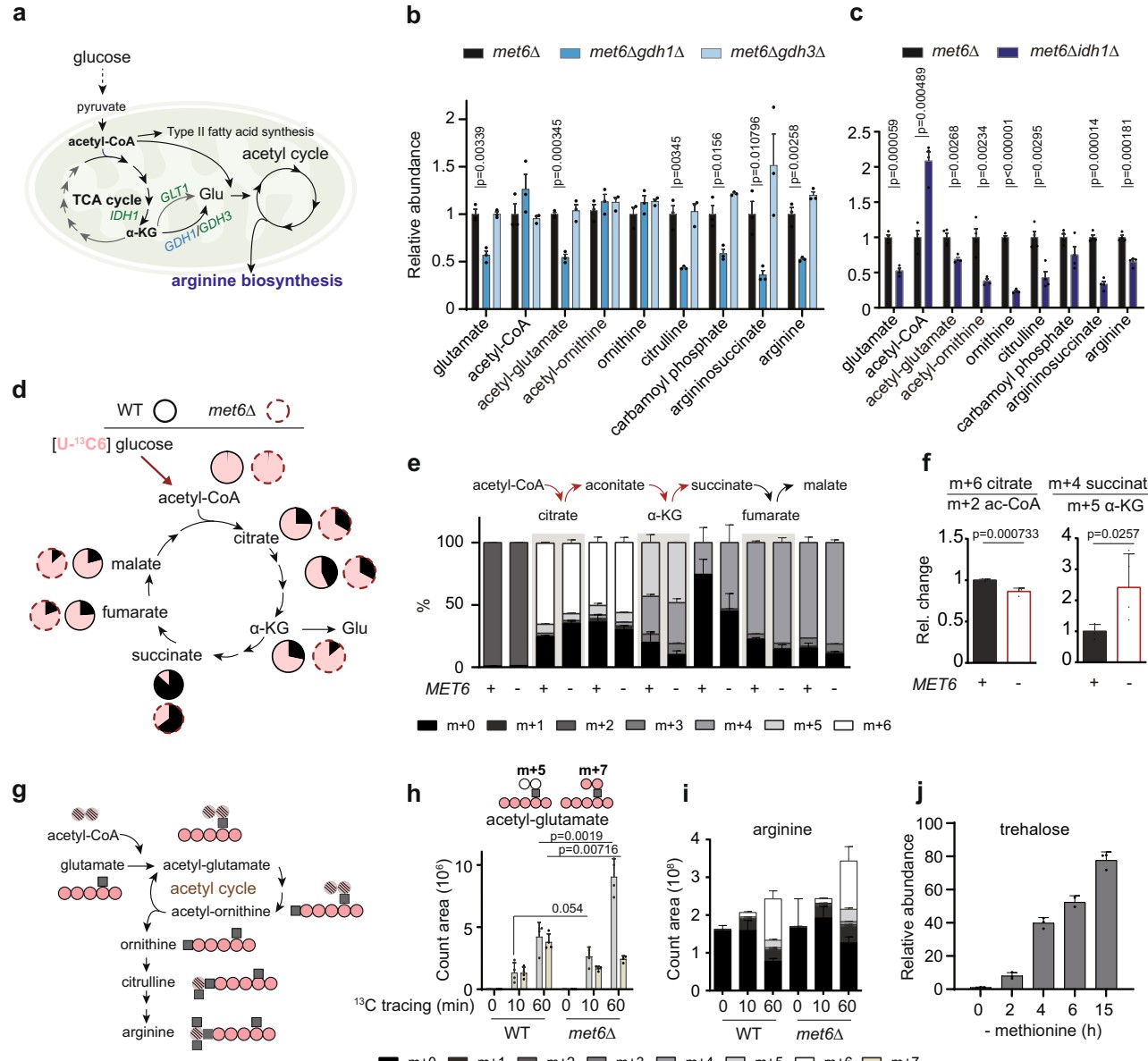

**Fig. 3 | Glucose-derived acetyl-CoA and αKG exit the TCA cycle to support arginine synthesis upon MR. a** Schematic of mitochondrial metabolic processes to fuel arginine synthesis. **b, c** Relative abundances of arginine metabolites in (**b**) *met6Δ*, *met6Δgdh1Δ*, *met6Δgdh3Δ*, and (**c**) *met6Δ*, *met6Δidh1Δ* cells harvested 4 h after MR. Data are represented as mean ± SD (Fig. 3b, *n* = 3; Fig. 3c, *n* = 4, *n* = biologically independent samples). Statistical analysis was performed using two-tailed Student's *t* test. **d** Schematic of [U-¹³C6] glucose tracing of the TCA cycle in WT and *met6Δ* cells. The circle charts plotting the percent abundance are shown in (**e**). **e** The percent abundance of ¹³C-labeled TCA metabolites in WT and *met6Δ* cells after 10 min MR. Data are represented as mean ± SD (*n* = 4, *n* = biologically independent samples). Data are representative of two independent experiments. **f** Relative change in ratios of m + 6 citrate to m + 2 acetyl-CoA and m + 4 succinate to

m + 5 αKG. Data are representative of two independent experiments. Data are represented as mean ± SD (*n* = 4, *n* = biologically independent samples). Statistical analysis was performed using two-tailed Student's *t* test. **g** Schematic of ¹³C tracing of the acetyl cycle and arginine synthesis. **h** Spectra intensities of isotopic acetyl-glutamates (m + 5 and m + 7) in WT and *met6Δ* cells in ¹³C tracing. Data are represented as mean ± SD (*n* = 4, *n* = biologically independent samples). Statistical analysis was performed using two-tailed Student's *t* test. **i** Spectra intensities of different arginine isotopologues in cells under MR with [U-¹³C6] glucose. Data are represented as mean ± SD (*n* = 4, *n* = biologically independent samples). **j** Relative abundances of trehalose in *met6Δ* cells after removing methionine from growth medium. Data are represented as mean ± SD (*n* = 3, *n* = biologically independent samples). Source data are provided as a Source Data file.

and *lip5Δ*, albeit arginine content was only diminished in the stationary phase (Fig. 4b and Supplementary Fig. 4a, b). It is possible that the steady level of arginine in the log phase can be maintained by the transcriptional upregulation of the biosynthesis pathway (Supplementary Fig. 4c). We further performed ¹⁵N tracing and found that arginine synthesis was indeed compromised in these mutants, as the rates of ¹⁵N incorporation into arginine were decreased (Fig. 4c). Furthermore, total cellular acetyl-CoA levels tended to decrease in *lat1Δ* and *gcv3Δ* cells (Fig. 4b), and the decreases became more pronounced

in the stationary phase (Supplementary Fig. 4a). ¹³C tracing with [U-¹³C6] glucose revealed a decreased turnover of acetyl-CoA in *lat1Δ* (Fig. 4d), confirming defective decarboxylation of pyruvate to acetyl-CoA in the mitochondria. The turnover of acetyl-CoA was also decreased in the *gcv3Δ* mutant, but not in *kgd2Δ* (Fig. 4d). Because Gcv3 is a lipoyl carrier upstream of Lat1 and Kgd2, these observations suggest that the lipoylation of PDH/Lat1 that ensures production of acetyl-CoA from pyruvate is required for the activation of glutamate in the mitochondria for arginine synthesis. In agreement with this, the

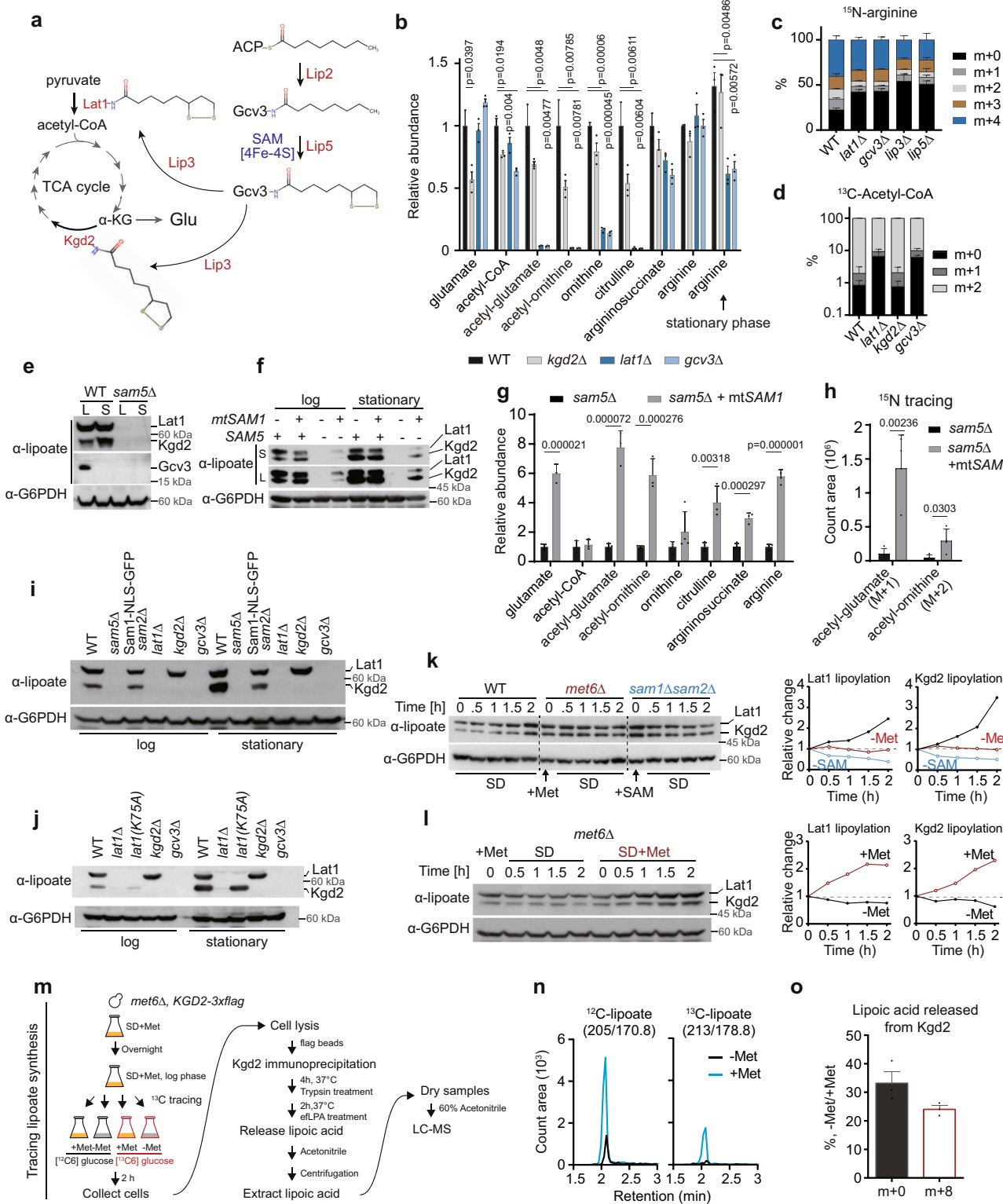

*mSam1* mutant with decreased arginine synthesis also exhibited a decrease in Lat1 lipoylation and a defect in the production of acetyl-CoA (Supplementary Fig. 4d, e).

Arginine deficiency in the lipoylation deficient mutants is consistent with the loss of protein lipoylation in *sam5Δ* cells (Fig. 4e). Overexpression of a SAM synthetase in the mitochondrial matrix of *sam5Δ* (mito*SAM1*) restored protein lipoylation and the growth defects (Fig. 4f and Supplementary Fig. 4f, g), and elevated acetyl-glutamate, arginine, and other arginine metabolites (Fig. 4g), with evident

increases in the quantities of $^{15}$N-incorporated acetyl-glutamate and acetyl-ornithine, compared to the parental mutant *sam5Δ* (Fig. 4h). Therefore, the shortage of mitochondrial SAM is responsible for the deficiencies in lipoylation and arginine biosynthesis in *sam5Δ* cells. The biosynthesis of biotin also requires a radical SAM enzyme in the mitochondria[49], and *sam5Δ* led to biotin auxotrophy (Supplementary Fig. 4h). Decreases in biotin levels under SAM deprivation in *met6Δ* and *sam1Δsam2Δ* cells (Supplementary Fig. 4i) suggest that the mitochondrial SAM radical enzymes have limited access to SAM under MR.

**Fig. 4 | Lipoylation is a sentinel modification responding to cellular SAM decrease to regulate arginine biosynthesis. a** Schematic of lipoate biosynthesis and lipoylation substrate enzymes. **b** Relative abundances of arginine metabolites in WT, $kgd2\Delta$, $lat1\Delta$, and $gcv3\Delta$ cells. Data are represented as mean ± SD ($n = 3$, $n$ = biologically independent samples). Statistical analysis in this figure was performed using two-tailed Student's $t$ test. **c** The percent abundance of [15]N-labeled arginine in indicated cells after 1 h of [15]N tracing. Data are represented as mean ± SD ($n = 4$, $n$ = biologically independent samples). **d** The percent abundance of [13]C-labeled acetyl-CoA in WT, $lat1\Delta$, $kgd2\Delta$, and $gcv3\Delta$ cells after 10 min of [13]C tracing. Data are represented as mean ± SD ($n = 4$, $n$ = biologically independent samples). **e** Western blots assaying protein lipoylation in WT and the $sam5\Delta$ mutant. L: log phase; S: stationary phase. Data are representative of at least three independent experiments. **f** Western blots assaying protein lipoylation in $sam5\Delta$ cells with or without $mtSAM1$ expression. S: short exposure; L: long exposure. Data are representative of at least three independent experiments. **g** Relative abundances of arginine metabolites in the $sam5\Delta$ mutant with or without $mtSAM1$ expression. Data

are represented as mean ± SD ($n = 3$, $n$ = biologically independent samples). Statistical analysis was performed using two-tailed Student's $t$ test. **h** Spectra intensities of isotopic acetyl-glutamate (m + 1) and acetyl-ornitine (m + 2) in $sam5\Delta$ and $mtSAM1$-expressing $sam5\Delta$ cells collected 1 h after shifting to the tracing medium containing [15]N-ammonium sulfate. Data are represented as mean ± SD ($n = 3$, $n$ = biologically independent samples). Statistical analysis was performed using two-tailed Student's $t$ test. **i, j** Western blots assaying protein lipoylation in indicated strains harvested in the log and stationary phases. Data are representative of at least three independent experiments. **k, l** Western blots assaying protein lipoylation under indicated conditions in WT, $met6\Delta$, and $sam1\Delta sam2\Delta$ cells with indicated quantification. Data are representative of three independent experiments. **m** Schematic of the assay of tracing lipoate synthesis (See the detail in Methods). **n, o** The ion chromatogram and quantification of [12]C-lipoate and [13]C-lipoate released from Kgd2 under indicated conditions. Data are represented as mean ± SD ($n = 3$, $n$ = biologically independent samples). Source data are provided as a Source Data file.

Taken together, these results are in line with the idea that mitochondrial SAM is required for lipoate metabolism, which in turn affects arginine biosynthesis.

Notably, $mtSAM1$ expression was not sufficient to fully restore $sam5\Delta$ lipoylation to WT levels and could decrease Lat1 lipoylation compared to WT (Fig. 4f and Supplementary Fig. 4e). This suggests that the spatiotemporal control of SAM supply for mitochondria is critical for the process of lipoylation. The ectopic synthesis of SAM in the $mSam1$ mutant may lead to metabolic state adapted to this low Lat1 lipoylation.

### Lipoylation is a sentinel modification responding to MR

We then wondered if lipoylation of the mitochondrial enzymes might have different sensitivities to alterations in SAM, thus activating a metabolic shunt of acetyl-CoA and αKG for arginine biosynthesis. Lipoylation of the KDH complex enzyme Kgd2 appeared more sensitive to the nuclear SAM supply, and its level was markedly decreased in the $nSam1$ mutant in the stationary phase (Fig. 4i and Supplementary Fig. 4j). In WT cells, lipoylation of Kgd2, but not the protein itself (Supplementary Fig. 4k), was greatly increased in the stationary versus logarithmic phase (Fig. 4e and Supplementary Fig. 4l), indicative of the dynamic nature of Kgd2 lipoylation. Interestingly, the increases in Kgd2 lipoylation over the growth phase were accompanied by decreases in Gcv3 lipoylation (Fig. 4e and Supplementary Fig. 4l), so the lipoyl group might be transferred from Gcv3 to Kgd2 upon entering the stationary phase. Moreover, $LAT1$ deletion abolished Kgd2 lipoylation (Fig. 4i, j), suggesting that Lat1 might serve as a scaffold protein required for the lipoyl transferring from Gcv3 to Kgd2. In agreement with this notion, the Lat1 mutant K75A, which blocks its own lipoylation, had little effect on the lipoylation of Kgd2 (Fig. 4j). With respect to installing lipoyl moieties, it is tempting to speculate that Kgd2, the final recipient of lipoyl groups, is more sensitive to conditions that limit lipoate synthesis, such as MR. Indeed, SAM was required for increases in protein lipoylation, as the increment of Lat1 and Kgd2 lipoylation over time in WT cells under a SAM-sufficient condition was abrogated under SAM deprivation conditions established by $met6\Delta$ and $sam1\Delta sam2\Delta$ (Fig. 4k), and the halted increase in $met6\Delta$ cells could be resumed by simply providing methionine (Fig. 4l). To further examine if MR constrains Kgd2 lipoylation, we established a [13]C tracing method to quantify lipoate released from Kgd2 by trypsin and lipoamidase treatment (Fig. 4m and Supplementary Fig. 4m–o). We found that the unlabeled lipoate and the newly synthesized, fully labeled species released from Kgd2 both were decreased under MR (Fig. 4n, o). Taken together, our findings indicate that MR likely imposes a different constraint on lipoylation substrates, causing the rewiring of the TCA cycle for arginine synthesis. It is worth noting that $kgd2\Delta$ alone was not sufficient to activate glutamate acetylation. Many arginine intermediary

metabolites were decreased moderately in this mutant, possibly due to glutamate deficiency (Fig. 4b).

### *SAM5* deletion alters arginine metabolon to avert leucine biosynthesis

While the responsiveness of lipoate metabolism to MR uncouples a fraction of acetyl-CoA from the TCA cycle, we continued to investigate how this highly reactive molecule was channeled into reactions for arginine biosynthesis in the mitochondria. To study this, we utilized the $sam5\Delta$ mutant as a model, in which the acetyl cycle relies on an alternative source of acetyl-CoA due to the PDH complex deficiency. Inspecting the mass isotopologue distribution of amino acids in the [15]N tracing experiments, we found a surprising ebb in the proportional quantities of m + 1 and m + 2 arginines in the $sam5\Delta$ mutant (Fig. 5a). This labeling feature suggests an arginine metabolon assembled for efficient channeling of arginine intermediary metabolites in the mitochondria (Fig. 5b).

We appended epitope tags to related enzymes Arg2 and Arg7 and found that the difference in their protein levels in WT and $sam5\Delta$ cells were growth phase-dependent (Fig. 5c), suggesting preferential utilization of acetyl-glutamate synthases. Serendipitously, we found that the C-terminal tagging of both $ARG2$ and $ARG5,6$ in WT conferred arginine auxotrophy, an unexpected phenotype that was abolished by deletion of $SAM5$ (Fig. 5d). Notably, the tagging of $ARG2$ or $ARG5,6$ alone in WT or the $sam5\Delta$ mutant did not affect cell growth (Supplementary Fig. 5a). Therefore, this growth defect in WT was synthetic and dependent on the tagging-based disruption of both enzymes. Arg2 and Arg5,6 are known to be involved in forming an arginine metabolon to limit wasteful consumption of acetyl-CoA in the mitochondria[50]. We speculate that the C-terminal tagging in WT possibly disrupts protein interaction between Arg2 and Arg5,6, which is required for this metabolon formation; and this interaction, however, becomes dispensable in the $sam5\Delta$ mutant. Strikingly, the C-terminal tagging of $ARG7$ and $ARG5,6$ turned $sam5\Delta$ into an arginine auxotroph (Fig. 5d and Supplementary Fig. 5a). These findings suggest that the arginine metabolon in WT, likely formed via the protein interaction of the C-terminal regions of Arg5,6 and Arg2, may be reconfigured by $sam5\Delta$ with an interaction between Arg5,6 and Arg7. In the $sam5\Delta$ mutant bearing mitochondrial acetyl-CoA deficiency, an Arg7-mediated metabolon may be more potent to channel acetyl-CoA into the acetyl cycle. This is further supported by a recent MitCOM study of the complexome of yeast mitochondria[51], which confirms the protein assembly of Arg2, Arg7, and Arg5,6 (Supplementary Fig. 5c).

In addition, the deletion of $ARG7$ in $sam5\Delta$ caused a severe growth defect in minimal medium (Fig. 5e and Supplementary Fig. 5b). This growth defect was rescued by a mixture of 20 amino acids (Fig. 5e), but to our surprise, not by arginine alone (Supplementary Fig. 5b). We next dropped out every amino acid individually and found that arginine and leucine were the only two amino acids required for the rescue effect

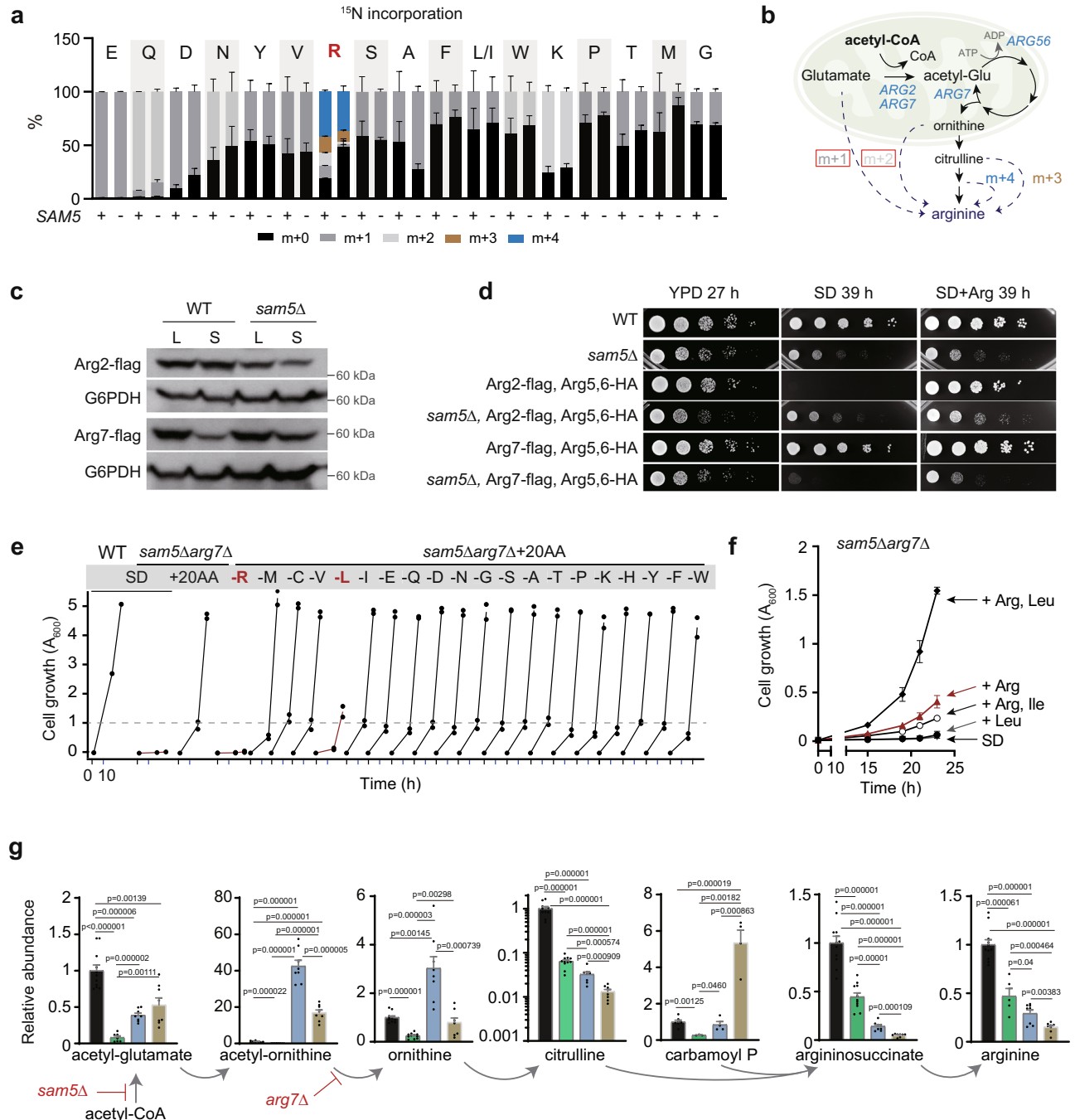

**Fig. 5 | *SAM5* deletion alters arginine metabolon to avert leucine biosynthesis.** **a** The mass isotopologue distribution of amino acids in WT and *sam5Δ* cells after 1 h tracing with [15]N-ammonium sulfate. Data are represented as mean ± SD ($n = 3$, $n$ = biologically independent samples). **b** Schematic of intermediary metabolite-based mass shift corresponding to [15]N-arginine isotopologues. **c** Western blots assaying Arg2 and Arg7 protein abundances in WT and *sam5Δ* cells. Data are representative of at least three independent experiments. **d** Growth of indicated strains on YPD, SD, and SD containing 1 mM arginine for indicated times. Note that epitope-tagging of Arg2, Arg5,6, or Arg7 alone did not affect growth (Supplementary Fig. 5a). **e** Growth curves of WT and *sam5Δarg7Δ* cells. Note that each of 20 amino acids was individually dropped out to examine the ones required for the growth of *sam5Δarg7Δ*. **f** Growth curves of *sam5Δarg7Δ* cells in SD or SD containing indicated amino acid(s). $n = 2$ independent experiments. **g** Relative abundances of arginine metabolites in WT, *sam5Δ*, *arg7Δ*, and *sam5Δarg7Δ* cells. Data are represented as mean ± SD, $n$ = biologically independent samples for each metabolites in WT, *sam5Δ*, *arg7Δ*, and *sam5Δarg7Δ* indicated below in sequence: acetyl-glutamate ($n = 11,7,8,8$), acetyl-ornithine ($n = 8,8,8,8$), ornithine ($n = 11,11,8,8$), citrulline ($n = 11,11,8,8$), carbamoyl phosphate ($n = 7,3,4,4$), arginesuccinate ($n = 11,11,8,8$), arginine ($n = 11,5,8,8$). Statistical analysis was performed using two-tailed Student's $t$ test. Source data are provided as a Source Data file.

(Fig. 5e). Indeed, adding arginine and leucine simultaneously could promote the growth of *sam5Δarg7Δ* cells (Fig. 5f and Supplementary Fig. 5b). These findings suggest that the Arg7-mediated arginine metabolon is linked to leucine biosynthesis.

We then performed metabolic analysis to find out how the syntheses of arginine and leucine were disrupted in the *sam5Δarg7Δ* mutant. Acetyl-ornithine levels were increased by more than 40-fold in *arg7Δ* (Fig. 5g), confirming an ornithine acetyltransferase activity of Arg7[52].

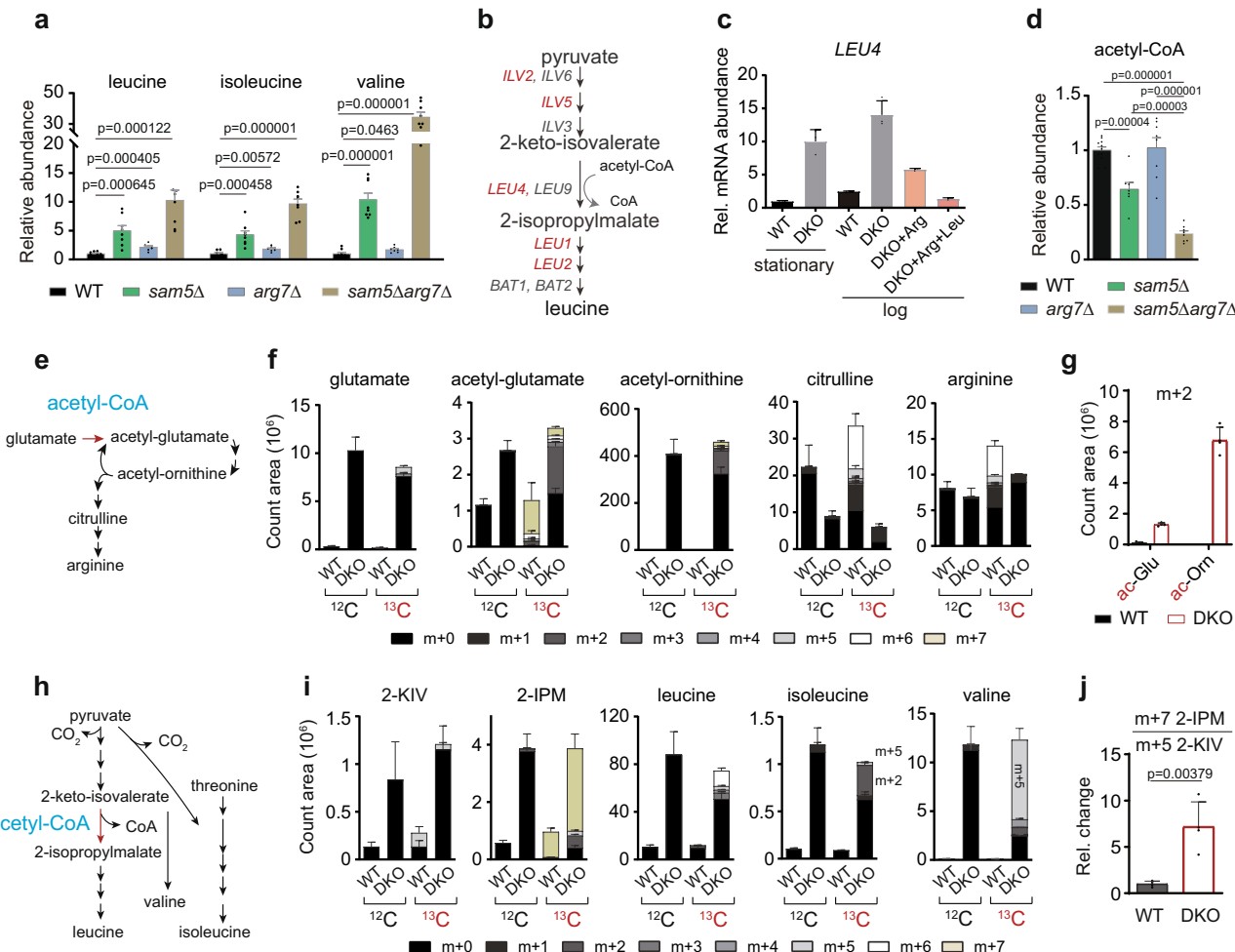

**Fig. 6 | The hierarchy of acetyl-CoA allocation in mitochondrial metabolism.**
**a** Relative abundances of BCAAs in WT, *sam5Δ*, *arg7Δ*, and *sam5Δarg7Δ* cells. Data are represented as mean ± SD ($n = 8$, $n$ = biologically independent samples). Statistical analysis was performed using two-tailed Student's *t* test. **b** The leucine biosynthesis pathway and corresponding metabolic genes at each reaction step. Genes co-regulated under Leu3 are highlighted in red. **c** Relative mRNA transcript levels of *LEU4* in WT and *sam5Δarg7Δ* (DKO) cells. Data are represented as mean ± SD ($n = 3$ biologically independent samples). **d** Relative abundances of acetyl-CoA in WT, *sam5Δ*, *arg7Δ*, and *sam5Δarg7Δ* cells. Data are represented as mean ± SD (WT, $n = 11$; *sam5Δ*, $n = 7$; *arg7Δ*, $n = 8$; *sam5Δarg7Δ*, $n = 8$; $n$ = biologically independent samples). Statistical analysis was performed using two-tailed Student's *t* test. **e** Schematic of arginine biosynthesis, with the acetyl-CoA step

highlighted. **f** Spectra intensities of isotopic metabolites in the arginine biosynthesis pathway. $^{12}$C: before $^{13}$C tracing; $^{13}$C: 1 h after $^{13}$C tracing. Data are mean ± SD ($n = 4$, $n$ = biologically independent samples). **g** Spectra intensities of m + 2 acetyl-glutamate (ac-Glu) and m + 2 acetyl-ornithine (ac-Orn). Data are mean ± SD ($n = 4$, $n$ = biologically independent samples). **h** Schematic of BCAA biosynthesis, with the acetyl-CoA step highlighted. **i** Spectra intensities of isotopic metabolites in the BCAA biosynthesis pathway. $^{12}$C: before $^{13}$C tracing; $^{13}$C: 1 h after $^{13}$C tracing. Data are mean ± SD ($n = 4$, $n$ = biologically independent samples). **j** Relative change of m + 7 2-IPM to m + 5 2-KIV ratio. Data are mean ± SD ($n = 4$, $n$ = biologically independent samples). Statistical analysis was performed using two-tailed Student's *t* test. Source data are provided as a Source Data file.

Citrulline, argininosuccinate, and arginine levels in the *sam5Δarg7Δ* mutant exhibited greater reductions compared to *sam5Δ*, with a surplus in carbamoyl phosphate (Fig. 5g). This aggravating effect of *arg7Δ* on the arginine deficiency in the *sam5Δ* mutant supports that Arg7 probably engages in an alternative metabolon for arginine biosynthesis. In sharp contrast to this arginine deficiency, levels of cellular leucine and the other branched-chain amino acids (BCAAs), isoleucine and valine, were not diminished by either *sam5Δ* or *arg7Δ* (Fig. 6a). In fact, the *sam5Δarg7Δ* mutant exhibited the highest contents of these BCAAs (Fig. 6a), implying an overproduction of leucine rather than a deficiency.

**Acetyl-CoA deviating from the arginine metabolon erroneously activates a transcriptional program, leading to the overproduction of leucine**
We next reasoned if acetyl-CoA occluded from the TCA cycle and the Arg7-mediated arginine metabolon led to cogent activation of BCAA biosynthesis by condensing with 2-keto-isovalerate (2-KIV) to produce

2-isopropylmalate (2-IPM) (Fig. 6b), a metabolic intermediate that can act directly to induce the expression of many genes for BCAA biosynthesis in *S. cerevisiae* (Supplementary Fig. 6a)[53]. We found that the expression of 2-IPM-responsive genes, which are co-regulated under the control of the transcriptional factor Leu3, including *LEU4*, *LEU1*, *ILV2*, and *ILV5*[54], were constitutively upregulated in *sam5Δarg7Δ* cells (Fig. 6c and Supplementary Fig. 6b, c). Further supporting the idea of hyperactivation of leucine synthesis, acetyl-CoA abundance exhibited a greater reduction in the *sam5Δarg7Δ* mutant (Fig. 6d). This reduction was likely caused by consumption from leucine synthesis but not from the acetyl cycle, as acetyl-CoA levels were not affected by *arg7Δ* (Fig. 6d).

**The hierarchy of acetyl-CoA allocation in mitochondrial metabolism**
To validate the flow of acetyl-CoA into BCAAs and determine the hierarchy of acetyl-CoA in the mitochondrial pathways, we performed

the isotope tracing experiment using [U-$^{13}$C6] glucose in WT and *sam5Δarg7Δ* cells. The unlabeled acetyl-CoA level was quickly and fully replaced by the newly synthesized in both WT and the mutant 1 h after the tracing (Supplementary Fig. 6d), thereby allowing examination of the flux of acetyl-CoA. Despite endogenous acetyl-CoA levels being relatively low in *sam5Δarg7Δ*, labeled acetyl-CoA amounts after 1 h $^{13}$C tracing were slightly higher than WT (Fig. 6d and Supplementary Fig. 6d). In line with this, the steady and $^{13}$C-incorporated levels of TCA cycle metabolites were both greatly accumulated in *sam5Δarg7Δ* cells (Supplementary Fig. 6e, f). Such increases are likely caused by the arrested growth of this mutant in the arginine and leucine-free condition, under which the disposal of TCA cycle metabolites is inert.

We next examined the flux of acetyl-CoA into arginine biosynthesis using [U-$^{13}$C6] glucose (Fig. 6e). In WT cells, a large amount of $^{13}$C was readily incorporated into arginine, accompanied by increases in the amounts of fully labeled acetyl-glutamate, citrulline, and arginine (Fig. 6f). However, the fully labeled citrulline and arginine were undetectable in *sam5Δarg7Δ* cells (Fig. 6f), consolidating the severe defect in arginine biosynthesis. Notably, this mutant also exhibited peculiar increases in the m + 2 form of acetyl-glutamate and acetyl-ornithine (Fig. 6g), which originated mainly from m + 2 acetyl-CoA and unlabeled glutamate. Such disproportional production of the m + 2 metabolites implicated an insufficient production of glutamate in this mutant, which was further corroborated by the findings of the low labeled fraction of glutamate (Supplementary Fig. 6g) and the great accumulation of a $^{13}$C-labeled precursor, αKG (Supplementary Fig. 6f). Our results indicate that in the *sam5Δarg7Δ* mutant a small fraction of acetyl-CoA can enter the acetyl cycle but is not sufficient to promote the αKG-to-glutamate shunt for arginine synthesis.

In contrast to the glucose-derived $^{13}$C incorporation deficiency into glutamate and arginine, this heavy isotope was readily absorbed into BCAAs in the *sam5Δarg7Δ* mutant, as evidenced by marked increases in the amounts of $^{13}$C-labeled leucine, isoleucine, and valine (Fig. 6h, i). Among the $^{13}$C-labeled BCAAs of this mutant, the most abundant isotopologues were m + 6 leucine, m + 2 isoleucine, and m + 5 valine, all corresponding to the precursors of [U-$^{13}$C6] glucose-derived pyruvate or acetyl-CoA. The elevated ratio of m + 7 2-IPM to m + 5 2-KIV in the mutant was consistent with an active, acetyl-CoA-driven leucine biosynthesis (Fig. 6j). As a key signaling metabolite activating BCAA biosynthesis, 2-IPM levels increased through biosynthesis, with much more abundant m + 7 2-IPM in *sam5Δarg7Δ* cells (Fig. 6i). Therefore, acetyl-CoA deviated from the TCA cycle and arginine biosynthesis commits the production of 2-IPM, which inexorably promotes the synthesis of BCAAs, a process costing pyruvate and acetyl-CoA (Fig. 6h). Such aberrant activation of BCAA synthesis is expected to cause incomplete glucose oxidation and retard cell growth even in the presence of arginine (Fig. 5f). Consistent with this, exogenous leucine transcriptionally repressed BCAA biosynthesis (Fig. 6c and Supplementary Fig. 6c), leading to the restorative effect on the growth of *sam5Δarg7Δ* (Fig. 5f). To this end, our findings reveal a general hierarchy: the TCA cycle > the acetyl cycle for arginine synthesis > 2-IPM production for leucine synthesis in terms of the allocation of acetyl-CoA in mitochondrial metabolism.

## Sam5 translates methionine and leucine availability into cell growth control

Lastly, we investigated what amino acids could integrate with mitochondrial SAM availability for growth control. To do so, we surveyed all 20 amino acids individually by examining how they could affect the growth of WT and *sam5Δ* cells (Supplementary Fig. 7a). Only methionine and leucine exhibited Sam5-dependent effects on cell growth (Fig. 7a), indicating that the availability information of methionine and leucine could not be transduced into growth control in the absence of the mitochondrial SAM transporter. This is unsurprising because *sam5Δ* cells were desensitized to extramitochondrial SAM boosted by

exogenous methionine, thereby feigning a SAM depletion signal to allocate acetyl-CoA from the deficient TCA cycle. The resulting diversion of acetyl-CoA may lead to an uncontrolled activation of leucine biosynthesis, which renders the unresponsiveness of cell growth to leucine. Consistently, the growth defect of the *sam5Δ* mutant was not rescued by supplementation of leucine but ameliorated when leucine biosynthesis was genetically blocked (*leu1Δ, leu2Δ, leu4Δ,* and *leu9Δ*) or downregulated by Decreased Abundance by mRNA Perturbation (DAmP) of *LEU1* and *LEU2* (*LEU1-DAmP* and *LEU2-DAmP*) (Fig. 7b). Therefore, the redirection of glucose-derived fuel metabolites into the leucine biosynthesis pathway likely occurs at the cost of cell growth.

The allosteric inhibition of 2-IPM synthase and reduction of 2-IPM level by leucine[55,56] can result in transcriptional deactivation of the BCAA synthesis pathways, which in turn limits the flux of fuel metabolites into BCAAs. In light of this, cellular leucine may constitute a key component in the feedback circuit by which the mitochondrial content of acetyl-CoA distributes between the TCA cycle and anabolic reactions to accommodate cellular energy demand. We found that the turnover of TCA cycle metabolites was facilitated in the presence of leucine (Fig. 7c). It is possible that leucine promotes more acetyl-CoAs to enter the TCA cycle where glucose is oxidized. It has been reported that mitochondrial oxidation of glucose is highly controlled for sustaining the yeast metabolic cycle YMC[57,58]. We observed that leucine terminated the synchronized, periodic growth cycle in a Leu4-dependent manner (Fig. 7d), confirming that leucine-imposed modulation of glucose oxidation was dependent on this Leu4-mediated reaction. Therefore, the availability information of methionine and leucine can converge on mitochondria to tune glucose oxidation.

## Discussion

Unicellular organisms rely on cell-intrinsic mechanisms to sense nutrients in the environment and adjust the activities of metabolic pathways. While the effect of methionine on reprogramming transcriptional response has been well-studied[9,41,42], it remains enigmatic how cells sense methionine levels and relay sufficiency signals to downstream metabolic pathways. Here, we report that MR is sensed by yeast mitochondria as a sign of SAM deprivation; and as a response, the mitochondria redirect carbon metabolism to fuel nitrogenic anabolism in a lipoate-dependent mechanism. We propose a working model whereby mitochondria translate the nutrient information of methionine into energetic and anabolic balance for proliferation control (Fig. 7e, f). Specifically, mitochondria can surveil methionine availability in the form of mitochondrial SAM adequacy, allowing oxidation of glucose for energy production. In the scarcity of SAM, SAM insufficiency disrupts lipoate metabolism, directing glucose-derived fuel metabolites such as acetyl-CoA to the synthesis of amino acids, particularly arginine and leucine. We propose that the low hierarchy of acetyl-CoA allocation to leucine biosynthesis can wire a feedback circuit to break off this diversion when leucine becomes abundant.

In eukaryotes, there are many conserved nutrient-responsive signaling networks coordinating the use of metabolic resources, such as mTOR signaling that emanates from the lysosome[59]. However, the well-characterized nutrient sensors of mTOR, such as Sestrin for leucine, CASTOR for arginine, and SAMTOR for SAM, are not evolutionarily conserved in unicellular organisms[60]. Here, we characterized mitoSIR signaling in the budding yeast *S. cerevisiae*. This SAM-responsive programming of arginine and leucine metabolism takes place, intriguingly, in the mitochondria. This suggests that unicellular organisms position the mitochondria as a critical compartment to orchestrate bioenergetic and biosynthetic pathways. Unlike mammalian cells that are surrounded by nourishing nutrients, yeast cells are often exposed to extreme nutrient environments and are more sensitive to direct biosynthetic inputs such as carbon, nitrogen, and sulfur sources. Therefore, the surveillance of SAM by mitochondria at the site

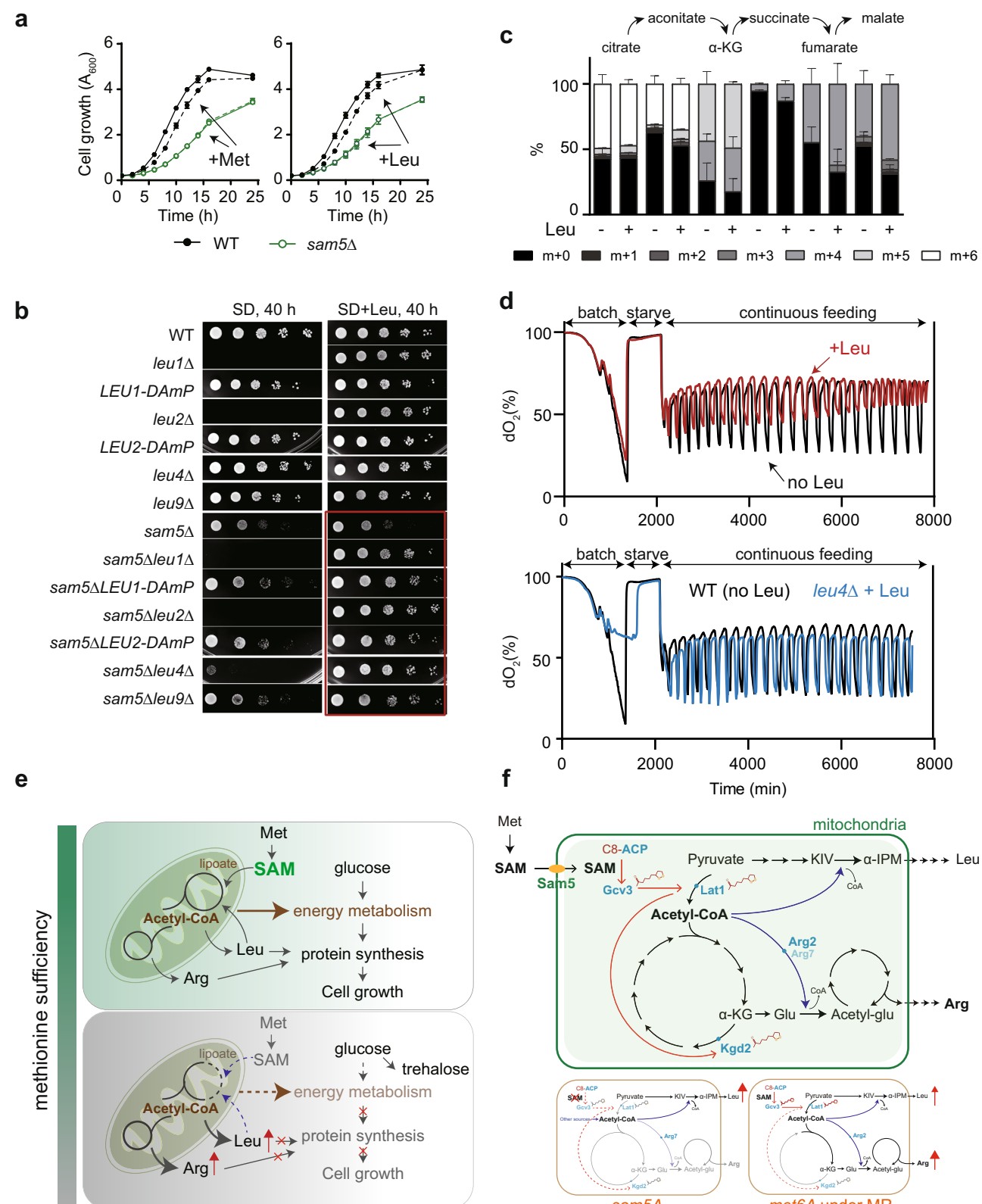

of bioenergetics and biosynthesis might have been effective for microorganisms in the metric of evolutionary fitness.

Is this mitoSIR-concerted programming of metabolism conserved in mammalian systems? This is not a trivial question to answer and awaits detailed investigation. First, the lipoate pathway remains less well-understood in mammals[47,61]. Besides the de novo lipoic acid synthesis pathway, which is similar to that in the budding yeast, a salvage pathway that makes use of exogenous lipoic acid might exist in mammalian systems[62]. Also, in mammalian cells, lipoylation can be removed by lipoamidase[63], which is not conserved in yeast[64]. These differences may complicate our understanding of how SAM might interfere with lipoylation in the mitochondria of higher eukaryotes. Second, some mammalian cells possess an alternative source for mitochondrial SAM in addition to uptake from the cytosol via the

**Fig. 7 | Methionine and leucine impose Sam5-dependent growth control.**
**a** Growth curves of WT and *sam5Δ* cells with or without 1 mM methionine or leu-
cine. Data are mean ± SD (*n* = 4, *n* = biologically independent samples) and repre-
sentative of at least three independent experiments. **b** Growth of indicated strains
on SD and SD containing 1 mM leucine. The decreased abundance by mRNA per-
turbation (DAmP) technique was used to reduce the expression of *LEU1* and *LEU2*.
**c** The effect of leucine on the percent abundance of $^{13}C$-labeled TCA cycle meta-
bolites. Data are represented as mean ± SD (*n* = 4, *n* = biologically independent
samples). **d** $O_2$ curves of WT and the *leu4Δ* mutant with or without supplementation

of 1 mM leucine. dO$_2$ refers to dissolved oxygen concentrations (% saturation) in the
media. **e, f** Models depicting how mitochondria convey SAM availability to balance
a tradeoff between energy production and anabolic metabolism. The *sam5Δ*
mutant with defective Lat1 and Kgd2 lipoylation decreases the production of acetyl-
CoA and arginine while increases leucine biosynthesis. MR constrains both Lat1 and
Kgd2 lipoylation in *met6Δ*, leading to the rewiring of the TCA cycle for the nitro-
genic syntheses of arginine and leucine. Source data are provided as a Source
Data file.

mitochondrial SAM transporter SAMC[65]. Recent studies show that the
SAM synthetase MATα1 is present in the mitochondria of
hepatocytes[66,67]. Last but not least, the metabolism of arginine and
leucine in mammals is different from that in yeasts. For example,
acetyl-glutamate serves as the first committed intermediate of arginine
biosynthesis in microbes but acts as an obligate allosteric activator of
carbamoyl phosphate synthetase in mammals[68], a mitochondrial
enzyme required for the biosynthesis of citrulline and arginine. It is
likely in mammals that MR may integrate mitochondrial SAM avail-
ability with lipoate metabolism for the regulation of acetyl-CoA-based
production of acetyl-glutamate. While this mitoSIR-concerted pro-
gramming of metabolism is instrumental to understanding metabolic
behaviors of mitochondria under MR, future research is warranted to
expand on the physiological importance and integrative components
of mitoSIR signaling in other organisms.

## Methods

### Yeast strains and media
All yeast strains used in this study are listed in Supplementary Data 1. All
strains and genetic manipulations were verified by sequencing and
phenotype. The prototrophic CEN.PK strain background was used in all
experiments. Gene deletions were carried out using either tetrad dis-
section or standard PCR-based strategies to amplify resistance cassettes
with appropriate flanking sequences, and replacing the target gene by
homologous recombination[69]. *LEU1-DAmP* and *LEU2-DAmP* were con-
structed to reduce *LEU1* and *LEU2* mRNA expression levels by disruption
of the 3' untranslated region with an antibiotic resistance cassette.
Carboxy-terminal tags were similarly made with the PCR-based method
to amplify resistance cassettes with flanking sequences. The primers for
constructing yeast strains were provided in Supplementary Data 2.

Media used in this study: Yeast extract peptone dextrose medium
YPD (2% yeast extract, 2% peptone, 2% glucose) and synthetic defined
medium SD (0.17% yeast nitrogen base without amino acids containing
0.5% ammonium sulfate (Difco), 2% glucose). Tracing media: SD-$^{15}$N
(0.17% yeast nitrogen base without amino acids containing 0.05%
$^{15}$N-ammonium sulfate, 2% glucose) and SD-$^{13}$C (0.17% yeast nitrogen
base without amino acids containing 0.5% ammonium sulfate, 0.2%
U-$^{13}$C6-glucose). For continuous culture in a bioreactor, we used the
yeast metabolic cycle (YMC) medium, a minimal medium consisting of
0.5% ammonium sulfate, 0.2% potassium dihydrogen phosphate,
0.05% magnesium sulfate heptahydrate, 0.008% calcium chloride,
0.0018% iron sulfate heptahydrate, 0.0009% zinc sulfate heptahy-
drate, 0.0005% copper sulfate, 0.00008% manganese chloride tetra-
hydrate, 0.1% yeast extract, 1% glucose, 0.035% v/v sulfuric acid, and
0.05% Antifoam 204.

### Determination of growth rates of yeast cells by absorbance at 600 nm or spotting assay
For determining cell growth rates in liquid medium, yeast cells cul-
tured overnight were diluted with fresh medium, and cell growth was
monitored by absorbance reading at 600 nm using a spectrometer
from MAPADA (Model P6). In Fig. 1b, *met6Δ* and *sam1Δsam2Δ* cells
were cultured in SD medium with 1 mM methionine or 1 mM SAM
overnight. The overnight cultures were diluted with the same medium
to 0.2 of A$_{600}$. When OD$_{600}$ reached 0.65, cells were collected, washed

with sterilized water, and re-suspended to fresh SD with or without
1 mM methionine or SAM. In Supplementary Fig. 2d, cells were cul-
tured overnight in SD and diluted to 0.2 of A$_{600}$ in fresh SD with or
without 1 mM arginine. In Fig. 5e, f, overnight cultured *sam5Δarg7Δ*
cells were diluted to 0.01 of A$_{600}$ in fresh SD or SD with 1 mM amino
acid(s) as indicated. In Fig. 7a and Supplementary Fig. 7a, the overnight
cultures of WT and *sam5Δ* cells were diluted to 0.2 of A$_{600}$ in fresh SD
with or without 1 mM indicated amino acid.

For spotting assay in Supplementary Figs. 2e and 4g, cells were
pre-cultured in YPD overnight to the stationary phase, and 3 µL ali-
quots of a series of 10-fold dilutions from 0.5 OD$_{600}$ were spotted on
YPD, SD, and YPGE plates and incubated at 30 °C until observation. In
Fig. 5d and Supplementary Fig. 5a, b, the overnight cultures of indi-
cated cells grown in SD media were used to make a series of 10-fold
dilutions for the spotting experiment. In Fig. 7b, cells were pre-
cultured in SD with 1 mM leucine overnight. 3 µL aliquots of a series of
10-fold dilutions from this overnight culture were spotted on SD and
SD with 1 mM leucine. The growth was recorded at indicated times.

### Fluorescence microscopy
Overnight cultures of indicated cells were diluted in fresh SD medium
and grew to the logarithmic phase for imaging. All images were taken
under a 100X, 1.4 NA oil-immersion objective lens with a Deltavision
Elite microscope (Applied Precision, GE).

### Metabolite extraction and quantitation
Intracellular metabolites were extracted using a previously established
method[58]. Care was taken to quench cells quickly and maintain meta-
bolites in acid to minimize oxidation. Briefly, equal OD units of cells
were rapidly quenched to stop metabolism by addition into 4 volumes
of quenching buffer containing 60% methanol and 10 mM Tricine, pH
7.4 that was pre-cooled to −40 °C. 5 min after holding at −40 °C, cells
were spun at 5000 g for 3 min at 0 °C, washed with the same buffer,
and then re-suspended in 1 mL extraction buffer containing 75%
ethanol and 0.5 mM Tricine, pH 7.4. Intracellular metabolites were
extracted by incubating at 75 °C for 3 min, followed by chilling on ice
for 5 min. Samples were spun at 20,000 g for 1 min to pellet cell debris,
and 0.9 mL of the supernatant was transferred to a new tube. After a
second spin at 20,000 g for 10 min, 0.8 mL of the supernatant was
transferred to a new tube. Metabolites in the extraction buffer were
dried using a vacuum concentrator system (Labconco) and stored at
−80 °C until analysis. Dried metabolite extracts were re-suspended in
60% acetonitrile for injection.

Cellular metabolites were quantitated by LC-MS/MS with a triple
quadrupole mass spectrometer (the QTRAP 6500+ System, ABSCIEX)
using previously established methods[19,70]. Briefly, metabolites were
separated chromatographically on a SeQuant Zic-pHILIC column (5 µm
polymer 150 × 2.1 mm, Millipore Sigma) using a high-performance
UHPLC system (Exion LC AD system) coupled to a triple quadrupole
mass spectrometer (QTRAP 6500+ System, AB SCIEX). For targeted
metabolomics, we performed a 34-min liquid chromatography on the
pHILIC column at a flow rate of 0.15 mL/min, with 20 mM ammonium
carbonate and 0.1% (v/v) ammonium hydroxide as Solvent A and
acetonitrile as Solvent B. The following gradient was employed:
0.01 min 80% B, 20 min 20% B, 20.5 min 80% B, 34 min 80% B.

Metabolites were detected by MRM transitions in positive or negative modes. The raw data were extracted with the software Analyst v1.7.2 and OS v1.7.

### Reverse stable isotope labeling (RIL) to estimate cellular concentrations of amino acids

WT yeast cells were inoculated in SD-$^{15}$N medium and cultured overnight. Cells were diluted to fresh SD-$^{15}$N medium and harvested when OD$_{600}$ reached 1.0. One OD unit of cells was used to extract metabolites. Specifically, cells were rapidly quenched to stop metabolism by addition into 4 mL of quenching buffer that was pre-cooled to −40 °C. 5 min after holding at −40 °C, cells were spun at 5000 g for 3 min at 0 °C, washed with the same buffer, and then re-suspended in 1 mL of the ethanol-based extraction buffer. Intracellular metabolites were extracted by incubating at 75 °C for 3 min, followed by chilling on ice for 5 min. Samples were spun twice to eliminate cell debris, and 0.85 mL of the supernatant was transferred to a new tube. Metabolites in the extraction buffer were dried using a vacuum concentrator system.

Dried metabolite extracts were re-suspended in 120 μL 60% acetonitrile. After several rounds of centrifugation, 90 μL of the supernatant was added into 10 μL of an amino acid mixture solution containing 10 μM of each of the 20 amino acids. 5 μL of the reconstituted sample was injected for MS analysis. Molar concentrations of $^{15}$N-incorporated amino acids in the sample were calculated based on the signal intensities of the added standards of $^{14}$N-amino acids. Cellular concentration for each amino acid in a single yeast cell was then converted based on the incurred dilution rate, the number of yeast cells in 1 OD$_{600}$ unit ($1.5 \times 10^7$), and the cell volume of a single yeast cell ($3.7 \times 10^{-14}$ L).

### Metabolic flux analysis using $^{15}$N ammonium sulfate

$^{15}$N labeled ammonium sulfate (($^{15}$NH$_4$)$_2$SO$_4$) was obtained from Cambridge Isotope Laboratories. $^{15}$N ammonium sulfate-based tracing experiments were described previously[21,23]. Cells were pre-cultured overnight to the stationary phase and diluted to the same fresh medium. When reaching the logarithmic phase, cells were spun down, washed with SD medium without ammonium sulfate, and re-suspended to the SD-$^{15}$N tracing medium where the sole nitrogen source ammonium sulfate was $^{15}$N labeled. Cells were collected at indicated times, and metabolites were extracted as described above. $^{15}$N metabolites were detected by LC-MS/MS, with the targeted parent and daughter ions specific to the $^{15}$N form of the metabolites.

*met6Δ* cells were cultured in a methionine-supplemented SD medium overnight, diluted to the same methionine medium, and grew to the logarithmic phase. The cells were then washed and re-suspended to the SD-$^{15}$N tracing medium without methionine. After 0.5 h, 1 h, 1.5 h, 2 h, and 4 h, the cells were collected for metabolite extraction. In Fig. 1i, the mass isotopologue distribution of amino acids in *met6Δ* cells was calculated based on the data obtained after 30 min MR and represented as the percent abundance of $^{15}$N-incorporated amino acids. In Fig. 1k, the abundance of each isotopic metabolite in the arginine pathway was represented as fold change that was the spectra intensity levels of metabolites at indicated times relative to those before MR.

In Fig. 2g, WT and *sam5Δ* cells were cultured and traced in SD-based medium. The abundances of metabolites were represented as the count areas of spectra intensity for labeled and unlabeled metabolites. In Fig. 5a, the mass isotopologue distribution of amino acids in WT and *sam5Δ* cells was represented as the percent abundance of $^{15}$N-incorporated amino acids 1 h after the $^{15}$N tracing condition.

### Metabolic flux analysis using $^{13}$C glucose

$^{13}$C-labeled glucose (U-$^{13}$C6, 99%) was obtained from Cambridge Isotope Laboratories. Metabolic flux analysis using $^{13}$C glucose was based on

published methods with modification[71,72]. In brief, cells were pre-cultured overnight to the stationary phase and diluted to the same fresh medium. When reaching the logarithmic phase, cells were spun down, washed with SD medium without glucose, and re-suspended to the SD-$^{13}$C tracing medium where the glucose carbons were all $^{13}$C labeled. Cells were collected at indicated times, and metabolites were extracted as described above. $^{13}$C metabolites were detected by LC-MS/MS, with the targeted parent and daughter ions specific to the $^{13}$C form of the metabolites.

For the tracing experiments in Fig. 3d–i and Supplementary 3h, i, WT and *met6Δ* cells cultured overnight in SD plus 1 mM methionine were diluted and grew to the logarithmic phase in the fresh methionine medium. The cells were then washed with a glucose-free SD medium and re-suspended in the SD-$^{13}$C tracing medium. Metabolite extraction was performed at indicated times. The mass isotopologue distribution of TCA cycle metabolites in WT and *met6Δ* cells after 10 min (Fig. 3e) and 60 min (Supplementary Fig. 3h) of MR was represented as the percent abundance of $^{13}$C-incorporated metabolites.

For the tracing experiments in Fig. 6e–j, WT and *sam5Δarg7Δ* cells were cultured in SD plus 1 mM arginine overnight. The overnight cultures were spun down, washed, and re-suspended in SD medium. WT cells were inoculated at 0.1 of A600, while *sam5Δarg7Δ* cells were inoculated at 0.5. After 6 h of culture at 30 °C, WT and the double deletion mutant cells were washed with the glucose-free SD medium and shifted to the SD-$^{13}$C tracing medium. Soluble metabolites were extracted after 1 h incubation under the $^{13}$C tracing condition. The abundances of metabolites in arginine biosynthesis, BCAA biosynthesis, and the TCA cycle were represented as accumulative count areas of the spectra intensity of each mass isotopologue.

For the tracing experiments in Fig. 7c, WT cells grown overnight in SD medium with or without 1 mM leucine were diluted to the corresponding fresh SD medium with or without 1 mM leucine. Cells growing logarithmically were washed and shifted to the SD-$^{13}$C tracing medium with or without leucine supplementation. One hour after in the $^{13}$C tracing condition, the cells were harvested for metabolite extraction as described above. The mass isotopologue distribution of TCA cycle metabolites in WT cells growing in the absence or presence of 1 mM leucine was represented as the percent abundance of $^{13}$C-incorporated metabolites.

### Tracing lipoate synthesis using $^{13}$C glucose

For the tracing experiment of lipoate synthesis, the detailed procedure was shown in Fig. 4m. *met6Δ* cells bearing Kgd2 C-terminally tagged with 3xflag were cultured in SD medium containing 1 mM methionine overnight. The cells were diluted and cultured with fresh methionine-containing SD medium to the logarithmic phase. The cells were collected, washed, and re-suspended to four different culture media for both $^{13}$C tracing and MR treatment. The cells of different groups were collected to extract protein after 2 h. Kgd2 enriched by immunoprecipitation was treated with trypsin for 4 h and lipoamidase from *Enterococcus faecalis* (efLPA) for 2 h at 37 °C to release the covalently attached lipoic acid[73]. Acetonitrile was added to extract lipoic acid. After centrifugation, the upper phase containing lipoic acid was saved to a new tube and dried using a vacuum concentrator system. The samples were reconstituted and injected for mass spectrometry analysis. Lipoic acid was detected by LC-MS/MS, with the targeted parent and daughter ions specific to the $^{12}$C or $^{13}$C form of the metabolite.

The assay of tracing lipoate with labeled glucose was rigorously validated. Kgd2 immunoprecipitation was highly efficient and the same in cells with or without MR (Supplementary Fig. 4m). Lipoamidase treatment alone can efficiently remove lipoate from the immunoprecipitated Kgd2 protein, however, with a slight degradation of the protein itself (Supplementary Fig. 4n). To further ensure the efficient release of lipoic acid, we performed trypsin treatment before lipoamidase treatment. This slightly increased lipoic acid release

(Supplementary Fig. o). Because lipomidase alone could free most lipoic acid from Kgd2, this increase indicates that the release of lipoic acid with trypsin and lipoamidase treatment was very efficient.

## Whole yeast cell extracts preparation and western blotting

A urea-based protocol was used to lyse yeast cells for western blots. Cells were spun down, quenched in 20% trichloroacetic acid on ice for 15 min, and washed with acetone. Cell pellets were re-suspended in urea buffer containing 50 mM Tris-Cl pH 7.5, 5 mM EDTA, 6 M urea, 1% SDS, 1 mM PMSF, 2 mM sodium orthovanadate, and 50 mM NaF, and lysed by bead-beating. After collecting supernatants, protein concentration was determined using Pierce BCA protein assay, and the same amounts of proteins were separated using SDS-PAGE gels. Proteins were transferred to a nitrocellulose membrane or a PVDF membrane and blotted with the corresponding antibodies. Blocking was performed in 5% dry milk/TBST, and antibody incubation was in 5% dry milk/TBST. For western blotting of protein lipoylation, cells were quenched directly in culture medium with 100% trichloroacetic acid and collected by centrifugation. In Fig. 4k, l, we avoided centrifugation during the switch step for removing methionine or SAM. Instead, a vacuum system was used to collect cells on filter papers. The cells were then washed with pre-warmed culture medium and re-suspended to minimal medium without methionine or SAM. Antibodies used in this study include mouse anti-FLAG M2 antibody (Sigma, Cat# F3165, 1:5000), rabbit anti-G6PDH (Sigma, Cat#A9521, 1:5000), rabbit anti-lipoic aicd (Milipore, Cat#437695, 1:3000). All antibodies are commercially available and validated. Uncropped and unprocessed scans of the blots are provided in the Source Data file. ImageJ (v1.8.0, National Institutes of Health) was used for Western blot densitometry analysis.

## RNA extraction and quantitative real-time PCR

Cells were collected for RNA extraction under different growth conditions, as depicted in Supplementary Fig. 6b. RNA extraction was carried out following the manufacturer's manual using MasterPure yeast RNA purification kit (Lucigen). RNA concentration was determined by $A_{260}$. 1 μg RNA was reverse transcribed to cDNA using a reverse transcriptase kit HiScript RT SuperMix with gDNA wiper from Vazyme. Real-time PCR was performed in triplicate with ChamQ SYBR qPCR Master Mix (Vazyme) using a Bio-Rad CFX96 Connect device. Transcript levels of genes were normalized to *ACT1*. All the primers used in RT-qPCR have efficiency close to 100%, and their sequences are listed below.

*LEU1*, forward 5′-GCCTTGTCTCGTACACATCTAA-3′;
*LEU1*, reverse 5′-GACCTCAACCTTTGGACTACTT-3′;
*LEU2*, forward 5′-CCCAACCCACCTAAATGGTATTA-3′,
*LEU2*, reverse 5′-TGGCAACAAACCCAAGGA-3′;
*LEU4*, forward 5′-ACGTACAGGTAATGTGGACTTG-3′;
*LEU4*, reverse 5′-CTCTTTGCGATACTGGGATCTT-3′;
*ILV2*, forward 5′-CCAACGACACAGGAAGACAT-3′;
*ILV2*, reverse 5′-CCGTAACCCATCGTACCTAAAC-3′;
*ILV5*, forward 5′-GTCGAAGAAGCTACCCAATCTC-3′;
*ILV5*, reverse 5′-CAGTCCAAAGCACCTCTTCT-3′;
*ACT1*, forward 5′-TCCGGTGATGGTGTTACTCA-3′;
*ACT1*, reverse 5′-GGCCAAATCGATTCTCAAAA-3′.

## Yeast metabolic cycle (YMC)

The bioreactor used in this study to perform YMC experiments is from INFORS (Model minifors 2, 3 L). The YMC was established as described previously[57]. The bioreactor was operated at an agitation rate of 1000 min⁻¹, an aeration rate of 1 L/min, a temperature of 30 °C, and a pH of 3.4, with a working volume of 1 L. Once the system was equilibrated, 10 OD$_{600}$ units of cells cultured overnight in YPD medium were inoculated. Once the batch culture reached maximal density, the culture was starved for a minimum of 12 h. The continuous culture was then initiated by the constant infusion of media containing 1% glucose at a dilution rate of 0.1/h. The batch culture medium and the infusion medium were the same. As indicated in Fig. 7d, leucine was present in both batch and continuous cultures.

## Statistical analysis

The normalized abundances of metabolites were log transformed, centered about the median, and clustered by Spearman rank correlation algorithm using Cluster 3, and heat maps were obtained by the software Treeview 1.2.0. MetaboAnalyst 5.0, a web-based analysis platform[74], was used for KEGG pathway analysis. The statistical significance in related figures was assessed using two-tailed Student's *t* test. *p* values and *n* in column plots from Student's *t* test were specified in corresponding figure legend.

## Reporting summary

Further information on research design is available in the Nature Portfolio Reporting Summary linked to this article.

## Data availability

All data supporting the findings of this work are available within the article, supplementary information, and source data. Targeted metabolic profiling raw data generated in this study have been deposited in Metabolomics Workbench under Project ID: PR001637 [https://doi.org/10.21228/M83M7P]. Source data are provided with this paper.

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

## Acknowledgements

We thank Huasong Lu, Ben Zhou, and Hanaa Hariri for critical reading of the manuscript and Jingyuan Xue for helping plot a model schematic. We thank technical support from the Core Facility of Life Sciences Institute, Zhejiang University. This work was supported by grants from the National Natural Science Foundation China (92057102 and 32270816), a research fund from Zhejiang Provincial Key Laboratory for Cancer Molecular Cell Biology, the startup fund from the Life Sciences Institute of Zhejiang University, and the 1000 Talents Program for Young Scholars to C.Y.

## Author contributions

L.J. and C.Y. initiated this research; Conceptualization: C.Y., L.J., W.F., and Y.Z.; Investigation: W.F., L.J., Y.Z., S.Y., H.Q., J.C., Q.L., and C.Y.; Supervision: C.Y. and Z.T.; Writing and Funding acquisition: C.Y.

## Competing interests

The authors declare no competing interests.
