## [Peer Review File · Nature Communications]

Methionine restriction constrains lipoylation and activates mitochondria for nitrogenic synthesis of amino acidsReviewer #1 (Remarks to the Author):

Fang et al present data exploring how budding yeast adapt to methionine restriction when made auxotrophic for this amino acid. By using yeast strains deficient in methionine (*met6*) or SAM (*sam*) synthesis they propose that cataplerosis disposes of TCA cycle intermediates through amino acid anabolism and arginine production. Reduced lipoylation of ketoacid dehydrogenases is postulated as the mechanism, as lipoic acid synthase (a radical SAM enzyme) activity is likely reduced. Downstream metabolic consequences for arginine, leucine and acetyl-CoA metabolism are explored, with changes in transcription observed.

The *met6/sam* deficient model is well characterised and the dependency of methionine auxotrophy and diversion of acetyl-CoA and alphaKG to amino acid metabolism is for the most part clear. However, the involvement of lipoylation as the sensor does not seem well supported by the experiments, and generally the manuscript may be better suited to a more specialised journal as it is of limited relevance to other organisms aside from budding yeast. The authors note that it is difficult to relate their findings to mammals, where there are clear differences in the sources of SAM and lipoate cycling. The overall interest in this model is therefore limited.

The involvement of lipoylation as the sensor is not well substantiated. The rescue with *mtSAM1* is minimal in log or stationary phase (Fig 4d), as is the growth (Ex Fig 4d). *Kgh2* and *lat1* deletion only altered arginine in the stationary phase, but no effect in log phase or in arginosuccinate. I do not think it is the same as *sam* deletion in Fig 1. *Lip3* deletion may have offered greater insight into how lipoate levels may differ on the different TCA cycle lipoylated complexes.

If altered sulfur is limiting for radical SAM enzymes, it is not limited to the lipoic acid synthase, but would alter biotin synthesis, as the authors note. This should be tested. Similarly, wouldn't alterations in polyamine synthesis, ergosterol etc alter growth and all be SAM dependent? Thus, it is not convincing that lipoate is really a sensor here.

Other points:

- Is there a growth defect in *CEN.PK met6 sam* deficient cells compared to the wildtype strain when supplemented with methionine? The control is not shown.
- Fig 1d is difficult to follow as it is virtually impossible to relate the key to the multiple curves shown.
- Fig 3a – The proposed cataplerosis mechanisms are not clear. What happens to fatty acid synthesis?
- Why does *lat1* not decrease acetyl-CoA further?
- Fig 4. Relative levels of lipoate are potentially interesting but difficult to measure flux, without labelling newly synthesised lipoate. *Lip3* deletion may be helpful here.

Reviewer #2 (Remarks to the Author):

Fang et al have submitted an interesting study on the metabolic impacts of Met- and/or SAM-deficiency in yeast. The study used strategic genetic models, steady state metabolomics, and stable isotope metabolic flux measurements. The most dramatic revelation of the response to Met- or SAM-deficiency is an increase in levels of amino acids including E, Q, G, D, N, V, and R, which are synthesized, in part, in the mitochondria. As well as the mitochondrial bias, there is a bias toward amino acids with N in their side-chains. Importantly, this is not a passive result of Met-deficiency-induced decreased translation rates, as labeling studies with [15N]-ammonium sulfate showed this involves augmented synthesis, not decreased utilization. This was further supported by labeling of intermediates in the arginine synthesis pathway, and by demonstration of a dramatic increase in assimilation of inorganic nitrogen into these amino acids. This response is mapped to a re-purposing of acetyl-CoA out of the TCA and into the synthesis of these amino acids. This, in turn, was driven by inactivation of lipoic acid (LA)-requiring TCA enzymes, suggesting that the decreased Met or SAM resulted in decreased mitochondrial (S-containing) LA, leading to this metabolic rewiring. Moreover, the response to Met deficiency was shown to be mediated by the

subsequent decrease in SAM that this causes, making SAM concentrations the ultimate determinant of LA levels and therefore the metabolic response. The involvement of SAM and LA is compelling, as LA biosynthesis is uniquely dependent on Met/SAM, in addition to FeS clusters (in which the S is more likely Cys-derived). The authors go on to investigate the interactions with Leu, the hierarchy of acetyl-CoA utilization in yeast mitochondria, and the impacts of Met + Leu on yeast growth control and metabolic cycling. Overall, the study is interesting, robust, and important. Some relatively minor concerns about the authors' interpretations need attention and the presentation needs to be improved.

Major.

1. The paper shows that the response to Met or SAM deficiency is dramatic increases in nitrogen assimilation into amino acids by the mitochondria but provides no explanation for how this might help the yeast survive Met or SAM deficiency. What benefit is it to the yeast to make these amino acids or assimilate so much inorganic nitrogen? Or is the critical factor the disruption of the TCA? Or are these both physiological non-helpful secondary consequences of LA deficiency? Due to this disconnection between the metabolic switch and the physiological function of this switch, these results are phenomenology and do not explain the biological relevance of the metabolic rewiring.

2. The paper describes the yeast as sensing a "sulfur" deficiency, or a "sulfur scarcity signal", through mitochondrial SAM levels. The study, however, does not appear to discriminate sulfur deficiency from Met/SAM deficiency. The study tested Met and SAM deficiency, not sulfur deficiency, so this conclusion appears to unnecessarily infer beyond what is shown. If cystine is abundant, does Met or SAM deficiency still induce the metabolic switch?

3. The results using the NLS-linked SAM synthase (Figs 2A, S2A) are confusing and likely need better justification and support. What impact does relocation of SAM synthase to the nucleus have on the Met cycle as a whole? Do the metabolites of the Met cycle equilibrate freely through nuclear pores? If not, then the entire cycle is likely disrupted, not just mitochondrial availability of SAM. If they do, then why would this impact mitochondrial availability of SAM?

4. I would appreciate similar explanations and support for the statements about Gdh1, as apparently a nuclear protein, having access to Glu and NADP+, but isolated from mitoSAM (line 232). The authors cite a 47-year old paper that isolated Gdh1 from yeast nuclei; however, to support this critical argument in the current submission, the authors should empirically show that Gdh1 is exclusively nuclear and none is mitochondria-associated.

Minor.

1. Unnecessary terms or non-standard jargon need to be used much more prudently or not at all. Examples:

- The title uses the authors' contrived term "MitoSIR", which not defined in the title or previously in the literature. This confused me at the onset and will not attract people to read the work.

- The abstract unnecessarily and confusingly contrives the term "chalcogen amino acid" to refer to methionine. The chalcogens are O, S, Se, Te, and Po. All amino acids contain O; none contain Te, Po, or (in yeast) Se. So, in this study the so-called "chalcogen amino acids" are either all amino acids (due to O) or the sulfur amino acids. Why not just call them the "sulfur amino acids", or refer to them by their precise name(s)?

- There is only one sulfur in SAM. Calling it the "trigonal sulfur of SAM" is unnecessary and confusing (line 51).

- What is meant by "massive ammonia" (line 137)? The study shows that large amounts of ammonia – detected with mass-increased nitrogen – were incorporated. If line 137 refers to the quantity assimilated, it is an amount irrespective of mass. If it refers to the isotope used, it is a mass irrespective of quantity.

2. Unnecessary anthropomorphic jargon used to describe metabolic responses diminish clarity. The

abstract discusses intracellular metabolic rewiring as involving nutrient "information" which "communicates" with mitochondria via a "metabolic language". This is not helpful. A more matter-of-fact presentation wherein concentrations, activities, and responses are presented for what they are, rather than interpreted as "information", "communication", or "languages", would better convey the facts of the study.

3. Line 220: I think the authors mean "therefore", not "wherefore".

4. Line 423. The metabolic source of the ¹³C needs to be stated here.

5. The DAmP alleles need to be explained and the experimental outcomes using these alleles need to be put in context of this approach and its potential caveats.

6. Line 501. The authors likely mean "positions", not "posits".

Reviewer #3 (Remarks to the Author):

The authors sought to dissect how mitochondria may respond to SAM depletion induced by methionine restriction in the budding yeast *S. cerevisiae*. They concluded that yeast mitochondria adapt to the signal of SAM deprivation by channeling metabolites derived from glucose oxidation such as acetyl-CoA to the synthesis of amino acids, mainly arginine and leucine in a lipoyl-dependent mechanism.

While the study is important for identification of cellular mechanisms that sense the environmental deprivation of methionine, further experiments are needed to validate the authors' conclusions and strengthen the paper.

In the manuscript, four models were used to study the SAM depletion in the mitochondria:

1. Yeast strain *met6Δ* with deletion of methionine synthase Met6, treated with methionine withdrawal – used in Fig 1, 3, and 4
2. Yeast strain *sam1Δsam2Δ* with deletion of SAM synthetases Sam1 and Sam2, treated with SAM withdrawal – used in Fig 1 and 4
3. Yeast strain *sam5Δ* with the deletion of the mitoSAM transporter gene SAM5 – used in Fig 2, 4, 5, 6, 7
4. Yeast strain *Sam1-NLS-GFP, sam2Δ* in which cytosolic SAM1 was sequestered into the nucleus using a nuclear localization signal (NLS) and SAM2 was deleted - used in Fig 2 and 4

Here are some questions that need to be addressed:

1. The depletion of SAM was only confirmed in *met6Δ* with methionine withdrawal and *sam1Δsam2Δ* with SAM withdrawal (Fig 1b). WT cells should be included in the experiment of Fig1b as a control.

Even though the total amount of cellular SAM was not changed in *sam5Δ* and *Sam1-NLS-GFP, sam2Δ* (Extended Data Fig. 2f), mitochondria SAM levels should be measured in these two mutants to confirm the depletion of mitoSAM.

2. From the data provided by the authors, methionine or SAM depletion led to increased arginine biosynthesis and amino acid accumulation in *met6Δ* and *sam1Δsam2Δ* (Fig1d-j). Adding SAM back blocked the amino acid accumulation in *sam1Δsam2Δ* (Extended Data Fig1c).

In contrast, *sam5Δ*, presumably with a shortage of mitoSAM, had significantly decreased arginine biosynthesis compared to WT (Fig 2b). This finding does not support the authors' conclusion that mitoSAM deprivation would increase arginine biosynthesis.

3. In Fig 2b and 2c, the levels of acetyl-CoA were significantly decreased in both *sam5Δ* and *Sam1-NLS-GFP, sam2Δ*, but arginine synthesis was significantly decreased in *sam5Δ*, and

significantly increased in Sam1-NLS-GFP, sam2Δ, suggesting that the change of arginine synthesis is independent of acetyl-CoA levels.

Again, in Fig 2, two mitoSAM-manipulating mutants had opposite response of arginine synthesis to presumed mitoSAM depletion, suggesting the arginine response might be independent of mitoSAM as well.

4. In Fig3 b-c, both met6Δgdh1Δ and met6Δidh1Δ showed significantly decreased arginine synthesis compared to met6Δ under MR, with acetyl-CoA not changed in met6Δgdh1Δ and increased in met6Δidh1Δ. This again suggests that acetyl-CoA production is irrelevant to arginine synthesis under MR.

5. In Fig 4i, depletion of methionine and SAM had no impact on the lipoylation of Lat1 and Kgd2 in met6Δ and sam1Δsam2Δ, suggesting increased arginine synthesis in the models of met6Δ - methionine and sam1Δsam2Δ-SAM is independent of lipoylation.

6. In Fig 6a, leucine was not changed in sam5Δ compared to WT, showing depletion of mitoSAM did not lead to leucine increase.

As shown above, several major conclusions regarding acetyl-CoA, lipoate, arginine and leucine synthesis in this manuscript are inconsistent with the displayed data. These discrepancies need to be clarified to make all the conclusions valid.

Minor points:

1. Please provide sequences for the PCR primers used for the construction of strains with enzyme deletions and tags.
2. Please provide the sources for the antibodies used in the Western blot.

Reviewer #4 (Remarks to the Author):

In this very interesting study the authors investigate metabolic changes due to sulfur scarcity (coming from methionine restriction), and its effect on metabolism. For this, the authors use two auxotrophs: met6Δ (which is a methionine auxotroph), and sam1Δsam2Δ (which specifically cannot make SAM). Under MR, these mutants appear to increase amino acid synthesis (nitrogen assimilation), specifically, arginine, glutamate, glutamine, aspartate and asparagine. The authors note these amino acids can produced via cataplerotic reactions from TCA cycle. In particular, arginine biosynthesis appears to be increased in MR. The authors find that depleting cytosolic SAM can induce this arginine biosynthesis in mitochondria, which depends on mitoSAM. They term this as mitoSIR (mitoSAM induced response). Upon mitoSAM depletion, lipoylation is affected, which in turn disrupts the TCA cycle leading to an overflow (?) of acetyl CoA, and induces an alternate metabolon for arginine biosynthesis, which is independent of Lipoic acid, but coupled to BCAAs. Therefore, the cells undergo a metabolic reprogramming under conditions of methionine restriction (MR). They finally suggest that methionine and leucine can regulate growth in a mitoSAM dependant manner.

This is an important study, with some strong observations. However, the manuscript suffers from some severe over interpretations of data, as well as over simplifications that can lead to misleading conclusions. A careful revision, with some key experiments can clarify many of these points, and make this a strong manuscript.

Major concerns and comments:

The main concerns I have with this manuscript are related to interpretations that come from

glossing over of contrasting observations, lack of data that unambiguously separates the role of mitochondrial SAM vs cytosolic SAM, and overinterpretations of Arg biosynthesis.

1) The premise of the paper is strong wherein the aim is to understand cellular responses to MR, with the idea of using 2 kinds of mutants (*met6* and *sam1,2*), since this can show that cellular SAM levels could signal these cellular responses. However, the data in many figures, specially figure 1, 2 and 4 contradict each other, and no clarifications or explanations are provided.

Specifically (points 2 and 3 are related):

2) Does mitoSAM actually increase during MR? And does this thereby lead to arg biosynthesis? Relatedly: 2. In Fig 2, the authors suggest that mitoSAM is important for arginine biosynthesis using *sam5Δ* and second, depleting cytosolic SAM can increase mitoSAM dependant arginine production. These conclusions come from data using *sam1-NLS-GFP* and *sam2Δ* (Fig2b and d). However, in both the mutants, the total level of cellular SAM remained same (extended data Fig 2f). No direct evidence for the change in mitoSAM levels leading to change in arginine levels is provided.

3) Relatedly, is this anabolic increase specific to SAM levels decreasing, and independent of methionine increase in this mutant?

4) Does lipoylation and subsequent arg synthesis depend on mitoSAM or cellular SAM decreases? According to fig 1, depleting cellular SAM pools activates arg biosynthesis. In other figures, the authors suggest that an increase in mitoSAM might activate arg biosynthesis.

For all of these it is unclear if and how much SAM accumulates in the mitochondria during methionine restriction, and then the authors need to explain how that is possible.

5) The authors also state ".....in contrast, the nuclear SAM mutant showed profound increases in cellular amounts of methionine...." (Extended data Fig 2d)
Authors observe an increase in arginine biosynthesis in the nuclear SAM mutant, but they do not see any difference in SAM levels between the nuclear mutant and *sam5Δ* (point 2). Separately they also find that methionine is high in the nuclear mutant. Given the now extensive data that methionine acts to signal an anabolic program (likely via cytosolic SAM increase) including synthesis of amino acids such as arginine and lysine, it isn't clear if the increase in arginine in the nuclear mutant (Fig 2d) is due to SAM depletion in cytosol, or due to increased methionine/SAM in the cell.

Adding to the previous points made, it becomes key to measure the levels of mitoSAM and cytosolic SAM in these mutants, and also assess if this anabolic increase is specific to reduced (nuclear?) SAM, or independent of methionine increase in this mutant. Alternately, they can also just recalibrate their interpretations accordingly.

6) In the section, "MitoSAM depletion alters arginine metabolism....." . Although the the claim is promising. The authors have over interpreted the data that is presented in the study.

7) From the figures 1 and 2 one interpretation is that the mito SAM pool/availability is what determines acetylation and arg biosynthesis, regardless of the nucleo/cytoplasmic pool. In the *sam5* mutant and the *sam1-NLS* mutant, there is no cytoplasmic depletion of SAM pool, yet opposite trends are observed for arg biosynthesis, purely based on mito SAM pool. While in the *sam1-NLS* mutant, SAM synthesis is relocated to the nucleus it that does not rule out the fact that cytoplasm is SAM depleted. An alternate experiment to test this would be to direct/increase SAM synthesis in the mito (target using a mito localization signal) and see responses.

8) Relatedly, can the authors explain how increased arg can benefit the growth of cells, when arg supplementation does not rescue growth in the MR mutants.

9) Questions related to how these intermediates are channelized towards arg biosynthesis: I think

for this story it would be enough to show how the carbon flux diverts to and away from arg synthesis in the context of SAM levels, which is already done in the flux experiments specially figure 6.

10) The authors state that ".....Because producing these amino acids needs metabolic inputs from mitochondria (Fig. 1h), such as cataplerotic reactions that dispose of TCA cycle intermediates...." In this case, the sulfur scarcity is created by using a mutant that is not capable of producing methionine or SAM. Both *met6* and *sam1sam2* are methionine cycle genes and their deletion will effect mitochondria anyway since the methionine cycle is coupled to the folate cycle as well as redox balance. If

mitochondria respond to sulfur scarcity as they claim, more direct evidence of the of TCA flux being diverted to arginine biosynthesis in a sulfur depleted state (WT cells grown in no sulfur media) is required.

11) Concern: 4. In Fig 3, the authors use *gdh1Δ* to reduce cataplerotically produced alpha-KG, to exclude this from TCA. In this regard, they have compared *met6Δgdh1Δ* to *met6Δ*. However, *Gdh1* is major enzyme that catalyzes conversion of alpha KG to glutamate. The deletion of *Gdh1* will also lower glutamate and other associated metabolites regardless MR. So, the comparison between these two mutants does not provide useful information, and is in fact actively misleading. Alternately, a comparison between *met6Δgdh1Δ* and *gdh1Δ* can help partly clarify if the decrease in upstream metabolites of arginine synthesis is significantly reduced in MR when compared to no MR condition (i.e., *Gdh1Δ*).

12) There is no real evidence that *Arg2* and *Arg5,6* in WT or *Arg2* and *Arg7* in *sam5Δ* physically interact/come together. If *Arg5,6* interact with *Arg2* in normally, whereas with *Arg7* in MR, some direct biochemical data should support this claim (of an alternate metabolon).

In general, while the concept, text and data for figure 5 are interesting, they are tangential (and require more substantial investigation to make a clearer story in itself), and only distract from the primary storyline of this study. I would strongly recommend that the authors think carefully about this section, and whether it should even be in this manuscript.

13) While Figure 3 suggests that glucose derived acetyl coA and 2-KG are channeled into arg synthesis upon MR, it isn't clear why the ratio of succinate to 2-KG is higher in the *met6* mutant in figure 3e and f. If the flux of acetyl CoA and thereby 2KG into arg synthesis is higher, how is this data explained?

Additional/other comments (some only require clarifications):

1. From fig 2, decreasing mitoSAM specifically (see *sam5* deletion) decreases arg biosynthesis. In addition, figure 4 suggests that mitoSAM and lipoylation of enzymes are necessary for arg biosynthesis. So how do the authors justify lines 274-276, wherein they say, during MR lipoic acid limitation results in arg biosynthesis by diverting carbon flux from the TCA cycle.

2. The title is misleading. The statement that methionine restriction activates lipoate dependent mitoSIR is an overstatement, since lipoylation itself depends on SAM, and SAM is reduced during MR. Unless the authors find that paradoxically SAM accumulates in the mitochondria during MR, in which case the title becomes more justified. I would suggest an appropriate modification of the title.

3. Please carefully write sections, without an overuse of adjectives, or overinterpretations. This is true for most of the text.

4. The fold change in levels of amino acids in the SAM mutant seems to be different in figure 1, panel e and extended figure 1, panel c. Why is this so? Could the authors include the heat map from the extended fig in the main one, so that it becomes clear that no matter what the mutant is, when SAM decreases, amino acids accumulate.

5. In fig1E, though amino acids accumulate in both mutants, it is more in the met6 mutant. Is there something else that is triggering increased amino acid accumulation than the SAM mutant? Do mitochondria sense SAM as a sulfur starvation signal? For instance, if you supplement a met6 deletion with exogenous SAM, do the levels of amino acids drop to normal?

6. The authors, claim that the "that the C-terminal tagging of ARG2 and ARG5,6..... which however becomes dispensable in the sam5Δ mutant." It is entirely possible that the growth defect due to C-terminal tagging can occur due to numerous reasons. Not clear at all.

7. The Blots needs quantification, especially Fig 4i.

8. The authors should additionally show the total levels of lipoic acid, and see whether it is affected under MR

9. The role of Sam5 in arginine biosynthesis (acetyl cycle) can be included in a schematic or text. Also, for figure 2 and associated text it isn't obvious how SAM levels can directly/indirectly determine acetylation of glutamate or acetyl CoA levels. So the authors can better clarify this.

10. In the text, the authors can cite prior data available for levels of amino acids under methionine or SAM deprivation in prototrophic strains (eg Sutter et al 2013), to especially contextualize why having the auxotrophs in this study is essential.

Additional note (to contextualize interpretations):

Extensive studies explain metabolic programs in cells when methionine is abundant (the exact opposite scenario of this study), in these same media conditions. Supplementing methionine induces amino acid biosynthesis, particularly arginine and lysine biosynthesis, along with changes in carbon metabolism (see PMIDs 33378328 and 30354837), likely in a SAM dependent manner. The authors are expected to be aware of this, so that they can contextualize their model in this context. Another interesting though peripheral paper (connecting to what methionine does to metabolism) is PMID 32821821

The model figure (final) is a bit euphemistic, and overly simplistic. It depicts that SAM depletion increases arg biosynthesis, but multiple figures show contrasting data in this regard. A more carefully constructed model to really reflect what is happening in the cell is warranted.

We would like to thank the reviewers for their valuable time and constructive comments. The point-to-point responses to the comments are as follows:

Reviewer #1 (Remarks to the Author):

Fang et al present data exploring how budding yeast adapt to methionine restriction when made auxotrophic for this amino acid. By using yeast strains deficient in methionine (met6) or SAM (sam) synthesis they propose that cataplerosis disposes of TCA cycle intermediates through amino acid anabolism and arginine production. Reduced lipoylation of ketoacid dehydrogenases is postulated as the mechanism, as lipoic acid synthase (a radical SAM enzyme) activity is likely reduced. Downstream metabolic consequences for arginine, leucine and acetyl-CoA metabolism are explored, with changes in transcription observed.

1. *“The met6/sam deficient model is well characterised and the dependency of methionine auxotrophy and diversion of acetyl-CoA and alpha-KG to amino acid metabolism is for the most part clear. However, the involvement of lipoylation as the sensor does not seem well supported by the experiments, and generally the manuscript may be better suited to a more specialised journal as it is of limited relevance to other organisms aside from budding yeast. The authors note that it is difficult to relate their findings to mammals, where there are clear differences in the sources of SAM and lipoate cycling. The overall interest in this model is therefore limited.”*

Response: We now provided additional evidence to support that methionine restriction (MR) can constrain protein lipoylation. We established an assay to trace lipoate synthesis using [U-¹³C₆] glucose (Fig. 4m), in which lipoic acid covalently attached to Kgd2 is released by treatment of trypsin and lipoamidase and measured with mass spectrometry. We found that the amount of lipoic acid released from the Kgd2 protein was decreased after 2 h MR treatment (Fig. 4m), with a greater decrease in the amount of the newly synthesized, ¹³C-fully labeled lipoic acid (Fig. 4n). These findings indicate that MR imposes a restriction on protein lipoylation, supporting lipoate metabolism being sensitive to MR.

Indeed, what is discovered in budding yeast does not straightforwardly mirror what can occur in mammals. However, this simple organism has been very instrumental in understanding complex biological processes conserved across the eukaryotic kingdom. The simplicity of this model system makes it possible to probe and gain valuable insights into biological processes that are challenging to study in a complex system, such as lipoate metabolism in mammals. Our findings of MR-elicited perturbation of lipoylation in the regulation of mitochondrial metabolism provide an entry point for understanding metabolic regulation and function for lipoylation. We realize the importance to compare with higher eukaryotes, which can provide a reference for savvy readers working with a different system. While doing so might seemingly have undermined the significance of our work, we respectfully argue that this mechanism is

still likely conserved in mammals, especially when the alternative source of SAM is not available (e.g. non-hepatocyte) or the lipoate cycling is not active, either of which, in fact, remains much understudied.

2. *“The involvement of lipoylation as the sensor is not well substantiated. The rescue with mtSAM1 is minimal in log or stationary phase (Fig 4d), as is the growth (Ex Fig 4d). kgd2 and lat1 deletion only altered arginine in the stationary phase, but no effect in log phase or in arginosuccinate. I do not think it is the same as sam deletion in Fig 1. Lip3 deletion may have offered greater insight into how lipoate levels may differ on the different TCA cycle lipoylated complexes.”*

- *“The involvement of lipoylation as the sensor is not well substantiated.”*

Response: As discussed above, we established an assay to trace lipoate synthesis. We found that the amount of lipoic acid released from the Kgd2 protein was decreased after 2 h MR treatment (Fig. 4n), with a greater decrease in the amount of the newly synthesized, ¹³C-fully labeled lipoic acid (Fig. 4l). These findings further indicate that lipoate metabolism is sensitive to MR.

- *“The rescue with mtSAM1 is minimal in log or stationary phase (Fig 4d), as is the growth (Ex Fig 4d).”*

Response: Mitochondrial protein lipoylation was not detectable in the *sam5Δ* mutant blocking the uptake of SAM to the mitochondria (Fig. 4e). Directing SAM synthesis into the mitochondrial matrix by expressing *mtSAM1* enabled lipoylation to occur in this mutant (Fig. 4f). We conclude that mitochondrial SAM is required for lipoylation. Admittedly, *mtSAM1* expression was not sufficient to fully restore lipoylation to WT levels (Fig. 4f). This is unsurprising because the ectopic synthesis of mitochondrial SAM by *mtSam1* is different from SAM uptake via the mitochondrial transporter *Sam5*. Because the latter process is likely coupled to extramitochondrial SAM availability, the observation of the partial rescue suggests that the temporal control of mitochondrial SAM, not only its availability, is critical for the process of lipoylation. We now added this to the manuscript.

The growth rescue with *mtSAM1* was strong. In a ten-fold serial dilution spotting assay, *sam5Δ* cells with *mtSAM1* grew one-spot faster on minimal medium and became able to grow on the non-fermentable medium (Supplementary Fig. 4g).

- *“kgd2 and lat1 deletion only altered arginine in the stationary phase, but no effect in log phase or in arginosuccinate. I do not think it is the same as sam deletion in Fig 1. Lip3 deletion may have offered greater insight into how lipoate levels may differ on the different TCA cycle lipoylated complexes.”*

Response: We think the reviewer meant *LAT1* and *GCV3* deletion here, since these two mutants, not *kgd2Δ*, decreased arginine in the stationary phase but not in the log phase. We also think that the reviewer meant the *sam5Δ* mutant in Fig. 2, not *sam1Δsam2Δ(sam)* in Fig. 1, because *sam* deletion resulted in arginine accumulation

under SAM limitation, and it is different from the observation of arginine deficiency discussed here. To reiterate, the reviewer commented that the *gcv3Δ* and *lat1Δ* mutants did not exhibit decreases in arginine levels in the log phase, different from the arginine reductions observed in the log phase cells of *sam5Δ*, thus the conclusion that lipoylation is required for arginine synthesis is not well-substantiated.

The steady level of a cellular metabolite is a balance between synthesis and consumption. The extent of change in a metabolite can be affected by the absolute abundance as well as the ratio of its accessible pool to inaccessible storage. Arginine is a very abundant metabolite, and its homeostasis is maintained via multiple mechanisms. The unchanged steady levels of arginine and arginosuccinate in the log phase cells (*gcv3Δ*, *lat1Δ*, *lip3Δ*, and *lip5Δ*) were not sufficient to rule out that the synthesis of arginine was not affected by lipoylation deficiency. We performed additional experiments to further support defective arginine synthesis in these mutants as discussed below.

First, the metabolic analyses of the abundances of arginine pathway intermediates revealed that acetyl-glutamate and acetyl-ornithine, the mitochondrial metabolites for the synthesis of arginine, were reduced greatly in the *lat1Δ*, *gcv3Δ*, *lip3Δ*, and *lip5Δ* mutants (Fig. 4b, Supplementary Fig. 4a-b), which was similar to those in the *sam5Δ* mutant (Fig. 2f). We also observed similar reductions in the levels of arginine precursor metabolites ornithine and citrulline in these mutants (Fig. 4b, Supplementary Fig. 4a-b). The metabolic down-regulation in arginine pathway metabolites suggests the requirement of lipoylation and the substrate proteins Lat1 and Gcv3 for arginine synthesis. We now made this conclusion in a more specific way that lipoylation was required for the mitochondrial steps for arginine synthesis.

Second, we performed additional tracing experiments to determine if the synthesis of arginine was indeed disrupted in the log phase cells of *lat1Δ*, *gcv3Δ*, *lip3Δ*, and *lip5Δ*. To do so, WT and mutant cells growing logarithmically were switched to growth medium containing ¹⁵N-ammonium sulfate as the only ammonium source. As shown in Fig. 4c, the turnover rates of unlabeled arginine (m+0) were decreased in *lat1Δ*, *gcv3Δ*, *lip3Δ*, and *lip5Δ* cells, indicating that the synthesis of arginine was indeed compromised by lipoylation deficiency.

It is curious why arginine deficits were evident in both log and stationary phases in the *sam5Δ* mutant but only pronounced in the stationary phase in *lat1Δ*, *gcv3Δ*, *lip3Δ*, and *lip5Δ* cells. We performed RT-PCR to profile gene expression in the arginine biosynthesis pathway. We found that all the mutants (*sam5Δ*, *lat1Δ*, *gcv3Δ*, *lip3Δ*, and *lip5Δ*) exhibited activated transcription of genes in the arginine biosynthesis pathway, with increases in mRNA levels of all these biosynthetic genes (Supplementary Fig. 4c). This transcriptional activation was in line with the characterized feedback response to arginine deficiency. Notably, compared to the *sam5Δ* mutant, the *lat1Δ*, *gcv3Δ*, *lip3Δ*, and *lip5Δ* mutants tended to have higher expression levels of these arginine

biosynthetic genes, which might lead to more potent restoration of the arginine deficits.

3. *“If altered sulfur is limiting for radical SAM enzymes, it is not limited to the lipoic acid synthase, but would alter biotin synthesis, as the authors note. This should be tested. Similarly, wouldn’t alterations in polyamine synthesis, ergosterol etc alter growth and all be SAM-dependent? Thus, it is not convincing that lipoate is really a sensor here.”*

Response: We very much appreciate this comment from the reviewer. We measured cell growth for the *sam5Δ* mutant in biotin-free medium as well as biotin levels in *met6Δ* and *sam1Δsam2Δ* cells with cellular SAM deprivation. The *sam5Δ* mutant that blocks mitochondrial SAM transport was auxotrophic for biotin (Supplementary Fig. 4h)¹. Also, biotin levels were dropped with the decreases in cellular SAM (Supplementary Fig. 4i). These findings thus suggest that decreases in cellular SAM can indeed limit radical SAM enzymes in the mitochondria. We also agree with the reviewer that many SAM-dependent processes can be sensitive to SAM under certain metabolic environments, for example, the methylation of histones² and the protein phosphatase 2A^{3,4}. As we are not claiming lipoate metabolism as the only process sensitive to changes in SAM level, whether or not other SAM-dependent processes may or may not respond to SAM limitation should not affect our conclusion with lipoate in this study.

We would like to further discuss what makes the radical SAM-dependent synthesis of lipoate unique for sensing and transducing the signal of cellular SAM deprivation. Considering both radical SAM enzymes for lipoate and biotin synthesis in the mitochondria, there are some key differences: First, lipoate metabolism in yeast and mammalian cells relies on the biosynthesis pathway, and the cells cannot utilize exogenous lipoate for protein lipoylation. Lipoic acid supplementation could not rescue the growth defect of the *sam5Δ* mutant (Supplementary Fig. 4h). Unlike lipoic acid, biotin taken up from the environment can be used undistinguishably from biosynthesis. For example, biotin present in yeast nitrogen base as an ingredient of culture medium is sufficient to support the growth of the *sam5Δ* mutant, otherwise a biotin auxotroph (Supplementary Fig. 4h). Second, the enzymes for lipoate metabolism and all three lipoylation substrate enzymes are exclusively in the mitochondria, and this is not the case for the synthesis and utilization of biotin. This mitochondria-localized lipoate metabolism may be featured as a compartment-specific SAM effector that is to coordinate mitochondrial metabolism with changes in cellular SAM levels.

Is lipoate metabolism qualified as a SAM sensor/effector. We would like to expand our discussion on this. Being a SAM sensor, a SAM enzyme-catalyzed process must be sensitive to cellular alterations in SAM content. The Michaelis constant (K_m) can provide a biochemical perspective on the affinity of SAM enzymes to the substrate SAM, assessing how a particular enzyme is sensitive to SAM levels. This becomes complicated in a biological context, as the K_m of any given enzyme can be affected by factors such as the metabolic environment or regulatory modifications on this enzyme. So, if a SAM-requiring process is sensitive to SAM, it is biological context-dependent. In this study, the conclusion that lipoylation is sensitive to SAM levels is corroborated

by several lines of evidence shown in Fig. 4. Specifically, 1) lipoylation requires mitochondrial SAM (Fig. 4e-f); 2) lipoylation of Kgd2/KDH is decreased in the *nSam1* mutant, in which SAM synthesis is relocated to the nucleus (Fig. 4i); 3) the increment of lipoylation over growth phase under a SAM-sufficient condition is abrogated in the mutants with SAM deprivation (Fig. 4k); 4) The halted increase in protein lipoylation in *met6Δ* cells can be resumed by simply providing methionine (Fig. 4l); and 5) the mass spectrometry-based tracing of lipoate synthesis indicates that MR constrained the synthesis of lipoate (Fig. 4m-o). Taken together, these findings indicate that protein lipoylation is sensitive to SAM levels.

Next, we propose that a SAM effector should timely transduce the information of SAM availability for functional output. In this study, we identified that the entry of acetyl-CoA into the TCA cycle and the oxidative decarboxylation of α KG both became disrupted under SAM deprivation (Fig. 3d-f), corresponding to the two complexes-pyruvate dehydrogenase (PDH) and α KG dehydrogenase (KDH)-that require lipoate as a covalently-bound coenzyme for catalysis. We confirmed that PDH/Lat1 lipoylation was required for the mitochondrial steps for arginine synthesis (Fig. 4b-c and Supplementary Fig. 4a-b) and that KDH/Kgd2 lipoylation is more sensitized under SAM alterations (Fig. 4i,4m-o). These findings lead us to propose that the constraint of lipoylation acts as a signal in the mitochondrion to rewire the TCA cycle; as a result, glucose oxidation is attuned from energy production to synthesizing amino acids such as arginine. This MR-induced arginine synthesis also benefits cell survival (Fig. 1l).

Other points:

4. *“Is there a growth defect in CEN.PK met6 sam deficient cells compared to the wildtype strain when supplemented with methionine? The control is not shown.”*

Response: No, there is no growth defect in the *met6Δ* and *sam1Δsam2Δ* mutants compared to wild type (WT) cells when methionine or SAM was supplemented (Supplementary Fig. 1a).

5. *“Fig. 1d is difficult to follow as it is virtually impossible to relate the key to the multiple curves shown.”*

Response: We highlighted Arg biosynthesis and Ala, Asp and Glu metabolism in Fig. 1d in color. To further improve clarity, we now included Supplemental Table 1 showing the actual p values from the KEGG pathway analysis. Arg biosynthesis and Ala, Asp and Glu metabolism are the most significant pathways among others under the conditions of the removal of methionine and S-adenosylmethionine.

6. *“Fig 3a – The proposed cataplerosis mechanisms are not clear. What happens to fatty acid synthesis?”*

Response: In Fig. 3a, we propose that the TCA cycle was disrupted under MR, causing the exit of acetyl-CoA and α -ketoglutarate to support arginine biosynthesis. This proposed mechanism was based on the findings in Figs. 1 and 2 and further confirmed in Fig. 3. First, the metabolic profiling results shown in Fig. 1 revealed the

increased synthesis of arginine under MR. Second, in Fig. 2, the genetic manipulation of SAM supply from the mitochondria suggested that acetylation of glutamate, the fate-committing step for the synthesis of arginine, was activated under MR. The generated acetyl-glutamate will irreversibly enter the acetyl cycle and exit as ornithine. Ornithine then leaves the mitochondria for the cytosolic steps of arginine synthesis. These findings together lead us to propose a cataplerosis mechanism whereby the TCA cycle is rewired to support arginine synthesis under MR.

Fatty acid synthesis takes place mostly in the cytosol^{5,6}. This pathway, also known as the type I pathway, is responsible for the bulk synthesis of all major fatty acids. It unlikely competes directly with the acetylation of glutamate for the mitochondrial acetyl-CoA pool. There is an independent fatty acid synthesis occurring in the mitochondria, known as the type II pathway⁵. The role of this pathway is not entirely clear, with a proposed role in synthesizing fatty acid up to C8 carbons, which can be used for the synthesis of lipoic acid⁶. The type II pathway for fatty acid synthesis might have some unknown metabolic connection to glutamate acetylation. We now depicted this process in the proposed model Fig. 3a. We respectfully suggest that how the two fatty acid biosynthetic processes may be affected under MR is beyond the scope of this study.

7. “Why does *lat1* not decrease acetyl-CoA further?”

Response: Lat1 is a major component of pyruvate dehydrogenase complex (PDH) that catalyzes oxidative decarboxylation of pyruvate to acetyl-CoA. It is unexpected that the *lat1Δ* mutant did not exhibit a greater decrease in acetyl-CoA in the log phase. We performed additional tracing experiments with [U-¹³C6] glucose to examine acetyl-CoA production. As shown in Fig. 4d, the turnover of unlabeled acetyl-CoA was decreased by *lat1Δ* and *gcv3Δ* (*GCV3* deletion can block Lat1 lipoylation, Fig. 4i-j), confirming that the production of acetyl-CoA was defective in *lat1Δ* cells. A possible explanation is that the metabolic pathways consuming acetyl-CoA in the *lat1Δ* cells during the log phase might be decreased as well, resulting in a marginal change in its steady level.

8. “Fig 4. Relative levels of lipoate are potentially interesting but difficult to measure flux, without labelling newly synthesised lipoate. Lip3 deletion may be helpful here.”

Response: Hypothetically, deletion of the *LIP3* gene encoding lipoate-protein ligase should block the transfer of lipoate and lead to an increased lipoylation of Gcv3. Then we may examine changes in the lipoylation level of Gcv3 in the *lip3Δ* mutant for assessing the effect of MR on lipoate synthesis. We followed this advice from the reviewer and performed experiments with the mutant. To our surprise, Gcv3 lipoylation was also not detectable in the *lip3Δ* mutant in the log phase like other mutants such as *sam5Δ*, *lip2Δ*, and *lip5Δ*, and slightly increased in the stationary phase (Figure A). It is unclear why *LIP3* deletion abolished Gcv3 lipoylation, but this precludes the use of this mutant for the proposed purpose.

Figure A: Protein lipoylation assayed with western blotting.

We established a mass spectrometry-based assay to trace the synthesis of lipoate using [U-¹³C6] glucose, as described above and detailed in our revised manuscript. We found that the amount of lipoic acid released from the Kgd2 protein was decreased after 2 h MR treatment (Fig. 4n), with a greater decrease in the amount of the newly synthesized, ¹³C-fully labeled lipoic acid (Fig. 4o). These findings provide additional evidence to support the constraint of lipoate metabolism imposed by MR.

Reviewer #2 (Remarks to the Author):

Fang et al have submitted an interesting study on the metabolic impacts of Met- and/or SAM-deficiency in yeast. The study used strategic genetic models, steady state metabolomics, and stable isotope metabolic flux measurements. The most dramatic revelation of the response to Met- or SAM-deficiency is an increase in levels of amino acids including E, Q, G, D, N, V, and R, which are synthesized, in part, in the mitochondria. As well as the mitochondrial bias, there is a bias toward amino acids with N in their side-chains. Importantly, this is not a passive result of Met-deficiency-induced decreased translation rates, as labeling studies with [15N]-ammonium sulfate showed this involves augmented synthesis, not decreased utilization. This was further supported by labeling of intermediates in the arginine synthesis pathway, and by demonstration of a dramatic increase in assimilation of inorganic nitrogen into these amino acids. This response is mapped to a re-purposing of acetyl-CoA out of the TCA and into the synthesis of these amino acids. This, in turn, was driven by inactivation of lipoic acid (LA)-requiring TCA enzymes, suggesting that the decreased Met or SAM resulted in decreased mitochondrial (S-containing) LA, leading to this metabolic rewiring. Moreover, the response to Met deficiency was shown to be mediated by the subsequent decrease in SAM that this causes, making SAM concentrations the ultimate determinant of LA levels and therefore the metabolic response. The involvement of SAM and LA is compelling, as LA biosynthesis is uniquely dependent on Met/SAM, in addition to FeS clusters (in which the S is more likely Cys-derived). The authors go on to investigate the interactions with Leu, the hierarchy of acetyl-CoA utilization in yeast mitochondria, and the impacts of Met + Leu on yeast growth control and metabolic cycling. Overall, the study is interesting, robust, and important. Some relatively minor concerns about the authors' interpretations need attention and the presentation needs to be improved.

Major.

1. *“The paper shows that the response to Met or SAM deficiency is dramatic increases in nitrogen assimilation into amino acids by the mitochondria but provides no explanation for how this might help the yeast survive Met or SAM deficiency. What benefit is it to the yeast to make these amino acids or assimilate so much inorganic nitrogen? Or is the critical factor the disruption of the TCA? Or are these both physiological non-helpful secondary consequences of LA deficiency? Due to this disconnection between the metabolic switch and the physiological function of this switch, these results are phenomenology and do not explain the biological relevance of the metabolic rewiring.”*

Response: We thank the comment from this reviewer. To investigate the physiological benefits of this MR-induced effect, we performed additional experiments to examine yeast survival under MR. We found that MR could evidently promote the survival of *met6Δ* (Fig. 1l and Supplementary Fig. 1k). This survival advantage in the *met6Δ* mutant was associated with a potent activation of autophagy (Supplementary Fig. 1l). While disruption of arginine synthesis by *arg7Δ* reduced the MR-activated autophagy in the *met6Δ* mutant, it also decreased the survival rate of this mutant (Fig. 1l and Supplementary Fig. 1l), indicating that MR-induced arginine synthesis was pro-survival.

2. *“The paper describes the yeast as sensing a “sulfur” deficiency, or a “sulfur scarcity signal”, through mitochondrial SAM levels. The study, however, does not appear to discriminate sulfur deficiency from Met/SAM deficiency. The study tested Met and SAM deficiency, not sulfur deficiency, so this conclusion appears to unnecessarily infer beyond what is shown. If cystine is abundant, does Met or SAM deficiency still induce the metabolic switch?”*

Response: We thank the reviewer for this comment. In the previous manuscript, we conclude that methionine/SAM deficiency is a type of sulfur scarcity that signals the mitochondria to respond. To avoid confusion and overinterpretation, we now carefully rephrased this and more accurately described it as methionine/SAM deficiency.

Whether cysteine/cystine can induce a similar metabolic response is a very intriguing question. We investigated this by adding cysteine to *met6Δ* cells after the switch. Unlike exogenous methionine and SAM, cysteine did not cause any reductions in arginine metabolites, with slight increases in some of the intermediary metabolites such as acetyl-glutamate, acetyl-ornithine, and ornithine (Supplementary Fig. 1i). This finding suggests that MR-induced response for arginine biosynthesis was unlikely due to cysteine deficiency.

3. *“The results using the NLS-linked SAM synthase (Figs 2A, S2A) are confusing and likely need better justification and support. What impact does relocation of SAM synthase to the nucleus have on the Met cycle as a whole? Do the metabolites of the Met cycle equilibrate freely through nuclear pores? If not, then the entire cycle is likely disrupted, not just mitochondrial availability of SAM. If they do, then why would this impact mitochondrial availability of SAM?”*

Response: The metabolites of the methionine cycle can in principle equilibrate freely through the nuclear pore. The relocation of SAM synthesis exclusively to the nucleus disrupts the spatiotemporal regulation of SAM, probably leading to a temporally delayed SAM supply for the mitochondria. This deregulation of SAM synthesis also likely entails secondary perturbations in many methionine/SAM-related pathways. For example, a great increase in SAH levels was observed in this mutant (Supplementary Fig. 2b). Thus, it is possible that the methionine cycle is also affected in this mutant. However, based on the findings from other genetic mutants, such as *met6Δ*, *sam1Δsam2Δ*, and *sam2Δ*, the upregulation of arginine biosynthesis is commonly associated with decreases in SAM (Figs. 1e-f and 2b). In addition, even in the *met6Δ* mutant, exogenous SAM was sufficient to reduce the mitochondrial response of amino acid accumulation under MR (Supplementary Fig. 1d). Together, we conclude that cellular SAM decrease is sufficient to activate the mitochondrial response. However, the relocation of SAM synthesis into the nucleus did not affect the total quantity of cellular SAM (Supplementary Fig. 2b), we infer that the spatiotemporal SAM supply is critical and, if disrupted, can signal the mitochondria to respond.

4. *“I would appreciate similar explanations and support for the statements about Gdh1, as apparently a nuclear protein, having access to Glu and NADP+, but isolated from mitoSAM (line 232). The authors cite a 47-year old paper that isolated Gdh1 from yeast nuclei; however, to support this this critical argument in the current submission, the authors should empirically show that Gdh1 is exclusively nuclear and none is mitochondria-associated.”*

Response: We now examined Gdh1 localization under a fluorescence microscope using the GFP-tagged Gdh1 strains with either the nuclear pore complex protein Nic96 or the mitochondrial matrix protein Idh1 mCherry-tagged. As shown in Supplementary Fig. 3e-f, in both WT and *met6Δ* cells, Gdh1-GFP was located in the cytosol and the nucleus, but not in the mitochondria. Also, a more recent study of the high-confidence mitochondrial proteome demonstrates that Gdh1 is not mitochondrial⁷. We now adjusted our statement in the manuscript.

Minor.

5. *“Unnecessary terms or non-standard jargon need to be used much more prudently or not at all. Examples:*

- *The title uses the authors’ contrived term “MitoSIR”, which not defined in the title or previously in the literature. This confused me at the onset and will not attract people to read the work.”*

Response: We agree with the reviewer and now changed our title to “Methionine restriction constrains lipoylation and activates mitochondria for nitrogenic synthesis of amino acids”.

- *“The abstract unnecessarily and confusingly contrives the term “chalcogen amino acid” to refer to methionine. The chalcogens are O, S, Se, Te, and Po. All amino acids contain O; none contain Te, Po, or (in yeast) Se. So, in this*

study the so-called “chalcogen amino acids” are either all amino acids (due to O) or the sulfur amino acids. Why not just call them the “sulfur amino acids”, or refer to them by their precise name(s)?”

Response: We agree with the reviewer and have changed this.

- *“There is only one sulfur in SAM. Calling it the “trigonal sulfur of SAM” is unnecessary and confusing (line 51).”*

Response: We thank the reviewer for this point. We now changed this.

- *“What is meant by “massive ammonia” (line 137)? The study shows that large amounts of ammonia – detected with mass-increased nitrogen – were incorporated. If line 137 refers to the quantity assimilated, it is an amount irrespective of mass. If it refers to the isotope used, it is a mass irrespective of quantity.”*

Response: We thank the reviewer for this comment. We now changed to “large amounts of ammonia” to improve clarity.

6. *“Unnecessary anthropomorphic jargon used to describe metabolic responses diminish clarity. The abstract discusses intracellular metabolic rewiring as involving nutrient “information” which “communicates” with mitochondria via a “metabolic language”. This is not helpful. A more matter-of-fact presentation wherein concentrations, activities, and responses are presented for what they are, rather than interpreted as “information”, “communication”, or “languages”, would better convey the facts of the study.”*

Response: We have now re-written the abstract and avoided unnecessary anthropomorphic jargons. We sincerely thank the reviewer for helping to improve the clarity of this manuscript.

7. *“Line 220: I think the authors mean “therefore”, not “wherefore”.”*

Response: We now corrected this.

8. *“Line 423. The metabolic source of the ^{13}C needs to be stated here.”*

Response: The ^{13}C is from [U- ^{13}C 6] glucose. We now added this.

9. *“The DAmP alleles need to be explained and the experimental outcomes using these alleles need to be put in context of this approach and its potential caveats.”*

Response: DAmP stands for Decreased Abundance by mRNA Perturbation. We explained this in the main text and added how to construct the DAmP strains in the method section. Because the disruption of various genes in the leucine biosynthesis pathway could all ameliorate the growth of *sam5Δ*, this rescue effect was unlike due to an unintended effect from genetic manipulation specific to DAmP.

10. *“Line 501. The authors likely mean “positions”, not “posits”.”*

Response: We have changed this.

Reviewer #3 (Remarks to the Author):

*The authors sought to dissect how mitochondria may respond to SAM depletion induced by methionine restriction in the budding yeast *S. cerevisiae*. They concluded that yeast mitochondria adapt to the signal of SAM deprivation by channeling metabolites derived from glucose oxidation such as acetyl-CoA to the synthesis of amino acids, mainly arginine and leucine in a lipoate-dependent mechanism.*

While the study is important for identification of cellular mechanisms that sense the environmental deprivation of methionine, further experiments are needed to validate the authors' conclusions and strengthen the paper.

In the manuscript, four models were used to study the SAM depletion in the mitochondria:

- 1. Yeast strain *met6*Δ with deletion of methionine synthase *Met6*, treated with methionine withdrawal – used in Fig 1, 3, and 4*
- 2. Yeast strain *sam1*Δ*sam2*Δ with deletion of SAM synthetases *Sam1* and *Sam2*, treated with SAM withdrawal – used in Fig 1 and 4*
- 3. Yeast strain *sam5*Δ with the deletion of the mitoSAM transporter gene *SAM5* – used in Fig 2, 4, 5, 6, 7*
- 4. Yeast strain *Sam1-NLS-GFP*, *sam2*Δ in which cytosolic *SAM1* was sequestered into the nucleus using a nuclear localization signal (NLS) and *SAM2* was deleted - used in Fig 2 and 4*

Here are some questions that need to be addressed:

*1. “The depletion of SAM was only confirmed in *met6*Δ with methionine withdrawal and *sam1*Δ*sam2*Δ with SAM withdrawal (Fig 1b). WT cells should be included in the experiment of Fig1b as a control.”*

Response: We now included WT as a control in the experiment of Fig. 1b (Supplementary Fig. 1a). The growth of WT and *met6*Δ cells was similar in the presence of methionine, while the growth of WT cells was slightly inhibited by SAM.

We further performed to examine the amounts of methionine, SAM, and other metabolites in WT and *met6*Δ cells under MR. As shown in Supplementary Fig. 1e-f, methionine and SAM levels were quickly dropped upon the withdrawal of methionine in both WT and *met6*Δ cells. Unlike a continuous decrease in SAM in the *met6*Δ mutant, SAM decreases in WT cells were slowed down after 2 h of MR, followed by a partial restoration. As a result, cellular SAM content was less depleted in WT than that in the *met6*Δ mutant. This indicates that the *Met6*-mediated methionine cycle contributes to SAM replenishment under MR.

*“Even though the total amount of cellular SAM was not changed in *sam5*Δ and *Sam1*-*

NLS-GFP, sam2Δ (Supplementary Fig. 2f), mitochondria SAM levels should be measured in these two mutants to confirm the depletion of mitoSAM.”

Response: We thank the review for this suggestion. It is technically challenging to accurately measure mitochondrial SAM levels. We have attempted to use different methods to isolate mitochondria for metabolic analysis, but thus far none of these methods passed our quality control that was set with mutants to alter mitochondrial metabolites in principle.

Metabolite analysis requires an immediate quenching process to stop metabolism, which is often done by adding methanol solution pre-cooled to -40°C directly to culture medium. However, procedures for the isolation of mitochondria and other organelles are not compatible with this quenching step. If the metabolism is not stopped during mitochondrial extraction, cellular and mitochondrial metabolism will be greatly affected by the extraction steps such as spheroplasting with zymolase, fractionation with high-speed centrifugation, etc. Additional technical challenges for measuring mitochondrial metabolites include the purity and integrity of isolated mitochondria. Extra procedures and buffer systems for isolating high-quality, purer mitochondria will further alter metabolism. In addition to these challenges, our research studies the metabolism of mitochondria under MR, and the extraction process itself creates a non-physiological condition that may mimic methionine/SAM depletion. From our trial experiments, we also noticed another concern that the isolation procedures might cause SAM to transport/leak through mitochondrial membranes possibly via permeases with broad specificities.

In the studies identifying the mitochondrial transporter Sam5/SAMC, reconstituted proteoliposomes were used to examine the activity and specificity of SAM transport^{8,9}. Mitochondrial SAM depletion was not tested in their transporter mutants. Because mitochondrial SAM is required for respiratory growth and biotin synthesis, the *sam5Δ* mutant exhibits growth defects on non-fermentable and biotin-free medium. The rescue of these growth defects by the ectopic expression of SAM synthetase Sam1 in the mitochondria provides genetic evidence of mitochondrial SAM insufficiency in *sam5Δ* cells. In our study, we used the same genetic approach to confirm the depletion of mitochondrial SAM in the *sam5Δ* mutant (Supplementary Fig. 4g-h). In addition, we showed that another mitoSAM-requiring process in the mitochondria, lipoate biosynthesis, was abolished in *sam5Δ* and rescued by mitochondrial *SAM1* expression (Fig. 4e-f). Thus, our genetic results support mitochondrial SAM depletion in the *sam5Δ* mutant.

In the *nSam1 (Sam1-NLS-GFP, sam2Δ)* mutant, we found that Kgd2 lipoylation was decreased (Fig 4i and Supplementary Fig. 4j), suggesting the SAM-imposed constraint of lipoylation. However, we do not think that this mutant had such depleted/low mitochondrial SAM as the *sam5Δ* mutant, because of the contrasting lipoylation of the Lat1 protein. Most likely, the relocation of SAM synthetase into the nucleus disrupts the temporal supply of SAM for the mitochondria, leading to a decrease in the

lipoylation of Kgd2. The fact that Kgd2 lipoylation is more sensitized under SAM restriction is probably because this protein is the last recipient of lipoate. We further performed tracing experiments and confirmed the installing of the newly synthesized lipoate to Kgd2 was decreased under MR.

2. *“From the data provided by the authors, methionine or SAM depletion led to increased arginine biosynthesis and amino acid accumulation in *met6Δ* and *sam1Δsam2Δ* (Fig1d-j). Adding SAM back blocked the amino acid accumulation in *sam1Δsam2Δ* (Supplementary Fig1c). In contrast, *sam5Δ*, presumably with a shortage of mitoSAM, had significantly decreased arginine biosynthesis compared to WT (Fig 2b). This finding does not support the authors’ conclusion that mitoSAM deprivation would increase arginine biosynthesis.”*

Response: We apologize for this confusion. Mitochondrial SAM levels are likely very low in the *sam5Δ* mutant and insufficient to support the mitochondrial SAM-requiring processes, leading to phenotypes including a respiratory growth defect (Supplementary Fig. 4g), biotin auxotrophy (Supplementary Fig. 4h), and the complete absence of protein lipoylation (Fig. 4e). Unlike the constant mitochondrial SAM transport deficiency in the *sam5Δ* mutant, mitochondrial SAM in the *met6Δ* and *sam1Δsam2Δ* mutants was passively restricted by cytosolic SAM after the withdrawal of methionine or SAM from culture medium. Therefore, mitochondrial SAM in these mutants experiences a transition of decrease to which extent that limits lipoate metabolism. Because we do not know precisely how mitochondrial SAM levels were altered due to technical limitations, we now carefully stated our conclusions without the speculation of changes in mitochondrial SAM.

It seems confusing that the opposite activities of arginine biosynthesis were observed in *sam5Δ* and *met6Δ*, *sam1Δsam2Δ* cells. This discrepant effect on arginine synthesis can be explained by how the lipoylation of different proteins was affected in the mutants.

The absence of Lat1 lipoylation in the *sam5Δ* mutant was responsible for decreased arginine synthesis because deletion of *LAT1* itself (Fig. 4b) or blocking Lat1 lipoylation by disrupting the lipoate donor protein *gcv3Δ* (Fig. 4b) or lipoate metabolism *lip3Δ* and *lip5Δ* (Supplementary Fig. 4b) all similarly decreased arginine biosynthesis. This is because the acetylation of glutamate step that commits arginine biosynthesis requires acetyl-CoA produced from pyruvate, the activity of which relies on the lipoylation of PDH/lat1.

Unlike *sam5Δ* cells, cellular decreases in SAM in the *met6Δ* or *sam1Δsam2Δ* mutants only constrained lipoate metabolism, with the halted increase in protein lipoylation of both Lat1 and Kgd2 upon methionine or SAM starvation. (Fig. 4k-l). Additional tracing experiments confirmed that MR led to reductions in the newly synthesized lipoate installed on Kgd2 (Fig. 4m-o). The constraint of lipoylation on Lat1 and Kgd2 disrupted the TCA cycle under MR, leading to the reprogramming of acetyl-CoA and α KG from the TCA cycle for the synthesis of arginine. We apologize for the confusion in the

previous manuscript, and we have revised it to improve clarity.

3. *“In Fig 2b and 2c, the levels of acetyl-CoA were significantly decreased in both *sam5Δ* and *Sam1-NLS-GFP*, *sam2Δ*, but arginine synthesis was significantly decreased in *sam5Δ*, and significantly increased in *Sam1-NLS-GFP*, *sam2Δ*, suggesting that the change of arginine synthesis is independent of acetyl-CoA levels.”*

Response: We agree with the reviewer, and we did not suggest that the change in arginine synthesis was dependent on acetyl-CoA levels. In addition to the contrasting observation mentioned by the reviewer, we also showed that the acetyl-CoA surplus by *idh1Δ* was associated with decreased arginine synthesis (Fig. 3c), indicating the increased level of acetyl-CoA was not sufficient to activate arginine biosynthesis.

As an electrophilic, biochemically reactive molecule, the synthesis and consumption of acetyl-CoA are highly compartmentalized and restricted to the local environments^{10,11}. The steady level of total acetyl-CoA does not necessarily reflect its metabolic state in a particular compartment, such as in the mitochondria. Thus, the total cellular acetyl-CoA level is not a good indicator of the activity of arginine synthesis. We performed ¹³C tracing experiments with [U-¹³C₆] glucose and found that mitochondrial acetyl-CoA production from pyruvate dehydrogenase (PDH) is required for the production of acetyl-glutamate which activates the mitochondrial steps for arginine synthesis (Fig. 4b-d).

“Again, in Fig 2, two mitoSAM-manipulating mutants had opposite responses of arginine synthesis to presumed mitoSAM depletion, suggesting the arginine response might be independent of mitoSAM as well.”

Response: We thank the reviewer for the comment. As our response to a similar comment above, we have explained in detail how the lipoylation deficiency of Lat1 in the *sam5Δ* mutant that blocks mitochondrial SAM transport results in decreased arginine synthesis, whereas methionine/SAM starvation in *met6Δ* and *sam1Δsam2Δ* cells constrains both Lat1 and Kgd2 lipoylation, leading to activated arginine synthesis. How the two mitoSAM-manipulating mutants (*sam5Δ* and *SAM1-NLS-GFP/sam2Δ*) affect arginine synthesis is dependent on how the lipoylation of Lat1 and Kgd2 is affected. The *sam5Δ* mutant with blocked mitochondrial SAM transport caused insufficient SAM in the mitochondria to support any lipoylation to occur (Fig. 4e-f). In contrast, the relocation of SAM synthetase to the nucleus in the *SAM1-NLS-GFP/sam2Δ* mutant led to decreased lipoylation of Kgd2 (Fig. 4i and Supplementary Fig. 4j), possibly due to the temporal disruption of SAM supply for the mitochondria. These contrasting effects on protein lipoylation likely convey a regulatory effect to PDH/Lat1 that produces acetyl-CoA from pyruvate and KGH/Kgd2 that operates the TCA cycle. As a result, the activities of arginine synthesis are differently regulated in the mutants. We now made this conclusion more clearly in the revised manuscript. In addition, we also avoided speculative descriptions of SAM changes in the mitochondria in various mutants, which might have brought up confusion.

4. *“In Fig3 b-c, both met6Δgdh1Δ and met6Δidh1Δ showed significantly decreased arginine synthesis compared to met6Δ under MR, with acetyl-CoA not changed in met6Δgdh1Δ and increased in met6Δidh1Δ. This again suggests that acetyl-CoA production is irrelevant to arginine synthesis under MR.”*

Response: This comment is similar to another comment above. We agree with the reviewer that bulk changes in cellular acetyl-CoA levels cannot reflect the activity of arginine synthesis. This is because acetyl-CoA is an electrophilic, biochemically reactive molecule, whose synthesis and consumption are highly compartmentalized and restricted to the local environments^{10,11}. The steady level of total acetyl-CoA does not necessarily indicate its metabolic state in the mitochondria. Based on the findings shown in Fig. 4b-c and Supplementary Fig. 4a-b,d, we conclude that mitochondrial acetyl-CoA produced from pyruvate dehydrogenase (PDH) is likely required for activating the mitochondrial steps for arginine synthesis.

5. *“In Fig 4i, depletion of methionine and SAM had no impact on the lipoylation of Lat1 and Kgd2 in met6Δ and sam1Δsam2Δ, suggesting increased arginine synthesis in the models of met6Δ -methionine and sam1Δsam2Δ-SAM is independent of lipoylation.”*

Response: We now quantified the lipoylation results assayed by western blotting. As shown in Fig. 4k, WT cells without cellular SAM deprivation exhibited a gradual increase in the lipoylation of Lat1 and Kgd2, whereas *met6Δ* and *sam1Δsam2Δ* cells under cellular SAM deprivation conditions exhibited no change or mild decreases in protein lipoylation (Fig.4k). Providing methionine to *met6Δ* cells allowed protein lipoylation to increase (Fig. 4l). These findings indicate that cellular SAM deprivation could constrain protein lipoylation. In addition, we established a mass spectrometry-based assay to trace the synthesis of lipoate with [U-¹³C₆] glucose (Fig. 4m). We found that the amount of lipoic acid released from the Kgd2 protein in this assay was decreased under MR (Fig. 4n), with a greater decrease in the amount of the newly synthesized, ¹³C-fully labeled lipoic acid (Fig. 4o). These findings indicate that MR-induced SAM deprivation constrains the lipoylation of Kgd2. In line with this, the decreased Kgd2 lipoylation in the SAM1-NLS-GFP/*sam2Δ* mutant was associated with increased arginine synthesis. Together, the MR-resulting constraint of Lat1 and Kgd2 lipoylation, with Kgd2 being more sensitive, likely leads to increased arginine synthesis.

6. *“In Fig 6a, leucine was not changed in sam5Δ compared to WT, showing depletion of mitoSAM did not lead to leucine increase.”*

Response: In the previous data, leucine exhibited an increase trend in *sam5Δ* compared to WT, which was not statistically significant. We now repeated this experiment with additional replicates, leucine was indeed increased significantly by ~5 fold in *sam5Δ* (Fig. 6a). Therefore, *sam5Δ* can lead to leucine increase.

As shown above, several major conclusions regarding acetyl-CoA, lipoate, arginine and leucine synthesis in this manuscript are inconsistent with the displayed data. These discrepancies need to be clarified to make all the conclusions valid.

Response: We thank this reviewer's comment. We have addressed each point above,

regarding acetyl-CoA, lipoate, arginine and leucine synthesis. We believe that the quality of this manuscript has been much improved.

Minor points:

7. *“Please provide sequences for the PCR primers used for the construction of strains with enzyme deletions and tags.”*

Response: All primer sequence information is now included in Supplemental Table 3.

8. *“Please provide the sources for the antibodies used in the Western blot.”*

Response: The information for the antibodies used in this study was provided in Reporting Summary with Nature portfolio (Mouse anti-FLAG M2 antibody, Sigma, Cat# F3165; Rabbit anti-G6PDH, Sigma, Cat#A9521; Rabbit anti-Lipoic Acid, Milipore, Cat#437695).

Reviewer #4 (Remarks to the Author):

In this very interesting study the authors investigate metabolic changes due to sulfur scarcity (coming from methionine restriction), and its effect on metabolism. For this, the authors use two auxotrophs: met6Δ (which is a methionine auxotroph), and sam1Δsam2Δ (which specifically cannot make SAM). Under MR, these mutants appear to increase amino acid synthesis (nitrogen assimilation), specifically, arginine, glutamate, glutamine, aspartate and asparagine. The authors note these amino acids can be produced via cataplerotic reactions from TCA cycle. In particular, arginine biosynthesis appears to be increased in MR. The authors find that depleting cytosolic SAM can induce this arginine biosynthesis in mitochondria, which depends on mitoSAM. They term this as mitoSIR (mitoSAM induced response). Upon mitoSAM depletion, lipoylation is affected, which in turn disrupts the TCA cycle leading to an overflow of acetyl CoA, and induces an alternate metabolon for arginine biosynthesis, which is independent of lipoic acid, but coupled to BCAAs. Therefore, the cells undergo a metabolic reprogramming under conditions of methionine restriction (MR). They finally suggest that methionine and leucine can regulate growth in a mitoSAM-dependant manner.

This is an important study, with some strong observations. However, the manuscript suffers from some severe over interpretations of data, as well as over simplifications that can lead to misleading conclusions. A careful revision, with some key experiments can clarify many of these points, and make this a strong manuscript.

Major concerns and comments:

1. *“The main concerns I have with this manuscript are related to interpretations that come from glossing over of contrasting observations, lack of data that unambiguously separates the role of mitochondrial SAM vs cytosolic SAM, and overinterpretations of Arg biosynthesis.”*

Response: We thank the reviewer for this general comment. We have made careful revisions to avoid over-interpretation. We also performed additional experiments and added reasonable explanations for the contrasting observations in *sam5Δ* and *met6Δ* mutants, in which arginine biosynthesis was either repressed or activated. Specifically, repressed arginine biosynthesis in the *sam5Δ* mutant was due to the defective acetylation of glutamate that was caused by a deficiency in PDH/Lat1 lipoylation and a resulting defect of mitochondrial acetyl-CoA production from the oxidation of pyruvate. The MR-induced arginine synthesis in the *met6Δ* mutant, however, was likely due to the differential constraint of PDH/Lat1 and KDH/Kgd2 lipoylation imposed by cellular SAM depletion.

We agree with the reviewer that we do not have direct data supporting the compartment-specific changes in SAM levels due to technical limitations of SAM measurement. We now constructed an additional mutant as suggested by the reviewer, in which SAM synthesis was exclusively relocated into the mitochondrial matrix. We observed repression in arginine synthesis in this mutant, coincident with defective acetyl-CoA production and Lat1 lipoylation. Furthermore, we made careful revisions to the manuscript to avoid ambiguous statements on mitochondrial SAM changes. We will address each point in detail below, and we believe that the quality of this revised manuscript is much improved.

2. *“The premise of the paper is strong wherein the aim is to understand cellular responses to MR, with the idea of using 2 kinds of mutants (met6 and sam1,2), since this can show that cellular SAM levels could signal these cellular responses. However, the data in many figures, specially figure 1, 2 and 4 contradict each other, and no clarifications or explanations are provided.”*

Response: We apologize for the confusion in the previous manuscript. We noticed one comment from the reviewer below: *“the authors suggest that an increase in mitoSAM might activate arg biosynthesis.”* We actually did not propose *“an increase in mitoSAM”* under MR. Speculatively, mitochondrial SAM levels should decrease, or be limited at least, when the total SAM levels drop in the extramitochondrial environment under MR. Bringing up this confusion is probably due to the ambiguity of our previous descriptions of mitoSAM depletion in various mutants, thus making the data of Fig. 1, 2, and 4 to seem contradictory. The metabolic observations in the genetic mutants that manipulate the sources of mitochondrial SAM consistently support that the spatiotemporal supply of SAM for the mitochondria is critical for the regulation of arginine biosynthesis. However, technical limitations preclude us to assess precise changes in SAM in the mitochondrial of various mutants. We now carefully described our observations in cellular SAM changes and avoided the speculation of mitochondrial SAM changes.

3. *“Does mitoSAM actually increase during MR? And does this thereby lead to arg biosynthesis?”*

Response: This comment is related to the comment above. We actually did not

propose “an increase in mitoSAM during MR”. Because of the technical limitations of mitochondrial SAM measurement, we now carefully presented our conclusions based on the actual observation. In particular, MR-induced cellular SAM depletion imposes constraints on the SAM-requiring lipoylation in the mitochondria. The different effects on the lipoylation of Lat1 and Kgd1 lead to the rewiring of the TCA cycle that funnels acetyl-CoA and α -ketoglutarate (α KG) towards the biosynthesis of arginine.

4. “Relatedly: 2. In Fig 2, the authors suggest that mitoSAM is important for arginine biosynthesis using *sam5* Δ and second, depleting cytosolic SAM can increase mitoSAM dependent arginine production. These conclusions come from data using *sam1-NLS-GFP* and *sam2* Δ (Fig2b and d). However, in both the mutants, the total level of cellular SAM remained same (Supplementary Fig 2f). No direct evidence for the change in mitoSAM levels leading to change in arginine levels is provided.”

Response: This comment is related to the two comments above. We now avoided speculative descriptions on mitochondrial SAM changes. We conclude that 1) arginine synthesis requires acetylation of glutamate, a process contingent on the lipoylation of PDH/Lat1 that ensures mitochondrial acetyl-CoA production; and 2) the activation of arginine synthesis in the *Sam1-NLS-GFP/sam2* Δ (*nSam1*) mutant is likely due to the decreased lipoylation of Kgd2 that rewires the TCA cycle for the mitochondrial steps for arginine synthesis. Second, the *mSam1* mutant with SAM synthesis exclusively in the mitochondria matrix exhibited decreased arginine synthesis, with decreases in acetyl-glutamate levels, acetyl-CoA production, and Lat1 lipoylation. Together with the observations in the *nSam1* mutant, these findings suggest that the relocating of SAM synthesis likely alters the temporal SAM supply for the mitochondria, mimicking the (in)sufficiency signals of cytosolic SAM to accordingly modulate the activity of the mitochondrial steps for arginine synthesis.

5. “Relatedly, is this anabolic increase specific to SAM levels decreasing, and independent of methionine increase in this mutant.”

Response: To answer this question, we can look into two sets of mutants. For the first set of mutants *met6* Δ and *sam1* Δ *sam2* Δ , the activities of arginine synthesis were both increased under cellular SAM deprivation conditions, with the methionine content was decreased in *met6* Δ whereas increased in *sam1* Δ *sam2* Δ . Therefore, methionine increase is dispensable for the anabolic increase in arginine. For the second set of mutants *nSam1* and *mSam1*, both mutants exhibited increases in methionine content (Supplementary Fig. 2f-g), but the activity of arginine synthesis was either increased or decreased. Therefore, the anabolic increase in arginine is independent of methionine. Also, because the *nSam1* mutant did not alter total cellular SAM levels, we conclude that SAM decreases are sufficient, but not necessary, for anabolic increases in arginine synthesis.

6. “Does lipoylation and subsequent arg synthesis depend on mitoSAM or cellular SAM decreases? According to fig 1, depleting cellular SAM pools activates arg biosynthesis. In other figures, the authors suggest that an increase in mitoSAM might activate arg

biosynthesis. For all of these it is unclear if and how much SAM accumulates in the mitochondria during methionine restriction, and then the authors need to explain how that is possible.”

Response: This comment is also related to several comments above. Of note, we did not “*suggest that an increase in mitoSAM might activate arg biosynthesis.*” Because of no direct data on mitochondrial SAM levels, we now carefully avoided speculative descriptions of mitochondrial SAM changes. We conclude that MR-resulting cellular SAM depletion can constrain protein lipoylation to alter arginine synthesis. The specific regulation of arginine synthesis is dependent on how lipoylation on the substrate proteins PDH/Lat1 and KDH/Kgd2 would be affected. For example, the MR-induced constraint of both Lat1 and Kgd2 lipoylation in *met6Δ* cells, possibly Kgd2 being more sensitive, resulted in the activation of arginine synthesis. However, in the *sam5Δ* mutant, Lat1 lipoylation was abolished, compromising oxidative decarboxylation of pyruvate to generate acetyl-CoA, a process required for glutamate acetylation to activate arginine synthesis.

7. “The authors also state “.....in contrast, the nuclear SAM mutant showed profound increases in cellular amounts of methionine....” (Supplementary Fig 2d). Authors observe an increase in arginine biosynthesis in the nuclear SAM mutant, but they do not see any difference in SAM levels between the nuclear mutant and sam5Δ (point 2). Separately they also find that methionine is high in the nuclear mutant. Given the now extensive data that methionine acts to signal an anabolic program (likely via cytosolic SAM increase) including synthesis of amino acids such as arginine and lysine, it isn’t clear if the increase in arginine in the nuclear mutant (Fig 2d) is due to SAM depletion in cytosol, or due to increased methionine/SAM in the cell.”

Response: This comment is similar to the 5th comment above. In *met6Δ* and *sam1Δsam2Δ* cells, the activities of arginine synthesis were both increased under cellular SAM deprivation conditions, with the methionine content decreased in *met6Δ* whereas increased in *sam1Δsam2Δ*. This finding indicates that methionine increase is dispensable for the anabolic increase in arginine. Moreover, methionine content was increased in both *nSam1* and *mSam1* cells with unchanged SAM levels, (Supplementary Fig. 2b, f-g), but the activity of arginine synthesis was increased in the *nSam1* mutant and decreased in the *mSam1* mutant. Therefore, the increase in arginine is independent of methionine. Because the *nSam1* mutant did not alter total cellular SAM levels, we conclude that SAM decreases are sufficient, but not necessary, for anabolic increases in arginine synthesis. Further on this, our genetic and metabolic data suggest that arginine synthesis is regulated by the lipoylation status of PDH/Lat1 and KDH/Kgd2, which can be affected by the spatiotemporal supply of SAM for the mitochondria.

“Adding to the previous points made, it becomes key to measure the levels of mitoSAM and cytosolic SAM in these mutants, and also assess if this anabolic increase is specific to reduced (nuclear?) SAM, or independent of methionine increase in this mutant. Alternately, they can also just recalibrate their interpretations accordingly.”

Response: We have addressed whether the arginine increase is (in)dependent of methionine increase above. Our data support that the arginine increase is independent of methionine increase, and SAM decreases are sufficient, but not necessary, for increases in arginine synthesis. We propose that it is the spatiotemporal supply of SAM for the mitochondria that conveys a signal to modulate the synthesis of arginine through a lipoate-dependent mechanism.

With respect to measuring the levels of mitochondrial and cytosolic SAM levels in various mutants, it is technically challenging to do so for several reasons. First, metabolite analysis requires an immediate quenching process to stop metabolism, which is often done with adding methanol solution pre-cooled to -40°C directly to culture medium. However, procedures for the isolation of mitochondria and other organelles are not compatible with this quenching step. If the metabolism is not stopped during mitochondrial extraction, cellular and mitochondrial metabolism will be affected by the extraction steps such as spheroplasting with zymolase, fractionation with high-speed centrifugation, etc. Second, additional technical challenges for measuring the mitochondrial metabolites include the purity and integrity of isolated mitochondria. Extra procedures and buffer systems for isolating high-quality, purer mitochondria will further alter metabolism. Third, from our trial experience, we are concerned that the isolation procedures might cause SAM to transport through mitochondrial membranes via permeases with broad specificities. These technical difficulties preclude accurate assessment of SAM levels in specific compartments in WT and various mutants. We have now carefully stated our findings without speculative descriptions of mitochondrial SAM changes.

8. *“In the section, “MitoSAM depletion alters arginine metabolon.....” . Although the claim is promising. The authors have over interpreted the data that is presented in the study.”*

Response: MitoSAM depletion is indeed not accurate, as we did not measure the SAM level in the *sam5Δ* mutant. We now changed this to “SAM5 deletion alters arginine metabolon”.

9. *“From the figures 1 and 2 one interpretation is that the mito SAM pool/availability is what determines acetylation and arg biosynthesis, regardless of the nucleo/cytoplasmic pool. In the sam5 mutant and the sam1-NLS mutant, there is no cytoplasmic depletion of SAM pool, yet opposite trends are observed for arg biosynthesis, purely based on mito SAM pool. While in the sam1-NLS mutant, SAM synthesis is relocated to the nucleus it that does not rule out the fact that cytoplasm is SAM depleted. An alternate experiment to test this would be to direct/increase SAM synthesis in the mito (target using a mito localization signal) and see responses.”*

Response: We very much thank this suggestion from the reviewer. We have now constructed the *mSam1* mutant strain (mtSam1-GFP/*sam2Δ*) as suggested, in which SAM synthesis was exclusively relocated into the mitochondrial matrix. We observed repression in arginine synthesis in this mutant, coincident with deficiencies in Lat1

lipoylation and acetyl-CoA production. These observations are similar to the findings in *sam5Δ* cells. However, it is curious that *mtSAM1* expression was not sufficient to fully restore lipoylation in *sam5Δ* to WT levels but decreased Lat1 lipoylation compared to WT (Fig. 4f and Supplementary Fig. 4e). This suggests that the temporal control of SAM supply for mitochondria is critical for the process of lipoylation. The ectopic synthesis of SAM in the mitochondria may adapt low Lat1 lipoylation to the metabolic state in the *mSam1* mutant.

10. *“Relatedly, can the authors explain how increased arg can benefit the growth of cells, when arg supplementation does not rescue growth in the MR mutants.”*

Response: In this study, we did not conclude that increased arginine can benefit the growth of cells. For the *met6Δ* and *sam1Δsam2Δ* mutants, cellular SAM deprivation led to arrested cell growth, with increased arginine synthesis. In *sam5Δ*, *nSam1*, and *mSam1* cells, arginine biosynthesis is perturbed in minimal growth medium, with imbalanced intracellular pools of amino acids and growth defects of different severity (Supplementary Fig. 2c-g), highlighting the importance of spatial SAM regulation in orchestrating amino acid homeostasis and cell proliferation. Because arginine was not the only defect in the MR mutants, it was unsurprising that arginine alone was insufficient to fully rescue the growth defects (Supplementary Fig. 2c).

We also performed additional experiments to investigate the physiological benefits of this MR-induced arginine synthesis. We found that MR could evidently promote the survival of *met6Δ* (Fig. 1l and Supplementary Fig. 1k). This survival rate in the *met6Δ* mutant was associated with more potent activation of autophagy (Supplementary Fig. 1l). While the disruption of arginine synthesis by *arg7Δ* reduced the MR-activated autophagy in the *met6Δ* mutant, it also decreased the survival rate of this mutant under MR (Fig. 1l), suggesting that MR-induced arginine synthesis was pro-survival.

11. *“Questions related to how these intermediates are channelized towards arg biosynthesis: I think for this story it would be enough to show how the carbon flux diverts to and away from arg synthesis in the context of SAM levels, which is already done in the flux experiments specially figure 6.”*

Response: We thank the reviewer for this comment. As shown in Fig. 6, we have shown how arginine pathway intermediates were channelized towards arginine biosynthesis in tracing experiments with [U¹³C₆]-glucose.

12. *“The authors state that “.....Because producing these amino acids needs metabolic inputs from mitochondria (Fig. 1h), such as cataplerotic reactions that dispose of TCA cycle intermediates....”. In this case, the sulfur scarcity is created by using a mutant that is not capable of producing methionine or SAM. Both *met6Δ* and *sam1Δsam2Δ* are methionine cycle genes and their deletion will effect mitochondria anyway since the methionine cycle is coupled to the folate cycle as well as redox balance. If mitochondria respond to sulfur scarcity as they claim, more direct evidence of the of TCA flux being diverted to arginine biosynthesis in a sulfur depleted state (WT*

cells grown in no sulfur media) is required.”

Response: We intended to conclude that sulfur scarcity from methionine/SAM deficiency can signal the mitochondria to respond. It is not our intention to expand this to all sulfur scarcity conditions, such as the condition without any sulfur source. To avoid confusion or overinterpretation, we now carefully rephrased this and did not refer to sulfur scarcity in the manuscript.

*13. “Concern: 4. In Fig 3, the authors use *gdh1Δ* to reduce cataplerotically produced alpha-KG, to exclude this from TCA. In this regard, they have compared *met6Δgdh1Δ* to *met6Δ*. However, *Gdh1* is major enzyme that catalyzes conversion of alpha KG to glutamate. The deletion of *Gdh1* will also lower glutamate and other associated metabolites regardless MR. So, the comparison between these two mutants does not provide useful information, and is in fact actively misleading. Alternately, a comparison between *met6Δgdh1Δ* and *gdh1Δ* can help partly clarify if the decrease in upstream metabolites of arginine synthesis is significantly reduced in MR when compared to no MR condition (i.e., *Gdh1Δ*).”*

Response: We respectfully disagree that the comparison between *met6Δ* and *met6Δgdh1Δ* “does not provide useful information, and is in fact actively misleading.” In this study, we established a sustained SAM deprivation regimen in the *met6Δ* mutant where we observed the arginine increase phenotype (Fig. 1). To understand which enzyme is responsible for providing glutamate for the increased arginine, we used a classic genetic method of epistasis analysis, in which three metabolic enzymes that are capable of synthesizing glutamate from αKG were deleted in *met6Δ*, the mutant with the arginine increase phenotype. The findings that *gdh1Δ*, but not *gdh3Δ* or *glt1Δ*, decreased glutamate, acetyl-glutamate, citrulline, carbamoyl phosphate, argininosuccinate, and arginine levels in *met6Δ* cells under MR (Fig. 3b and Supplementary Fig. 3a) indicate that *Gdh1* contributed to providing glutamate for arginine synthesis.

The comparison between *gdh1Δ* and *met6Δgdh1Δ* is to test if *met6Δ* can still boost arginine increase under MR in the absence of *GDH1*. However, we have to characterize if WT and *gdh1Δ* mutants have the arginine increase phenotype under MR first, which is really not the focus of the study. As we discussed in the main text, MR-induced transcriptional feedback likely replenishes methionine and SAM that can veil the discovery of responses otherwise found in the *met6Δ* mutant with a stringent SAM depletion. Having explained this, we still performed metabolic analysis to compare WT, *gdh1Δ*, *met6Δ*, and *met6Δgdh1Δ* cells cultured in minimal medium with 1 mM methionine to these cells 4 h after the switch for methionine starvation. Unlike *met6Δ*, arginine was decreased in WT cells (Supplementary Fig. 3d). Another time course experiment in WT and the *met6Δ* mutant reproduced this finding and revealed this decrease in arginine could be partially restored later on (Supplementary Fig. 1e). Furthermore, we found that *gdh1Δ* could prevent the *met6Δ* mutant under MR from increasing acetyl-glutamate. However, arginine levels could still increase in *gdh1Δ* under MR (Supplementary Fig. 3d), suggesting that other pathways might also

contribute to the synthesis of arginine in this mutant.

14. *“There is no real evidence that Arg2 and Arg5,6 in WT or Arg2 and Arg7 in sam5Δ physically interact/come together. If Arg5,6 interacts with Arg2 in normally, whereas with Arg7 in MR, some direct biochemical data should support this claim (of an alternate metabolon).”*

Response: The study by Abadjieva et al¹² that proposed the arginine metabolon shows that Arg2 physically interacts Arg5,6 and that the acetyl-glutamate synthase (Arg2) activity is only active when it stoichiometrically associates with Arg5,6. In addition, a very recent study published during the reviewing process also confirmed a mitochondrial complexome of Arg2, Arg7, and Arg5,6¹³ (Supplementary Fig. 5c). In the study, native mitochondrial protein assemblies are separated using blue native gel electrophoresis, and the gel lanes are then cut into 245 slices with high-resolution cryo-slicing, followed by quantitative mass spectrometry. After retrieving their published data, we find that Arg2 and Arg7 are completely overlapped, indicating a physical interaction. Arg7 exhibited two peaks, with both peaks co-migrate with and under the peak of Arg2 and Arg5,6. This observation suggests that Arg7 is physically close to Arg2 and Arg5,6, whereas its interaction with Arg2 or Arg5,6 might not be as strong as the interaction between Arg2 and Arg5,6. As such, it is possible that their associations might respond to a metabolic cue, such as SAM depletion. In summary, the published studies support the physical interactions between Arg2, Arg7, and Arg5,6.

In our work, we showed that the C-terminal tagging of both ARG2 and ARG5,6 led to arginine auxotrophy in WT, and the C-terminal tagging of both ARG7 and ARG5,6 led to arginine auxotrophy in the sam5Δ mutant. Of note, the tagging of ARG2, ARG7, or ARG5,6 alone in WT or the sam5Δ mutant does not affect cell growth. These findings underscore the importance of the C-terminal regions of these proteins for interaction, which, however, precludes the use of these strains to perform immunoprecipitation studies. We additionally constructed the N-terminal tagging of ARG2, ARG7, and ARG5,6, and we found the N-terminal tagging of a single gene ARG5,6 could cause arginine auxotrophy, which is likely due to the disruption of mitochondrial targeting sequence. Therefore, it is technically difficult to use traditional methods to examine their interactions without antibodies commercially available.

15. *“In general, while the concept, text and data for figure 5 are interesting, they are tangential (and require more substantial investigation to make a clearer story in itself), and only distract from the primary storyline of this study. I would strongly recommend that the authors think carefully about this section, and whether it should even be in this manuscript.”*

Response: We thank the reviewer for this comment. We consider that the findings in Fig. 5 are logically connected to Figs. 1-4 and have further collaborated in Figs. 6-7. Given the importance of MR-induced acetylation of glutamate in rewiring the TCA cycle towards arginine synthesis, it is critical to understand how acetyl-CoA uncoupled from the TCA cycle under MR will condense with glutamate and enter the acetyl cycle for

arginine synthesis. The investigation in Fig. 5 reveals an alternative acetyl-glutamate synthase Arg7 in preventing wasteful consumption of acetyl-CoA. A serendipitous discovery of the growth of *arg7Δsam5Δ* requiring both arginine and leucine unveils an unexpected metabolic rewiring of acetyl-CoA and pyruvate into an overproduction of leucine, which is further investigated in Fig. 6 and Fig.7. This SAM-responsive programming of the syntheses of arginine and leucine takes place in the mitochondria, underlying an important role of the mitochondria in orchestrating bioenergetic and biosynthetic pathways. We agree that “more substantial investigation” can be done as often true for most published research, particularly for good research that opens doors to new directions. While we are thankful for this suggestion, we hope that the reviewer would appreciate our own intellectual way of organizing this research.

16. *“While Figure 3 suggests that glucose derived acetyl coA and 2-KG are channeled into arg synthesis upon MR, it isn’t clear why the ratio of succinate to 2-KG is higher in the met6 mutant in figure 3e and f. If the flux of acetyl CoA and thereby 2KG into arg synthesis is higher, how is this data explained?”*

Response: The higher ratio of succinate to α KG in the *met6Δ* mutant suggests that the disposal of α KG may be faster than succinate. This is likely due to the disruption of KDH lipoylation and the increased synthesis of arginine, a process requiring the synthesis of glutamate that consequentially pulls α KG out of the TCA. Another interesting note is that the process of arginine synthesis (the last two steps of the conversion of citrulline to arginine) is coupled with the conversion of aspartate to fumarate, a metabolite downstream of succinate in the TCA cycle. With fumarate generated during arginine biosynthesis, the disposal of succinate can be further decreased under MR.

Additional/other comments (some only require clarifications):

17. *“From fig 2, decreasing mitoSAM specifically (see sam5 deletion) decreases arg biosynthesis. In addition, figure 4 suggests that mitoSAM and lipoylation of enzymes are necessary for arg biosynthesis. So how do the authors justify lines 274-276, wherein they say, during MR lipoic acid limitation results in arg biosynthesis by diverting carbon flux from the TCA cycle.”*

Response: We conclude that MR-resulting cellular SAM depletion can constrain protein lipoylation to alter arginine synthesis. The specific regulation of arginine synthesis is dependent on how lipoylation on the substrate proteins PDH/Lat1 and KDH/Kgd2 would be affected. For example, the MR-induced constraint of both Lat1 and Kgd2 lipoylation in *met6Δ* cells, possibly Kgd2 being more sensitive, resulted in the exit of acetyl-CoA and α KG from the TCA cycle for the synthesis of arginine synthesis. However, in the *sam5Δ* mutant, Lat1 lipoylation was abolished, compromising oxidative decarboxylation of pyruvate to generate acetyl-CoA, a process required for glutamate acetylation to activate arginine synthesis. We now avoided speculative statements on mitochondrial SAM levels, as which might have raised confusion.

18. *“The title is misleading. The statement that methionine restriction activates lipoate dependent mitoSIR is an overstatement, since lipoylation itself depends on SAM, and SAM is reduced during MR. Unless the authors find that paradoxically SAM accumulates in the mitochondria during MR, in which case the title becomes more justified. I would suggest an appropriate modification of the title.”*

Response: We have changed the title to “Methionine restriction constrains lipoylation and activates mitochondria for nitrogenic synthesis of amino acids.”

We noticed some confusion and misunderstanding from this comment. “MR activates lipoate-dependent mitoSIR”, we mean that MR activates mitoSIR, and the occurring of mitoSIR is dependent on lipoate metabolism. We do not think that SAM by itself can promote the synthesis of lipoate. Lipoate metabolism is a SAM-requiring process involving multiple steps in the mitochondria. Originating from octanoyl group attached to the carrier protein Gcv3, lipoic acid is matured on this protein via a SAM and iron-sulfur dependent reaction of sulfur insertion. This lipoyl group can be subsequently transferred to the substrate proteins Lat1 and Kgd2. It is very unlikely that abundant SAM in the mitochondria is able to promote this multi-step reaction to make and transfer lipoic acid. However, SAM inadequacy may limit the sulfur insertion step and subsequently perturb the transfer to Lat1 and Kgd2 (Fig. 4).

19. *“Please carefully write sections, without an overuse of adjectives, or overinterpretations. This is true for most of the text.”*

Response: We have made changes in the revised manuscript.

20. *“The fold change in levels of amino acids in the SAM mutant seems to be different in figure 1, panel e and extended figure 1, panel c. Why is this so? Could the authors include the heat map from the extended fig in the main one, so that it becomes clear that no matter what the mutant is, when SAM decreases, amino acids accumulate.”*

Response: The relative fold changes are the same, while the comparison has to be performed within a single heatmap. The color appears different because different matrices of data are compared in the heatmaps. Fig. 1e compares *met6Δ* and *sam1Δsam2Δ*, and Fig. 1g (the previous Supplementary Fig. 1c) compares the *sam1Δsam2Δ* mutant with or without SAM supplementation. To plot heatmaps, data are log-transformed, centered about the mean, and clustered by Spearman rank correlation with Cluster software. Plotting comparisons with different datasets will have a different color appearance in each heatmap, but the relative fold change of the same data set even in different heatmaps remains the same.

The heatmap depicting changes in both mutants (Fig. 1e) was in the main figure. We have now moved a supplemental figure depicting metabolic changes in *sam1Δsam2Δ* with or with SAM to the main figure (Fig. 1g) if the reviewer meant this one.

21. *“In fig1E, though amino acids accumulate in both mutants, it is more in the met6*

mutant. Is there something else that is triggering increased amino acid accumulation than the SAM mutant? Do mitochondria sense SAM as a sulfur starvation signal? For instance, if you supplement a met6 deletion with exogenous SAM, do the levels of amino acids drop to normal?"

Response: The accumulation of amino acids is related to their catabolic and anabolic rates in the cells. We showed that amino acids were actively synthesized under cellular SAM starvation with respective removals of exogenous methionine or SAM, indicating anabolism contributing to the amino acid accumulation phenotype (Fig. 1i). The arrested cell growth will also contribute as their catabolic rates decrease. Notably, in Fig. 1b, the growth of *met6Δ* cells plateaued a bit below 2 of OD₆₀₀ after the withdrawal of methionine, while *sam1Δsam2Δ* cells were arrested near 2 after the withdrawal of SAM. The less cell growth in the *met6Δ* mutant, the less catabolic disposal of amino acids. This likely leads to a higher accumulation of amino acids in the *met6Δ* mutant.

We think that the mitochondria are sensitive to SAM starvation, a form of sulfur scarcity. It is not our intention to expand this to any type of sulfur-limiting conditions. To avoid confusion, we now did not refer to sulfur scarcity in the revised manuscript.

We performed the experiments as suggested by the reviewer. In Supplementary Fig. 1d, exogenous SAM could significantly reduce amino acid accumulation in *met6Δ* cells under MR, albeit not to the extent of the addition of methionine. This is unsurprising because *met6Δ* is a methionine auxotroph. Exogenous methionine restored amino acid metabolism at both anabolic and catabolic levels to meet the demand of cell proliferation, whereas exogenous SAM likely only affected mitoSAM-dependent regulation of amino acid anabolism under MR without the rescuing of amino acid accumulated due to the arrested cell growth. This is consistent with our observations in *met6Δ* and *sam1Δsam2Δ* cells under cellular SAM deprivation conditions.

22. "The authors, claim that the "that the C-terminal tagging of ARG2 and ARG5,6..... which however becomes dispensable in the sam5Δ mutant." It is entirely possible that the growth defect due to C-terminal tagging can occur due to numerous reasons. Not clear at all."

Response: We respectfully disagree with the reviewer. We showed that the tagging of ARG2 or ARG5,6 alone in WT or the *sam5Δ* mutant does not affect cell growth. Only if ARG2 and ARG5,6 were simultaneously C-terminally tagged, the cells became unable to growth in minimal medium. Therefore, this growth defect is synthetic and dependent on the tagging-based disruption of both enzymes. The growth defect was fully restored by exogenous arginine, indicating that this synthetic growth defect was due to the disruption of arginine synthesis. Because that (1) Arg2 and Arg5,6 are known to have physical interaction to form an arginine metabolon¹², and that (2) both enzymes are in a chain reaction of the first two steps for catalysis of acetyl-CoA and glutamate for arginine synthesis, it is reasonable to infer that the C-terminal tagging affects the due interaction between Arg2 and Arg5,6, leading to a defect in arginine synthesis and thereby the growth requiring arginine. This doubly tagging-resultant arginine

auxotrophy was rescued by deletion of *SAM5*. This means, as a logical inferring, that Arg2 and Arg5,6 interaction became not that important in this mutant. In contrast, the C-terminal tagging of both Arg7 and Arg5,6, however, turned *sam5Δ* arginine auxotrophy. (It is noted that the tagging of Arg7 alone did not cause any growth phenotype nor did doubly tagging of Arg7 and Arg5,6 in WT.) Like Arg2, Arg7 possesses acetylglutamate synthase activity. Thus, in the *sam5Δ* mutant, Arg7 likely replaced Arg2 for the interaction with Arg5,6 for catalysis of acetyl-CoA and glutamate, which makes Arg7 and Arg5,6 interaction indispensable for arginine synthesis in *sam5Δ*. These findings indicate that Arg7 is an alternative acetylglutamate synthase in the *sam5Δ* mutant to form arginine metabolon with Arg5,6.

23. *“The Blots needs quantification, especially Fig 4i.”*

Response: We now quantified the blots in Fig. 4k-l (including the previous Fig. 4i).

24. *“The authors should additionally show the total levels of lipoic acid, and see whether it is affected under MR”*

Response: Yeast cells do not produce or use free lipoic acids^{14,15}.

25. *The role of Sam5 in arginine biosynthesis (acetyl cycle) can be included in a schematic or text. Also, for figure 2 and associated text it isn't obvious how SAM levels can directly/indirectly determine acetylation of glutamate or acetyl CoA levels. So the authors can better clarify this.*

Response: Sam5 was included in the previous figures and is also depicted in current Fig. 2e and 7f. The role of Sam5 is a characterized mitochondrial transporter for SAM. This was also described in the main text.

In this study, we now know that *SAM5* deletion abolishes protein lipoylation. This modification is required for the catalytic activity of PDH/Lat1 for the oxidative decarboxylation of pyruvate to acetyl-CoA in the mitochondria. The PDH-catalyzed synthesis of acetyl-CoA is required for the acetylation of glutamate, thus channeling glutamate into arginine synthesis. In Fig.2, we found that acetyl-CoA production is associated with arginine synthesis, yet not knowing how. We further mapped to this step and the other step in the TCA cycle by the tracing of the TCA cycle in Fig.3 and next realized the role of lipoylation in this process in Fig. 4.

26. *“In the text, the authors can cite prior data available for levels of amino acids under methionine or SAM deprivation in prototrophic strains (eg Sutter et al 2013), to especially contextualize why having the auxotrophs in this study is essential.”*

Response: The reference (Petti et al 2011) showing similar increases in levels of amino acids under methionine deprivation in the *met13Δ* strain was cited. The reference mentioned by the reviewer did not use the same condition as ours for cell culture. However, the mentioned reference was cited in the manuscript to highlight the important role of methionine/SAM in cellular regulation. We also added another reference for this purpose.

Being stated in the last paragraph of the Introduction and the first paragraph of the Results, the autonomous replenishment of methionine and SAM from the activation of transcriptional programs may have veiled the discovery of metabolic coordination under MR, so we used the auxotrophs to block the recovery of cellular methionine or SAM in this study. We have performed additional experiments to support this. As shown in Supplementary Fig. 1e, the withdrawal of methionine did not cause amino acid accumulation in WT cells, in which cellular SAM was partially restored over time and less depleted.

Additional note (to contextualize interpretations):

27. "Extensive studies explain metabolic programs in cells when methionine is abundant (the exact opposite scenario of this study), in these same media conditions. Supplementing methionine induces amino acid biosynthesis, particularly arginine and lysine biosynthesis, along with changes in carbon metabolism (see PMIDs 33378328 and 30354837), likely in a SAM dependent manner. The authors are expected to be aware of this, so that they can contextualize their model in this context. Another interesting though peripheral paper (connecting to what methionine does to metabolism) is PMID 32821821

The model figure (final) is a bit euphemistic, and overly simplistic. It depicts that SAM depletion increases arg biosynthesis, but multiple figures show contrasting data in this regard. A more carefully constructed model to really reflect what is happening in the cell is warranted."

Response: We thank this suggestion from the reviewer. In the publications mentioned above (PMIDs 33378328 and 30354837), one used rich medium with lactate as the carbon source and the switch condition to the lactate minimal medium with or without methionine for cell culture, and the other used glucose minimal medium with or without methionine but no switch conditions. Both studies had focused on transcriptional regulation elicited by methionine availability, leading to very important findings of the role of methionine in the coordination of metabolic programs and proliferation control. We have cited these studies in this manuscript to highlight the importance of methionine-induced transcriptional regulation.

In our study, we established a sustained MR regimen to deplete cellular SAM using the *met6Δ* and *sam1Δsam2Δ* mutants. We combined genetics and metabolomics to study the mitochondrial response under cellular SAM deprivation, with a focus on the mitochondrial response directing the metabolic flow of carbon and nitrogen into amino acids orchestrated by a lipoate-dependent mechanism. The fact is that this study on its own is very complicated, partly because of the complex metabolic network itself. If we contextualize the model by incorporating the findings that are not of direct relevance, not investigated in this study but from studies of not the same culture conditions, for instance, adding an arrow by methionine in the schematic to show the transcriptional link to arginine and lysine biosynthesis, it is likely going to be confusing and very distracting from the focus of this work.

We felt sorry that the reviewer found the simple model figure euphemistic, which is likely caused by the confusion the reviewer had. We have addressed each comment in detail to explain contrasting observations, provided further evidence to support the proposed model, and carefully revised the manuscript to improve clarity. We hope that the reviewer will now find that this model is helpful to comprehend this work. Nonetheless, we also generated another model figure (Fig. 7f), with more information about the flow of metabolic pathways and mechanistic regulation.

Reference:

- 1 Marobbio, C. M. T., Agrimi, G., Lasorsa, F. M. & Palmieri, F. Identification and functional reconstitution of yeast mitochondrial carrier for S-adenosylmethionine. *Embo Journal* **22**, 5975-5982, doi:DOI 10.1093/emboj/cdg574 (2003).
- 2 Ye, C., Sutter, B. M., Wang, Y., Kuang, Z. & Tu, B. P. A Metabolic Function for Phospholipid and Histone Methylation. *Molecular cell* **66**, 180-193 e188, doi:10.1016/j.molcel.2017.02.026 (2017).
- 3 Ye, C. *et al.* Demethylation of the Protein Phosphatase PP2A Promotes Demethylation of Histones to Enable Their Function as a Methyl Group Sink. *Molecular cell* **73**, 1115-1126.e1116, doi:10.1016/j.molcel.2019.01.012 (2019).
- 4 Sutter, B. M., Wu, X., Laxman, S. & Tu, B. P. Methionine inhibits autophagy and promotes growth by inducing the SAM-responsive methylation of PP2A. *Cell* **154**, 403-415, doi:10.1016/j.cell.2013.06.041 (2013).
- 5 Tehlivets, O., Scheuringer, K. & Kohlwein, S. D. Fatty acid synthesis and elongation in yeast. *Biochimica et biophysica acta* **1771**, 255-270, doi:10.1016/j.bbalip.2006.07.004 (2007).
- 6 Henry, S. A., Kohlwein, S. D. & Carman, G. M. Metabolism and regulation of glycerolipids in the yeast *Saccharomyces cerevisiae*. *Genetics* **190**, 317-349, doi:10.1534/genetics.111.130286 (2012).
- 7 Morgenstern, M. *et al.* Definition of a High-Confidence Mitochondrial Proteome at Quantitative Scale. *Cell Rep* **19**, 2836-2852, doi:10.1016/j.celrep.2017.06.014 (2017).
- 8 Marobbio, C. M., Agrimi, G., Lasorsa, F. M. & Palmieri, F. Identification and functional reconstitution of yeast mitochondrial carrier for S-adenosylmethionine. *EMBO J* **22**, 5975-5982, doi:10.1093/emboj/cdg574 (2003).
- 9 Agrimi, G. *et al.* Identification of the human mitochondrial S-adenosylmethionine transporter: bacterial expression, reconstitution, functional characterization and tissue distribution. *Biochem J* **379**, 183-190, doi:10.1042/BJ20031664 (2004).
- 10 Ye, C. & Tu, B. P. Sink into the Epigenome: Histones as Repositories That Influence Cellular Metabolism. *Trends in endocrinology and metabolism: TEM*

- 29**, 626-637, doi:10.1016/j.tem.2018.06.002 (2018).
- 11 Walsh, C. T., Tu, B. P. & Tang, Y. Eight Kinetically Stable but Thermodynamically Activated Molecules that Power Cell Metabolism. *Chem Rev* **118**, 1460-1494, doi:10.1021/acs.chemrev.7b00510 (2018).
- 12 Abadjieva, A., Pauwels, K., Hilven, P. & Crabeel, M. A new yeast metabolon involving at least the two first enzymes of arginine biosynthesis - Acetylglutamate synthase activity requires complex formation with acetylglutamate kinase. *Journal of Biological Chemistry* **276**, 42869-42880, doi:DOI 10.1074/jbc.M103732200 (2001).
- 13 Schulte, U. *et al.* Mitochondrial complexome reveals quality-control pathways of protein import. *Nature* **614**, 153-159, doi:10.1038/s41586-022-05641-w (2023).
- 14 Chen, B. B., Foo, J. L., Ling, H. & Chang, M. W. Mechanism-Driven Metabolic Engineering for Bio-Based Production of FreeR-Lipoic Acid in *Saccharomyces cerevisiae* Mitochondria. *Front Bioeng Biotech* **8**, doi:ARTN 965 10.3389/fbioe.2020.00965 (2020).
- 15 Schonauer, M. S., Kastaniotis, A. J., Kursu, V. A., Hiltunen, J. K. & Dieckmann, C. L. Lipoic acid synthesis and attachment in yeast mitochondria. *The Journal of biological chemistry* **284**, 23234-23242, doi:10.1074/jbc.M109.015594 (2009).

Reviewer #1 (Remarks to the Author):

I appreciate the authors additional explanations and experiments. I remain sceptical regarding the relevance outside of budding yeast but agree that this model of MR linking to lipoylation-mediated control of the TCA cycle could be more broadly relevant.

While most of my concerns have been addressed, I do think this remains a difficult and unnecessarily complex manuscript, mostly due to the complexity of the writing style and lack of clear narrative at times. This seems to have become even more problematic in the review process. For instance, there are numerous examples where the language used is ambiguous, non-specific, and unnecessarily complex. A couple of examples:

'This mitochondrial SAM-induced response, namely mitoSIR, promotes cell fitness through the coordination of mitochondrial fuel metabolism with the nitrogenic synthesis of amino acids.'

This does not make grammatical sense? What is really meant by cell fitness? Is it cell survival? What does mitochondrial fuel metabolism really mean in this context?

'Mechanistically, we demonstrate that protein lipoylation constrained by MR conveys the SAM depletion signal to critical allocation of mitochondrial acetyl-CoA for programming nitrogenic anabolism.'

I would suggest that these types of convoluted sentences could be improved by more simply stating the findings.

The second remaining concern relates to the tracing of labelled glucose into lipoate. As presented, it is not an adequately controlled experiment, or at least the relevant controls have not been included. There are no details on the efficiency of the lipoamidase to cleave lipoyl moieties from immunoprecipitated Kgd2. Is all lipoic acid removed? How is the efficiency of the immunoprecipitation normalised between the MR and control conditions? Lipoic acid will vary in its retention time depending on other lipoic acid modifications that occur throughout the enzymatic action of Kgd2 or through additional conjugates, which can also alter retention times. Was lipoate kept in its reduced form?

The lipoylation diagram in Fig 4a is now problematic if lip3 is involved at earlier conjugation steps in these authors hands. I do find this concerning, as the prior literature from multiple groups has been clear on this finding, and similar findings are observed in germline mutations in patients. I accept it is not central to the manuscript but it does need to be addressed, as lipoyl conjugation is not occurring as depicted.

Reviewer #2 (Remarks to the Author):

In the revised manuscript, the authors have addressed my concerns and suggestions. I find the re-submission acceptable for publication.

Reviewer #3 (Remarks to the Author):

The authors have addressed our major concerns. Explanation and new data are provided to clarify issues regarding compartment of SAM and acetyl-CoA, arginine response, lipoylation, and leucine synthesis.

The revised manuscript has downplayed the statements about mitoSAM and mitoSIR that were highly mentioned in the previous version, including removing mitoSIR from the new title. We understand this is partly due to the technical limitations in measuring the mitochondrial SAM levels. Nonetheless, to accurately reflect the conclusions of this study, we suggest that the term "mitoSIR" is also not included in the abstract.

Reviewer #4 (Remarks to the Author):

The authors have done a very good job with their revision, and the manuscript is also easier to read with these reorganisations. In the current form of the manuscript, I am fairly convinced that not only is this mechanistic finding important (and opens new directions of research), but the now revised title merits the data presented. This is an excellent study.

I have only two concerns remaining, the first fairly straightforward and minor, the second that requires thoughtful addressing (and perhaps an attempted experiment).

1) The experiments with the nuclear localised Sam1 and mSam1 enzyme forms are compelling, with the expected directionality of SAM and SAH increases. However, what was the extent of overexertion of Sam1 in the nucleus (how much enzyme was localised there)? Please quantify and indicate this information, since it is important for the conclusions. The conclusions should be tempered accordingly.

2) The data presented in the revision make me think that the implied conclusions regarding mitochondrial SAM are inferred correctly. However, again, there is no indication of changes in mitochondrial SAM and/or SAH (and all their accounting comes from bulk readings), so the compartmentalisation changes remain implied (and is not reflected in the text appropriately). The authors state however that it is technically impossible to measure mitochondrial SAM. This would not be true, since it is quite straightforward to isolate mitochondria fairly rapidly in yeast cultures (several methods now exist), and SAM does not exchange quickly between the mitochondria and outside. The quantification of SAM is by M/S, so does not require much mitochondria material. It remains possible that the changes in mitochondrial SAM and/or SAH may not be large, and still the conclusions are correct, but without attempting this direct experiment, this statement cannot in all fairness be made, and the conclusions will have to be tempered accordingly.

Minor note: there are a few small typos in the text. Please proof carefully.

We would like to thank the reviewers for their constructive comments. The point-to-point responses to the comments are as follows:

Reviewer #1 (Remarks to the Author):

1. *"I appreciate the authors additional explanations and experiments. I remain sceptical regarding the relevance outside of budding yeast but agree that this model of MR linking to lipoylation-mediated control of the TCA cycle could be more broadly relevant."*

Response: We appreciate the reviewer's agreement with "this model of MR linking to lipoylation-mediated control of the TCA cycle could be more broadly relevant." The relevance outside of budding yeast requires additional studies in different systems. My laboratory has a follow-up study of this MR effector mechanism in mouse models of human disease. Our preliminary data support that the beneficial effect of MR is associated with the reprogramming of mitochondrial metabolism. We are currently testing if the lipoylation-mediated control of the TCA cycle might be an underlying mechanism.

"While most of my concerns have been addressed, I do think this remains a difficult and unnecessarily complex manuscript, mostly due to the complexity of the writing style and lack of clear narrative at times. This seems to have become even more problematic in the review process. For instance, there are numerous examples where the language used is ambiguous, non-specific, and unnecessarily complex. A couple of examples:

'This mitochondrial SAM-induced response, namely mitoSIR, promotes cell fitness through the coordination of mitochondrial fuel metabolism with the nitrogenic synthesis of amino acids.' This does not make grammatical sense? What is really meant by cell fitness? Is it cell survival? What does mitochondrial fuel metabolism really mean in this context?

'Mechanistically, we demonstrate that protein lipoylation constrained by MR conveys the SAM depletion signal to critical allocation of mitochondrial acetyl-CoA for programming nitrogenic anabolism.'

I would suggest that these types of convoluted sentences could be improved by more simply stating the findings."

Response: We regret that there were problems with the language. We thoroughly went through the text to simplify the language and clarify some sections.

2. *"The second remaining concern relates to the tracing of labelled glucose into lipoate. As presented, it is not an adequately controlled experiment, or at least the relevant controls have not been included. There are no details on the efficiency of the lipoamidase to cleave lipoyl moieties from immunoprecipitated Kgd2. Is all lipoic acid removed? How is the efficiency of the immunoprecipitation normalised between the MR and control conditions? Lipoic acid will vary in its retention time depending on other lipoic acid modifications that occur throughout the enzymatic action of Kgd2 or through additional conjugates, which can also alter retention times. Was lipoate kept in its reduced form?"*

Response: The assay of tracing lipoate with labeled glucose was rigorously validated. We now included the relevant control experiments in Supplementary Figs. 4m-o and added relevant descriptions in Methods. Kgd2 immunoprecipitation was highly efficient and the same in cells with or without MR (Supplementary Fig. 4m). Lipoamidase treatment alone can efficiently remove lipoate from the immunoprecipitated Kgd2 protein, however, with a slight degradation of the protein itself (Supplementary Fig. 4n). To further ensure the efficient release of lipoic acid (and to avoid the time retention issue mentioned by the reviewer), we performed trypsin treatment before lipoamidase treatment. This slightly increased lipoic acid release (Supplementary Fig. o). Because lipomidase alone could free most lipoic acid from Kgd2, this increase indicates that the release of lipoic acid with trypsin and lipoamidase treatment was very efficient. Also, lipoate was extracted and examined under an acid condition, so it is likely kept in its reduced form.

“The lipoylation diagram in Fig 4a is now problematic if lip3 is involved at earlier conjugation steps in these authors hands. I do find this concerning, as the prior literature from multiple groups has been clear on this finding, and similar findings are observed in germline mutations in patients. I accept it is not central to the manuscript but it does need to be addressed, as lipoyl conjugation is not occurring as depicted.”

Response: We appreciate that the reviewer considers this as not central to the manuscript. In the previous comment, the reviewer suggested using the *lip3Δ* mutant to study lipoylation change under MR. We found that *LIP3* deletion did not result in lipoylation accumulation in the carrier protein Gcv3, therefore precluding the proposed idea of using this mutant. We established an isotope-based assay for tracing lipoate synthesis and successfully showed the MR-restricted effect on lipoate metabolism.

The difference from the prior literature and observations in germline mutations in patients may be explained by several reasons. First, it is likely that the Lip3 protein itself is required for the sulfur insertion step. If the germline mutations in patients are the loss of function mutations, it may allow lipoate synthesis to occur but not the following transfer, thus leading to the accumulation of lipoated GCSH, the human homolog of Gcv3. Second, it is also possible that the regulation of lipoate metabolism is different among organisms, even in yeasts with different genetic backgrounds. We hypothesize, for instance, that the lipoated Gcv3 protein might not be stable and could be readily degraded in the *lip3Δ* mutant cells in the prototrophic background. These ideas need further investigation and are beyond the scope of this study.

Reviewer #2 (Remarks to the Author):

“In the revised manuscript, the authors have addressed my concerns and suggestions. I find the re-submission acceptable for publication.”

Response: We thank the reviewer for accepting our responses.

Reviewer #3 (Remarks to the Author):

“The authors have addressed our major concerns. Explanation and new data are

provided to clarify issues regarding compartment of SAM and acetyl-CoA, arginine response, lipoylation, and leucine synthesis. The revised manuscript has downplayed the statements about mitoSAM and mitoSIR that were highly mentioned in the previous version, including removing mitoSIR from the new title. We understand this is partly due to the technical limitations in measuring the mitochondrial SAM levels. Nonetheless, to accurately reflect the conclusions of this study, we suggest that the term “mitoSIR” is also not included in the abstract.”

Response: We thank the reviewer for accepting our responses. We now removed “mitoSIR” from the abstract.

Reviewer #4 (Remarks to the Author):

“The authors have done a very good job with their revision, and the manuscript is also easier to read with these reorganisations. In the current form of the manuscript, I am fairly convinced that not only is this mechanistic finding important (and opens new directions of research), but the now revised title merits the data presented. This is an excellent study.”

Response: We appreciate the nice words from the reviewer.

“I have only two concerns remaining, the first fairly straightforward and minor, the second that requires thoughtful addressing (and perhaps an attempted experiment). 1) The experiments with the nuclear localised Sam1 and mSam1 enzyme forms are compelling, with the expected directionality of SAM and SAH increases. However, what was the extent of overexpression of Sam1 in the nucleus (how much enzyme was localised there)? Please quantify and indicate this information, since it is important for the conclusions. The conclusions should be tempered accordingly.”

Response: The NLS signal was appended to the C-terminus of SAM1 at its chromosomal locus. SAM1 expressed under its native promoter in the Sam1-NLS-GFP/sam2Δ cells was not overexpressed and exhibited similar protein levels as Sam1-GFP/sam2Δ cells and the other strains (Supplementary Fig. 2b).

“2) The data presented in the revision make me think that the implied conclusions regarding mitochondrial SAM are inferred correctly. However, again, there is no indication of changes in mitochondrial SAM and/or SAH (and all their accounting comes from bulk readings), so the compartmentalisation changes remain implied (and is not reflected in the text appropriately). The authors state however that it is technically impossible to measure mitochondrial SAM. This would not be true, since it is quite straightforward to isolate mitochondria fairly rapidly in yeast cultures (several methods now exist), and SAM does not exchange quickly between the mitochondria and outside. The quantification of SAM is by M/S, so does not require much mitochondria material. It remains possible that the changes in mitochondrial SAM and/or SAH may not be large, and still the conclusions are correct, but without attempting this direct experiment, this statement cannot in all fairness be made, and the conclusions will have to be tempered accordingly.”

Response: In the previous comments, this reviewer and Reviewer 2 suggested the

measurement of mitochondrial SAM levels. We had performed the measurement before our initial submission (and repeated it during the second submission), and we thoroughly explained to both reviewers why the measurement of mitochondrial SAM is technically challenging. We did not state that “it is technically impossible to measure mitochondrial SAM”. In fact, my laboratory routinely isolates mitochondria for another research focus in the lab which is on the spatial transfer of phospholipids into the mitochondria, and my laboratory also has expertise in mass spectrometry. As we responded previously, we have attempted to use different methods (zymolase-based and glass bead-based disruption of yeast cells) to isolate mitochondria for metabolic analysis, but thus far none of these methods passed our quality control, with mutants that are known to alter mitochondrial metabolites in principle. It is not because we are reluctant to attempt this important experiment (which we actually did), but it is because of technical limitations that preclude accurately measuring SAM in isolated mitochondria. Without the measurement of actual SAM levels in the mitochondria, we had thoroughly tempered our conclusion regarding mitochondrial SAM, and Reviewer 2 and Reviewer 3 have accepted this point.

We are cautious about the validity of SAM measurement in isolated mitochondria, as this study investigates the functional communication of SAM between the mitochondria and the extra-mitochondrial environment. Here we summarize the technical limitations and explain why the obtained results were inconclusive as follows:

1. Mitochondrial SAM measurement is incompatible with the vital quenching step in the metabolomics workflow. To comply with the community requirements (<https://www.nature.com/articles/s41592-021-01197-1>), the rapid stopping, or quenching, of metabolism is required. Only by doing so, extracted metabolites are quantitatively reflective of the endogenous metabolite levels in the original living cell. However, procedures for isolating mitochondria and other organelles are incompatible with the quenching step. This becomes a particular problem in yeast, as an additional spheroplasting step with zymolase for removal of the cell wall.
2. During the isolation of mitochondria, cells cannot maintain the same metabolic environment under which methionine is replete. For example, cells are switched from a growth condition with methionine supplementation to a methionine depletion condition. During mitochondrial isolation, the cells grown with methionine are basically gone through a condition similar to methionine depletion.
3. The reviewer’s comment that “SAM does not exchange quickly between the mitochondria and outside” was not substantiated. In fact, if SAM does not readily exchange quickly between the mitochondria and the cytosol, we could not observe the ready response of the mitochondria to SAM depletion. Furthermore, the mutant with SAM synthesis exclusively in the mitochondria is viable for growth (Supplementary Fig. 2d), so mitochondrial SAM synthesized in the mitochondria can likely transport back to the cytosol by certain unknown mechanisms, as many essential SAM-dependent methyltransferases are present in the cytosol. With these complications for mitochondrial SAM transport, the measurement of mitochondrial SAM without quenching is problematic.

4. In our attempted experiments, we observed comparable levels of SAM in WT and *sam5Δ* cells in isolated mitochondria. However, this result is inconclusive because SAM metabolism and mitochondrial SAM transport were not stopped by quenching during mitochondrial isolation. This observation is also inconsistent with the role of Sam5 as the only mitochondrial SAM transporter. A possible explanation is that SAM might diffuse out from the WT mitochondria and/or diffuse into the *sam5Δ* mitochondria via non-specific permeases. This may occur under circumstances where mitochondrial SAM transport and metabolism were not stopped and in an uncontrolled status during the isolation. For example, WT mitochondrial SAM levels are higher than the extraction buffer, and SAM levels in the *sam5Δ* mitochondria are lower than in its cytosol. SAM may diffuse and move down a concentration gradient during mitochondrial extraction, such as the spheroplasting step. This SAM diffusion can diminish an existing difference in mitochondrial SAM content in WT and *sam5Δ* cells.
5. In the studies identifying the mitochondrial transporter Sam5/SAMC, reconstituted proteoliposomes were used to examine the activity and specificity of SAM transport. Mitochondrial SAM depletion was not tested in the transporter mutants using isolated mitochondria. Instead, the study provides supporting genetic evidence. Mitochondrial SAM is required for respiratory growth and biotin synthesis. They show that the *sam5Δ* mutant exhibits growth defects on non-fermentable and biotin-free medium. The rescue of these growth defects by the ectopic expression of SAM synthetase Sam1 in the mitochondria is indicative of mitochondrial SAM insufficiency in *sam5Δ* cells. In our study, we used the same genetic approach to confirm the depletion of mitochondrial SAM in the *sam5Δ* mutant (Supplementary Fig. 4g-h). In addition, we showed that another mitoSAM-requiring process in the mitochondria, lipoate biosynthesis, was abolished in *sam5Δ* and rescued by mitochondrial *SAM1* expression (Fig. 4e-f). These genetic results support mitochondrial SAM depletion in the *sam5Δ* mutant. This also reveals that SAM measured in isolated mitochondria is not reflective of the endogenous mitochondrial SAM levels.

“Minor note: there are a few small typos in the text. Please proof carefully.”

Response: We thoroughly went through the text to correct typos.